# Machine Learning Hamiltonians are Accurate Energy-Force Predictors

Seongsu Kim [1]   Chanhui Lee [2]   Yoonho Kim [1]   Seongjun Yun [2]   Honghui Kim [1]   Nayoung Kim [1]
Changyoung Park [3]   Sehui Han [3]   Sungbin Lim [† 3 4]   Sungsoo Ahn [† 1]

## Abstract

Recently, machine learning Hamiltonian (MLH) models have gained traction as fast approximations of electronic structures such as orbitals and electron densities, while also enabling direct evaluation of energies and forces from their predictions. However, despite their physical grounding, existing Hamiltonian models are evaluated mainly by reconstruction metrics, leaving it unclear how well they perform as energy–force predictors. We address this gap with a benchmark that computes energies and forces directly from predicted Hamiltonians. Within this framework, we propose QHFlow2, a state-of-the-art Hamiltonian model with an SO(2)-equivariant backbone and a two-stage edge update. QHFlow2 achieves $40\%$ lower Hamiltonian error than the previous best model with fewer parameters. Under direct evaluation on MD17/rMD17, it is the first Hamiltonian model to reach NequIP-level force accuracy while achieving up to $20\times$ lower energy MAE. On QH9, QHFlow2 reduces energy error by up to $20\times$ compared to MACE. Finally, we demonstrate that QHFlow2 exhibits consistent scaling behavior with respect to model capacity and data, and that improvements in Hamiltonian accuracy effectively translate into more accurate energy and force computations.

## 1. Introduction

Recently developed machine learning Hamiltonians (MLH; Schütt et al., 2019; Unke et al., 2021a; Li et al., 2022; Gong et al., 2023; Yu et al., 2023b;a; Li et al., 2025; Luo et al., 2025; Xia et al., 2025) predict the Kohn-Sham Hamilto-

nian from molecular geometry. Unlike machine-learning interatomic potentials (MLIPs; Unke et al., 2021b), they predict an electronic-structure object that provides access to quantities such as electron density while also enabling downstream evaluation of energies and forces.

However, prior work has focused primarily on using predicted Hamiltonians to accelerate self-consistent field (SCF) convergence (Kim et al., 2025; Liu et al., 2025), leaving it unclear how accurately MLH models perform when energies and forces are computed directly from the predicted Hamiltonian. This question is particularly pressing given a recent study reporting that even strong Hamiltonian models yield far less accurate energy predictions than MLIP approaches (Kaniselvan et al., 2025).

Motivated by this gap, we re-examine the capabilities of MLH under direct evaluation and ask:

> "How far have MLH models progressed, and can they achieve the energy–force accuracy required for practical atomistic simulations?"

We answer this question by first establishing a benchmark that directly evaluates energies and forces from predicted Hamiltonians. Our analysis reveals that existing Hamiltonian models do not meet the MLIP level of accuracy, which serves as a practical reference for downstream applications.

To close this gap, we propose QHFlow2, which improves both scalability and robustness. First, we redesign the equivariant architecture for scalability by adapting an SO(2)-equivariant backbone based on eSEN (Fu et al., 2025), whose efficient edge updates are well suited for modeling orbital interactions. Second, we introduce a two-stage pair update that improves the robustness of the off-diagonal Hamiltonian blocks to cutoff and radial-basis choices. In addition, we extend a standard Hamiltonian benchmark to directly evaluate energies and forces from predicted Hamiltonians, facilitating controlled comparisons with MLIP baselines. Finally, we analyze the scaling behavior of Hamiltonian error and downstream energy and force accuracy with respect to model size and training-set size.

Under the extended benchmark, we show that sufficiently

[†]Corresponding authors [1]Korea Advanced Institute of Science and Technology [2]Department of Artificial Intelligence, Korea University [3]LG AI Research [4]Department of Statistics, Korea University. Correspondence to: Sungbin Lim <sungbin@korea.ac.kr>, Sungsoo Ahn <sungsoo.ahn@kaist.ac.kr>.

*Proceedings of the 43rd International Conference on Machine Learning*, Seoul, South Korea. PMLR 306, 2026. Copyright 2026 by the author(s).

Source code: https://github.com/seongsukim-ml/QHFlow2.

accurate Hamiltonian prediction yields accurate energies and forces under direct evaluation. On MD17 and rMD17, QHFlow2 achieves up to $20\times$ lower energy MAE than NequIP (Batzner et al., 2022), while reaching force accuracy comparable to MLIPs for the first time among Hamiltonian predictors. On QH9, QHFlow2 reduces energy error by up to $20\times$ relative to MACE (Batatia et al., 2022) and improves upon EquiformerV2 (Liao et al., 2024). Notably, QHFlow2 further reduces Hamiltonian prediction error by 40–50% relative to the prior state of the art while using roughly half the parameters and achieving $2.8\times$ faster inference.

Finally, we demonstrate that QHFlow2 exhibits consistent scaling behavior with respect to model capacity and data, and that improvements in Hamiltonian accuracy effectively translate to downstream energy and force predictions. Together, these results establish Hamiltonian models as a viable approach for atomistic modeling, combining accurate energies, competitive forces, and access to electronic-structure objects within a single framework.

Overall, our contributions are as follows:

- We propose QHFlow2, a scalable MLH that modernizes an eSEN-based SO(2)-equivariant backbone for Hamiltonian prediction and introduces a two-stage edge update, improving the robustness of cutoff and radial-basis choices while achieving the best accuracy with fewer parameters.

- We establish a unified benchmark that directly computes total energies and analytic forces from predicted Hamiltonians, and we construct an rMD17 benchmark with recomputed Hamiltonians, energies, and forces to facilitate controlled comparisons with MLIP baselines.

- We study model and data scaling in Hamiltonian prediction under direct evaluation, showing that increased scale consistently reduces Hamiltonian error and improves downstream energy and force accuracy.

## 2. Related Work

**Machine learning Hamiltonians (MLHs).** Deep learning approaches predict Kohn-Sham Hamiltonians from molecular geometries using equivariant message passing with orbital-based matrix representations (Schütt et al., 2019; Unke et al., 2021a; Li et al., 2022; Gong et al., 2023). Later work improves scalability through efficient equivariant operations and training objectives (Yu et al., 2023a;b; Li et al., 2025; Luo et al., 2025; Xia et al., 2025; Yu et al., 2025). Beyond regression, Kim et al. (2025) introduces a generative formulation that treats Hamiltonians as structured objects.

Existing studies mainly evaluate Hamiltonian models using matrix and orbital-energy metrics, while some evaluate their utility within DFT workflows, such as improving SCF convergence (Liu et al., 2025). More recently, Kaniselvan et al. (2025) used Hamiltonian models as a pre-training objective to transfer representations to downstream energy and force models via multi-head predictors. In contrast, our work evaluates Hamiltonian predictors through direct downstream accuracy, reporting energies and analytic forces computed from predicted Hamiltonians under a unified benchmark.

**Machine learning interatomic potentials (MLIPs).** MLIPs regress energies and forces directly from molecular configurations, typically using message passing over local atomic graphs and permutation-invariant aggregation (Deringer et al., 2019; Unke et al., 2021b). Directional and angular message passing improve the modeling of anisotropic interactions (Gasteiger et al., 2020; 2021), while group theory-based equivariant updates further enhance accuracy across molecular benchmarks (Batzner et al., 2022; Liao & Smidt, 2023; Batatia et al., 2022). Overall, most MLIPs focus on predicting and evaluating energy and force. In contrast, Hamiltonian models predict an electronic-structure object, thereby enabling the computation of energies, forces, and other observables simultaneously.

**Equivariant architectures and efficiency.** Rotational equivariance is a central inductive bias in molecular modeling and is widely used in both MLIPs and Hamiltonian prediction. State-of-the-art approaches often stem from SO(3)-equivariant updates with group-theoretic features and symmetry-preserving tensor operations (Thomas et al., 2018; Brandstetter et al., 2021; Geiger & Smidt, 2022), but the training cost can grow rapidly with higher-order angular features. Recent architectures mitigate this overhead by performing equivariant updates in local frames and exploiting SO(2) structure (Passaro & Zitnick, 2023; Liao et al., 2024; Fu et al., 2025). Building on this line of work, we adopt an SO(2) backbone to improve scalability, in addition to enhancing accuracy in Hamiltonian prediction.

## 3. Method

### 3.1. Overview: Machine Learning Hamiltonians

In this section, we overview machine learning Hamiltonian (MLH) models and how predicted Hamiltonians can be used to evaluate electronic-structure quantities, energies, and forces. We refer to Section A for a tutorial on Kohn-Sham (KS) density functional theory (DFT).

In KS DFT (Hohenberg & Kohn, 1964; Kohn & Sham, 1965), the KS Hamiltonian $\mathbf{H} \in \mathbb{R}^{B \times B}$ is the central quantity that determines the ground-state electronic structure, where $B$ is the number of basis functions under a fixed DFT setup. We propose QHFlow2, which predicts $\mathbf{H}$ directly from molecular geometry $\mathcal{M}$ and then evaluates the density

matrix $\mathbf{D} \in \mathbb{R}^{B \times B}$, which uniquely determines the electron density $\rho$, as well as the KS total energy $E_{\mathrm{KS}}$ and atomic forces $\mathbf{F}$ from the prediction. Our pipeline follows:

$$\mathcal{M} \to \hat{\mathbf{H}} \to \hat{\mathbf{D}} \to (\hat{E}_{\mathrm{KS}}, \hat{\mathbf{F}}), \tag{1}$$

where $\hat{\mathbf{H}}$, $\hat{\mathbf{D}}$, $\hat{E}_{\mathrm{KS}}$, and $\hat{\mathbf{F}}$ denote the predicted Hamiltonian, density matrix, energy, and forces, respectively.

**Electronic structure computation.** The Kohn-Sham Hamiltonian provides a direct interface to downstream electronic structure quantities through the Roothaan–Hall equation (Roothaan, 1951; Hall, 1951):

$$\hat{\mathbf{H}}\mathbf{C} = \mathbf{S}\mathbf{C}\boldsymbol{\epsilon}. \tag{2}$$

where $\mathbf{S} \in \mathbb{R}^{B \times B}$ is the overlap matrix computed for the molecule $\mathcal{M}$ under the fixed DFT setup, $\mathbf{C} \in \mathbb{R}^{B \times n}$ contains the molecular-orbital coefficients for $n$ orbitals, and $\boldsymbol{\epsilon} \in \mathbb{R}^{n \times n}$ is the diagonal matrix of the orbital energies.

**Energies and forces evaluation from $\hat{\mathbf{H}}$.** Given a predicted Hamiltonian $\hat{\mathbf{H}}$ and the overlap matrix $\mathbf{S}$, we solve for the orbital coefficients $\mathbf{C}$ satisfying Equation (2) under the same DFT setup with the reference dataset. Assuming a restricted closed-shell Kohn-Sham (RKS; Szabo & Ostlund, 1996) setting, we assign two electrons to every occupied orbital following eigenvalue ordering. With $N_e$ electrons, we define the occupation vector $\mathbf{o} \in \mathbb{R}^B$ with

$$o_p = \begin{cases} 2, & p \le \lceil N_e/2 \rceil, \\ 0, & p > \lceil N_e/2 \rceil, \end{cases} \tag{3}$$

where $o_p$ is the $p$-th element of the occupation vector $\mathbf{o}$. This computes the approximation of the density matrix $\mathbf{D} = \mathbf{C}\,\mathrm{diag}(\mathbf{o})\,\mathbf{C}^\top \in \mathbb{R}^{B \times B}$. The density matrix $\mathbf{D}$ uniquely determines the electron density $\rho$. We evaluating the KS energy functional $E_{\mathrm{KS}}[\rho]$ and the corresponding forces as analytic gradients $\mathbf{F}_i = -\nabla_{\mathbf{r}_i} E_{\mathrm{KS}}[\rho]$. Further calculation details are provided in Section A.1.

### 3.2. Learning Hamiltonians with Flow Matching

We train our model using equivariant flow matching (Lipman et al., 2023; Song et al., 2023; Kim et al., 2025). Our model learns to map the molecule $\mathcal{M}$ to the target Hamiltonian matrix $\mathbf{H}$ via the time-dependent vector field $v_t^\theta$, which satisfies rotational equivariance. To this end, we draw an initial noisy $\mathbf{H}_0$ from a rotation invariant distribution and sample intermediate $\mathbf{H}_t$ where $t \sim \mathcal{U}(0,1)$.

$$\mathbf{H}_t = (1-t)\mathbf{H}_0 + t\mathbf{H}. \tag{4}$$

Then the model predicts a terminal Hamiltonian $\mathbf{H}_1^\theta$ as

$$\mathbf{H}_1^\theta = f_\theta\big(\mathcal{M}, t, \mathbf{H}_t; \{\mathbf{M}_k\}\big), \tag{5}$$

where $\{\mathbf{M}_k\}$ is an auxiliary matrix feature, such as the overlap matrix or the initial Hamiltonian, which can be easily computed from $\mathcal{M}$. This induces the vector field

$$v_t^\theta(\mathbf{H}_t, \mathcal{M}) = \frac{\mathbf{H}_1^\theta - \mathbf{H}_t}{1-t}, \tag{6}$$

which enables distributional probability path modeling of Hamiltonians. Corresponding training minimizes the conditional flow-matching objective

$$\mathcal{L}_{\mathrm{CFM}} = \mathbb{E}\left[ \left\| v_t^\theta(\mathbf{H}_t, \mathcal{M}) - \frac{\mathbf{H} - \mathbf{H}_0}{1-t} \right\|^2 \right], \tag{7}$$

encouraging $v_t^\theta$ to match the oracle velocity along the interpolation path. Further details about group and representation theory, invariant prior, and the training and inference algorithms, are in Sections B, C.2 and C.4 respectively.

### 3.3. QHFlow2 Architecture

We introduce **QHFlow2**, an MLH model that improves scalability by combining an SO(2)-equivariant message-passing backbone with an explicit two-stage update of pairwise features for Hamiltonian construction while preserving an SO(3)-equivariant feature structure. Figure 1 provides an overview of the method. Background on the rotational equivariance of Hamiltonian matrices and the associated representation theory is given in Section B. Model inputs and parameterization follow Equation (5).

**Embeddings.** We initialize atom-wise node features from the molecule $\mathcal{M}$, the flow time $t$, and the intermediate Hamiltonian state $\mathbf{H}_t$. The atomic number $Z_i$ is embedded in an SO(3)-invariant feature $\mathbf{a}_i = g_{\mathrm{atom}}(Z_i)$, and the $t$ is embedded using a sinusoidal encoding $\boldsymbol{\tau}_t = g_{\mathrm{time}}(t)$.

The intermediate Hamiltonian $\mathbf{H}_t$ is embedded in an SO(3)-equivariant irreducible representation (irrep) feature using a matrix encoder:

$$\mathbf{f}_H = g_{\mathrm{matrix}}^{(H)}(\mathbf{H}_t) = \{\mathbf{f}_H[\ell]\}_{\ell=0}^{\ell_{\max}}, \tag{8}$$

where $\mathbf{f}_H[\ell] \in \mathbb{R}^{2\ell+1}$ denotes the irreps component of degree $\ell$. The additional conditioning matrices $\{\mathbf{M}_k\}$ are embedded in the same manner, i.e., $\mathbf{f}_k = g_{\mathrm{matrix}}^{(k)}(\mathbf{M}_k) = \{\mathbf{f}_k[\ell]\}_{\ell=0}^{\ell_{\max}}$.

The initial node feature is then constructed by combining only the invariant ($\ell = 0$) components,

$$\begin{aligned} \mathbf{x}_i^{(0)}[0] &= g_{\mathrm{mix}}\big(\mathbf{a}_i, \boldsymbol{\tau}_t, \mathbf{f}_H[0], \{\mathbf{f}_k[0]\}\big), \\ \mathbf{x}_i^{(0)}[\ell] &= \mathbf{f}_H[\ell] + \sum_k \mathbf{f}_k[\ell], \qquad \ell > 0. \end{aligned} \tag{9}$$

Here, $g_{\mathrm{mix}}$ produces atom-dependent scalar features via an MLP. All ($\ell > 0$) components are initialized by summing the matrix embeddings while preserving equivariance.

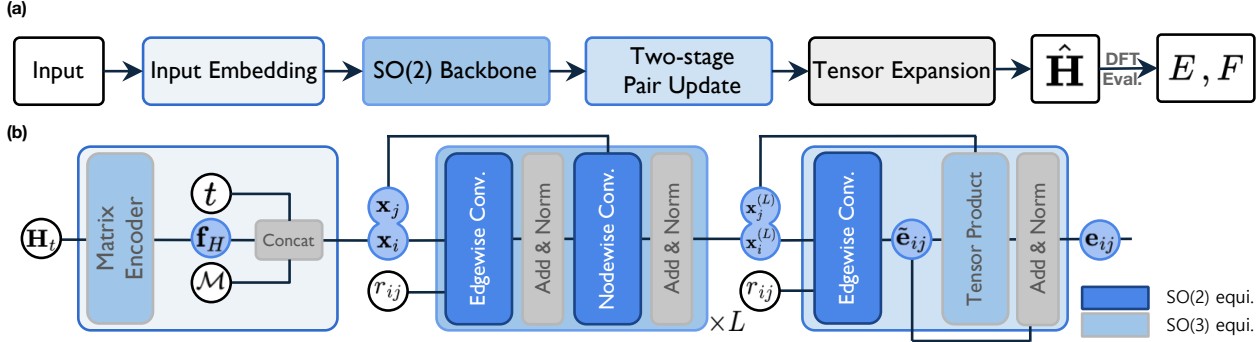

*Figure 1.* **QHFlow2 overall workflow.** Given the molecular structure $\mathcal{M}$, flow time $t$, and an intermediate Hamiltonian state $\mathbf{H}_t$, QHFlow2 applies an SO(2) backbone, a two-stage pairwise update, and construct Hamiltonian via tensor expansion for energy and force evaluation. *(a)* Pipeline overview. *(b)* Detailed architecture of the matrix encode, the SO(2) backbone, and the two-stage pair update.

**SO(2)-equivariant message-passing backbone.** We adopt the SO(2)-equivariant edgewise–nodewise update scheme of eSEN (Fu et al., 2025). The backbone takes the initial node embeddings $\mathbf{x}_i^{(0)}$ as input and stacks $L$ layers, each alternating an SO(2)-equivariant edgewise convolution and a nodewise update.

At layer $l$, we construct an SO(2)-equivariant edgewise message on the fly for each directed pair $(i, j)$ using relative geometry. With $i$-th atomic position $\mathbf{r}_i$ and $\mathbf{r}_{ij} = \mathbf{r}_j - \mathbf{r}_i$ and $r_{ij} = \|\mathbf{r}_{ij}\|$, define the neighborhood $\mathcal{N}(i) = \{ j \neq i \mid r_{ij} < r_c \}$ with cutoff $r_c$. The distance $r_{ij}$ is encoded by a radial basis expansion

$$\phi_{\text{rbf}}(r_{ij}) = \left[\, b_1(r_{ij}), \ldots, b_M(r_{ij}) \,\right], \quad (10)$$

where $\{b_m\}_{m=1}^M$ are smooth radial basis functions (e.g., Gaussian or Bessel bases) modulated by a smooth cutoff envelope (Gasteiger et al., 2020).

An edgewise message is computed as

$$\mathbf{m}_{ij}^{(l)} = \text{SO2Conv}_{\text{pair}}^{(l)}\left([\mathbf{x}_i^{(l)}\|\mathbf{x}_j^{(l)}], \phi_{\text{rbf}}(r_{ij})\right), \quad (11)$$

where $[\cdot\|\cdot]$ denotes concatenation. This is the SO(2)-equivariant edgewise convolution of eSEN (Fu et al., 2025); it operates in a local SO(2) frame while maintaining the node features in an SO(3)-irrep decomposition.

The edgewise messages are then aggregated over neighbors and used to update node features via a nodewise equivariant feed-forward block.

$$\mathbf{x}_i^{(l+1)} = \mathbf{x}_i^{(l)} + \text{SO2Conv}_{\text{self}}^{(l)}\left(\mathbf{x}_i^{(l)}, \sum_{j\in\mathcal{N}(i)} \mathbf{m}_{ij}^{(l)}\right). \quad (12)$$

Here, $\text{SO2Conv}_{\text{self}}^{(l)}$ is the nodewise feed-forward block of eSEN, applied independently to each SO(3) irrep component, thereby preserving the SO(3)-equivariant feature structure. Stacking $L$ yields the final node features $\{\mathbf{x}_i^{(L)}\}$.

**Two-stage pair update.** The Hamiltonian decomposes into on-atom terms (diagonal blocks) and inter-atomic couplings (off-diagonal blocks). To model these couplings explicitly, we construct pairwise features $\mathbf{e}_{ij}$ for atom pairs $(i, j)$ from the final node features $\{\mathbf{x}_i^{(L)}\}$ produced by the backbone. Unlike the backbone, which performs local message passing within a cutoff, this module forms pair features for all atom pairs to parameterize off-diagonal blocks.

We first initialize an intermediate pair feature using an SO(2)-equivariant pair encoder conditioned on a radial distance embedding,

$$\tilde{\mathbf{e}}_{ij} = \text{SO2Conv}_{\text{pair}}\left([\mathbf{x}_i^{(L)}\|\mathbf{x}_j^{(L)}], \phi_{\text{rbf}}(r_{ij})\right), \quad (13)$$

where $\phi_{\text{rbf}}(r_{ij})$ denotes the radial basis embedding. For consistency, we use the same cutoff and radial basis configuration as in the backbone for this initialization.

In practice, we find that relying solely on this radial initialization can make Hamiltonian prediction sensitive to the choice of radial-basis hyperparameters. To reduce this dependence, we apply an additional refinement that injects pairwise interactions through an SO(3)-equivariant tensor-product coupling:

$$\mathbf{e}_{ij} = \tilde{\mathbf{e}}_{ij} + g_{\text{ref}}\left(\tilde{\mathbf{e}}_{ij}, \mathbf{x}_i^{(L)}, \mathbf{x}_j^{(L)}, r_{ij}\right), \quad (14)$$

where $g_{\text{ref}}$ is an SO(3)-equivariant tensor-product layer that refines $\tilde{\mathbf{e}}_{ij}$ using node irreps and pair geometry. Implementation details are provided in Section C, and ablations are reported in Table 10.

**Equivariant vector-to-matrix Hamiltonian construction.** The backbone produces final node features $\{\mathbf{x}_i^{(L)}\}$ and explicit pairwise features $\{\mathbf{e}_{ij}\}$, which we convert into the predicted Hamiltonian $\hat{\mathbf{H}}$ in an atom-centered orbital basis. Let $B_i$ denote the number of basis functions (orbitals) associated with atom $i$; in our setting, $B_i$ is determined by

the atomic species $Z_i$. We construct $\hat{\mathbf{H}}$ as a block matrix $\hat{\mathbf{H}} = [\hat{\mathbf{H}}_{ij}]_{i,j=1}^N$ with $\hat{\mathbf{H}}_{ij} \in \mathbb{R}^{B_i \times B_j}$.

For each atom pair $(i, j)$, we first select an equivariant feature to represent their interaction,

$$\mathbf{q}_{ij} = \begin{cases} \mathbf{x}_i^{(L)}, & i = j, \\ \mathbf{e}_{ij}, & i \neq j, \end{cases}$$

and map it to the orbital-pair block via a learnable tensor expansion module adapted from QHNet (Yu et al., 2023b),

$$\tilde{\mathbf{H}}_{ij} = \mathcal{E}_{Z_i, Z_j}(\mathbf{q}_{ij}), \quad \mathcal{E}_{Z_i, Z_j} : \{\mathbf{q}[\ell]\}_{\ell=0}^{\ell_{\max}} \to \mathbb{R}^{B_i \times B_j}.$$

The expansion $\mathcal{E}_{Z_i, Z_j}$ is an SO(3)-equivariant linear readout that converts irreps components into the matrix block of the corresponding atom-type pair; details are provided in Section C. Finally, we enforce Hermiticity by symmetrization:

$$\hat{\mathbf{H}} = \frac{1}{2}(\tilde{\mathbf{H}} + \tilde{\mathbf{H}}^\top). \tag{15}$$

**Relation to prior eSEN/SO(2)-based MLHs.** QHFlow2 shares high-level design choices with two recent MLHs but differs in how Hamiltonian blocks are produced. HELM (Kaniselvan et al., 2025) uses an eSEN-style SO(2) backbone and maps node and edge embeddings to Hamiltonian blocks via gated nonlinearities, without an additional SO(3) refinement of pair features. QHNetV2 (Yu et al., 2025) applies a single SO(2) refinement stage in local frames for off-diagonal blocks. QHFlow2 instead uses a *two-stage pair update* that combines an SO(2) initialization with an SO(3) tensor-product refinement, which we find improves off-diagonal accuracy and reduces sensitivity to the edge cutoff (Table 10). A detailed comparison is in Section D.3.

## 4. Experiments

In this section, we evaluate QHFlow2 under a direct downstream benchmark for Hamiltonian prediction. We also construct an rMD17 benchmark by re-computing reference Hamiltonians that are consistent with the provided energy and force labels. Our evaluation computes total energies and analytic forces from predicted Hamiltonians using PySCF (Sun et al., 2018), enabling controlled comparisons with MLIP baselines on molecular dynamics benchmarks. We report downstream energy and force accuracy (Figures 2 and 3), Hamiltonian prediction accuracy (Tables 1 and 2 and Figure 4), scaling behavior of MLH models (Figures 5 and 6), and runtime analysis (Figure 7). Additional analyses ( Sections G.8 and G.9) examine the energy-force error gap, finite-difference force consistency, MO-basis error structure, and local PES fidelity and Hessian quality.

### 4.1. Experimental Setup

**Benchmarks and labels.** We evaluate two benchmarks: an *MD benchmark* that combines MD17 and rMD17, and

*QH9* (Chmiela et al., 2017; Schütt et al., 2019; Christensen & Von Lilienfeld, 2020; Ruddigkeit et al., 2012; Ramakrishnan et al., 2015; Yu et al., 2023a). QH9 is derived from QM9 and provides DFT labels, including Hamiltonians and orbital quantities. For QH9, we use an in-distribution split aligned with the QM9 molecule set (*QH9-stable-id*) for MLIP reference comparisons and additional out-of-distribution and dynamic splits to assess generalization in Hamiltonian prediction. MD benchmark labels are computed with PBE/def2-SVP and QH9 labels with B3LYP/def2-SVP. For rMD17, we compute reference Hamiltonians as well as downstream energies and forces, and we will release the benchmark upon publication. Full dataset construction and split definitions are provided in Section E.1.

**Training and comparison.** On the MD benchmark, we retrain both MLIP baselines and Hamiltonian predictors using the same train/validation/test splits and the same downstream evaluation pipeline. On QH9, all Hamiltonian prediction methods are trained and evaluated on identical splits and reference Hamiltonians. For MLIP comparisons on QH9, we report published QM9 results from the corresponding papers as reference points. Training configurations are provided in Sections E and F.

**Models and baselines.** For MLIP baselines, we compare against DimeNet, GemNet-T, and NequIP on the MD benchmark (Gasteiger et al., 2020; 2021; Batzner et al., 2022), and against MACE and Equiformer/EquiformerV2 on QH9 (Batatia et al., 2022; Liao & Smidt, 2023; Liao et al., 2024). For Hamiltonian baselines, we compare against QHNet, QHNetV2, WANet, SPHNet, and QHFlow (Yu et al., 2023b; 2025; Luo et al., 2025; Kim et al., 2025).

**Evaluation metrics.** We report Hamiltonian MAE ($H$), occupied orbital energy mean absolute error (MAE) ($\epsilon_{\text{occ}}$), coefficient similarity ($S_c$), energy and forces MAE ($E$,$F$). For QH9, we additionally report HOMO, LUMO, and the HOMO–LUMO gap. For downstream evaluation, MLIPs predict energies and forces directly, whereas Hamiltonian predictors output $\hat{\mathbf{H}}$ from which energies and forces are computed. Detailed metric definitions and implementation details are provided in Section E.2.

### 4.2. Energy and Force Accuracy of MLH

**MD benchmark.** Figure 2 reports energy and force MAE under direct evaluation from predicted Hamiltonians on the MD benchmark. Overall, Hamiltonian-based evaluation provides competitive energy accuracy on MD trajectories, while force accuracy has been more challenging for prior Hamiltonian predictors. Within this setting, QHFlow2 substantially reduces energy error across all six systems and, for the first time among Hamiltonian predictors, reaches force accuracy

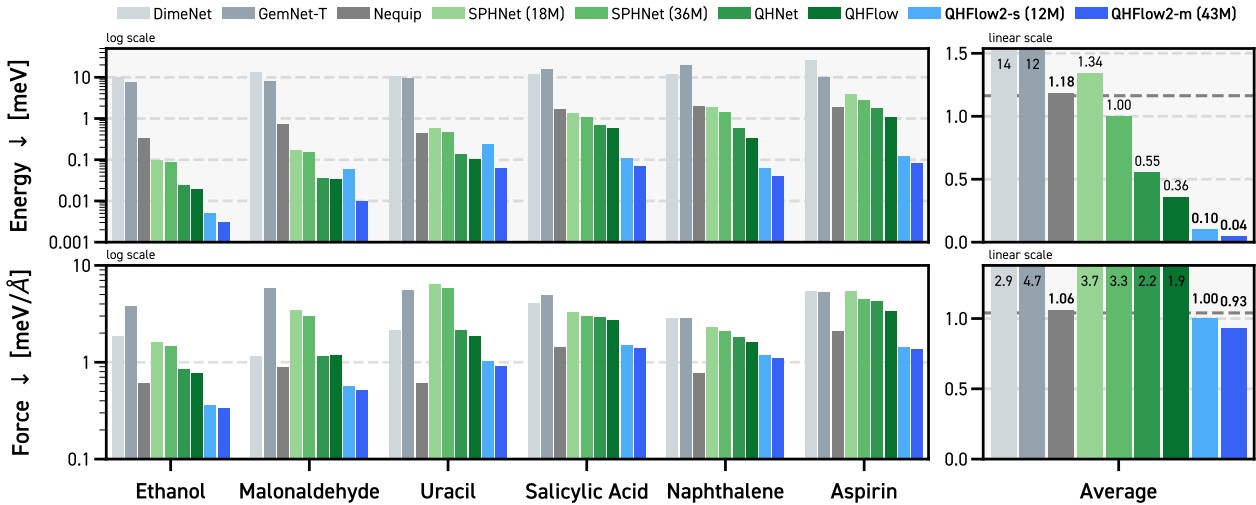

*Figure 2.* **Energy and force accuracy under direct evaluation on the MD benchmark.** We report mean absolute errors (MAE) of total energy (top) and forces (bottom) computed from predicted Hamiltonians on six molecular systems. Gray bars denote MLIP baselines, green bars denote prior Hamiltonian predictors, and blue bars denote QHFlow2. All methods use the same data splits and evaluation setup. Overall, the results show that Hamiltonian-based direct evaluation yields competitive energy and force accuracy on MD trajectories, with prior Hamiltonian models narrowing the gap to MLIPs and QHFlow2 providing the strongest improvements. The right-hand panels summarize mean MAE across the six systems, and the dashed line marks the NequIP reference. QHFlow2 attains low energy error and, for the first time among Hamiltonian predictors, reaches NequIP-level force accuracy. Each number is reported in Table 15.

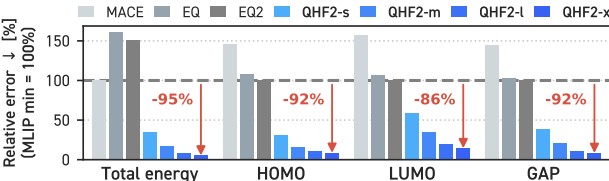

*Figure 3.* **Relative performance of MLIP baselines and QHFlow2 on QH9.** Relative errors for total energy and orbital quantities are normalized by the best MLIP result of baselines for each quantity. EQ and EQ2 denote Equiformer and EquiformerV2, respectively. MLIP numbers are taken from published results on QM9 as reference points, whereas QHFlow2 is trained and evaluated on QH9-stable-id. All exact values are reported in Section G.

comparable to NequIP under this benchmark. We connect these downstream trends to Hamiltonian reconstruction and orbital-level errors in Section 4.3. We additionally compare against eSEN (Fu et al., 2025) on rMD17, with full results provided in Section G.5.

**QH9 benchmark.** Figure 3 reports total-energy and orbital-energy errors on QH9. We train and evaluate all Hamiltonian predictors on the QH9-Stable-id split, which shares the same set of molecules as QM9 but uses fewer training molecules. We compare against MLIP reference results reported on QM9, including MACE, Equiformer, and EquiformerV2, as well as Hamiltonian prediction baselines. QHFlow2 achieves strong accuracy across all reported quantities, including total energy, HOMO, LUMO, and the HOMO-LUMO gap. We report exact values, including force

errors, for Hamiltonian predictors in Table 2. MLIP force comparisons are omitted because the corresponding QM9 references do not report force metrics. We further analyze scaling with model and data size in Figure 6.

### 4.3. Hamiltonian Prediction Accuracy

**MD benchmark.** Table 1 compares QHFlow2 with prior Hamiltonian models on the MD benchmark. We report Hamiltonian MAE ($H$), occupied orbital energy MAE ($\epsilon_{occ}$), and coefficient similarity ($S_c$), together with downstream energy and force MAE ($E$, $F$) computed from each predicted Hamiltonian. QHFlow2 improves both Hamiltonian accuracy and orbital-level metrics, and these improvements coincide with lower energy errors while preserving competitive force accuracy. This trend suggests that reducing Hamiltonian errors and improving the accuracy of the occupied orbital quantities are both associated with better downstream accuracy on MD trajectories.

**QH9 benchmark.** Table 2 compares Hamiltonian and orbital-energy metrics on QH9, along with energy and force errors. We report both the QH9-stable and QH9-dynamic subsets, with detailed splits depicted in Section E.1. We observe that QHFlow2 performs strongly across Hamiltonian and orbital metrics, and that these gains are accompanied by improved downstream accuracy.

**Predicted Hamiltonians as SCF initial guesses on QH9.** Although SCF acceleration is not a primary goal, we use

*Table 1.* **Hamiltonian prediction model performance on the MD benchmark.** Comparison of Hamiltonian prediction methods on six molecular systems. Reported metrics include Hamiltonian MAE $H$ ($\mu$Ha), occupied orbital energy MAE $\epsilon_{\text{occ}}$ ($\mu$Ha), coefficient similarity $S_c$ (%), and derived total-energy and force MAE ($E$ in meV; $F$ in meV/Å) computed from each predicted Hamiltonian. QHFlow2 rows are highlighted with skyblue, and the best result in each column is shown in **bold**. Size suffixes -s/-m denote increasing model capacity.

| | | Ethanol (9 atoms) | | | | | malondialdehyde (9 atoms) | | | | | Uracil (12 atoms) | | | | |
| --- | --- | --- | --- | --- | --- | --- | --- | --- | --- | --- | --- | --- | --- | --- | --- | --- |
| Model | Param. | $H\downarrow$ | $\epsilon_{\text{occ}}\downarrow$ | $S_c\uparrow$ | Energy$\downarrow$ | Force$\downarrow$ | $H\downarrow$ | $\epsilon_{\text{occ}}\downarrow$ | $S_c\uparrow$ | Energy$\downarrow$ | Force$\downarrow$ | $H\downarrow$ | $\epsilon_{\text{occ}}\downarrow$ | $S_c\uparrow$ | Energy$\downarrow$ | Force$\downarrow$ |
| *Regression* | | | | | | | | | | | | | | | | |
| SPHNet | 18M | 11.39 | 30.29 | 99.99 | 0.097 | 1.609 | 8.81 | 25.47 | 99.99 | 0.166 | 3.430 | 11.42 | 34.63 | 99.99 | 0.576 | 6.399 |
| SPHNet | 36M | 9.97 | 26.74 | 99.99 | 0.086 | 1.469 | 7.55 | 22.14 | 99.99 | 0.155 | 2.989 | 9.66 | 29.52 | 99.98 | 0.454 | 5.846 |
| QHNet | 27M | 6.57 | 36.53 | **100.00** | 0.024 | 0.854 | 4.52 | 23.90 | 99.99 | 0.035 | 1.145 | 4.71 | 39.85 | 99.97 | 0.139 | 2.139 |
| *Flow-based* | | | | | | | | | | | | | | | | |
| QHFlow | 30M | 5.60 | 31.16 | **100.00** | 0.019 | 0.775 | 4.26 | 23.61 | 99.98 | 0.034 | 1.166 | 3.99 | 34.31 | 99.99 | 0.106 | 1.869 |
| **QHFlow2-s** | 14M | 3.21 | 14.51 | **100.00** | 0.005 | 0.355 | 2.66 | 11.32 | **100.00** | 0.058 | 0.568 | 2.52 | 16.91 | 99.99 | 0.234 | 1.012 |
| **QHFlow2-m** | 43M | **2.63** | **13.56** | **100.00** | **0.003** | **0.338** | **2.03** | **10.33** | **100.00** | **0.010** | **0.514** | **1.85** | **15.88** | **100.00** | **0.062** | **0.901** |

| Model | Param. | Salicylic Acid (16 atoms) | | | | | Naphthalene (19 atoms) | | | | | Aspirin (21 atoms) | | | | |
| --- | --- | --- | --- | --- | --- | --- | --- | --- | --- | --- | --- | --- | --- | --- | --- | --- |
| *Regression* | | | | | | | | | | | | | | | | |
| SPHNet | 18M | 8.17 | 54.72 | 99.82 | 1.337 | 3.264 | 6.09 | 37.98 | 98.83 | 1.905 | 2.315 | 10.74 | 98.05 | 99.51 | 3.972 | 5.347 |
| SPHNet | 36M | 7.25 | 49.39 | 99.84 | 1.106 | 2.960 | 5.36 | 36.30 | 98.90 | 1.418 | 2.076 | 8.87 | 80.98 | 99.61 | 2.773 | 4.456 |
| QHNet | 27M | 5.45 | 58.66 | 99.81 | 0.701 | 2.909 | 3.79 | 40.78 | 98.95 | 0.589 | 1.816 | 6.42 | 86.92 | 99.61 | 1.828 | 4.239 |
| *Flow-based* | | | | | | | | | | | | | | | | |
| QHFlow | 30M | 4.76 | 54.10 | 99.83 | 0.589 | 2.701 | 3.22 | 36.07 | 99.04 | 0.335 | 1.610 | 4.99 | 71.50 | 99.70 | 1.071 | 3.357 |
| **QHFlow2-s** | 14M | 2.60 | 31.61 | **99.91** | 0.110 | 1.482 | 2.42 | **28.30** | **99.28** | 0.063 | 1.171 | 2.33 | 32.98 | 99.88 | 0.124 | 1.423 |
| **QHFlow2-m** | 43M | **2.05** | **30.31** | **99.91** | **0.070** | **1.381** | **2.11** | 28.74 | 99.27 | **0.040** | **1.106** | **1.68** | **31.14** | **99.90** | **0.084** | **1.350** |

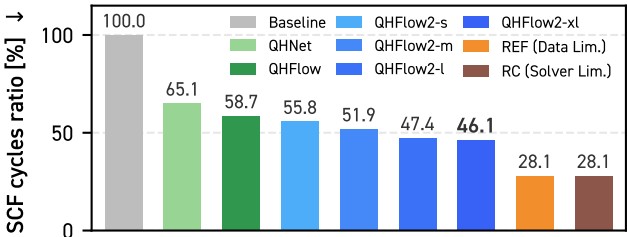

*Figure 4.* **Relative reduction of SCF cycles on QH9.** We report the SCF cycle ratio (lower is better): SCF iterations to converge when initializing with a predicted Hamiltonian $\hat{\mathbf{H}}$, normalized by the MinAO baseline (100%). All SCF/DFT settings are fixed; only the initialization is changed. Stronger predictors reduce SCF cycles and approach REF, which initializes with the dataset Hamiltonian $\mathbf{H}$ (data limit), while RC initializes with a converged Hamiltonian recomputed under our evaluation settings (solver limit).

it to assess the utility of predicted Hamiltonians as initial guesses in iterative KS-DFT. We run SCF on 300 molecules from the QH9-stable-iid test split using a fixed solver and convergence criteria in Section E, and report the SCF-cycle ratio relative to the default MinAO initialization. As shown in Figure 4, stronger predictors consistently reduce SCF cycles. We include two references: *REF*, which initializes SCF with the dataset Hamiltonian $\mathbf{H}$, and *RC*, which initializes SCF with a Hamiltonian that has converged under our evaluation settings. We use *REF* as a practical upper bound in our evaluation stack and report how closely each model approaches it under identical SCF settings. Table 17 in Section G.3 additionally reports per-stage wall-clock timings and end-to-end speedup, where QHFlow2-l achieves a $2.18\times$ speedup over the baseline.

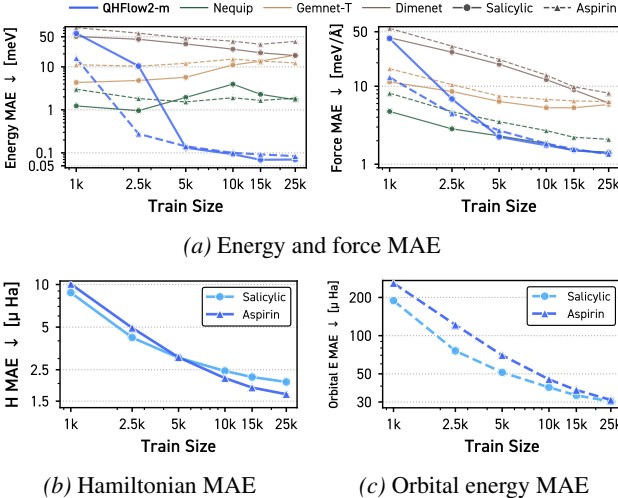

*(a)* Energy and force MAE

*(b)* Hamiltonian MAE    *(c)* Orbital energy MAE

*Figure 5.* **Data scaling on the MD benchmark.** *(a)* Energy and force MAE versus training set size for QHFlow2-m and MLIP baselines on salicylic acid and aspirin. *(b)* Hamiltonian MAE versus training set size for QHFlow2-m. *(c)* Occupied orbital energy MAE versus training set size for QHFlow2-m. Improved Hamiltonian accuracy consistently accompanies improved orbital, energy, and force accuracy.

### 4.4. Scaling Behavior

**Data scaling on the MD benchmark.** Figure 5 reports energy and force MAE ($E$, $F$) as a function of training set size, together with Hamiltonian MAE ($H$) and occupied orbital energy MAE ($\epsilon_{\text{occ}}$). As the training set increases, QHFlow2 exhibits consistent reductions in Hamiltonian and orbital errors, accompanied by corresponding improvements in en-

*Table 2.* **Hamiltonian prediction model performance on QH9 benchmark.** Comparison of Hamiltonian prediction methods across QH9-Stable (id/ood) and QH9-Dynamic (geo/mol) splits. Reported metrics include Hamiltonian MAE $H$ ($\mu$Ha), occupied orbital energy MAE $\epsilon_{\text{occ}}$ ($\mu$Ha), coefficient similarity $S_c$ (%), and orbital errors for HOMO, LUMO, and the HOMO–LUMO gap (all in $\mu$Ha), as well as total-energy and force MAE ($E$ in meV; $F$ in meV/Å) computed from each predicted Hamiltonian. QHFlow2 rows are highlighted with skyblue, and the best result in each column is shown in **bold**. Size suffixes -s/-m/-l/-xl denote increasing model capacity.

| Model | Param. | \multicolumn{8}{c}{QH9-stable-id} | \multicolumn{8}{c}{QH9-stable-ood} |
| | | $H(\downarrow)$ | $\epsilon_{\text{occ}}(\downarrow)$ | $S_c(\uparrow)$ | HOMO($\downarrow$) | LUMO($\downarrow$) | GAP($\downarrow$) | E($\downarrow$) | F($\downarrow$) | $H(\downarrow)$ | $\epsilon_{\text{occ}}(\downarrow)$ | $S_c(\uparrow)$ | HOMO($\downarrow$) | LUMO($\downarrow$) | GAP($\downarrow$) | E($\downarrow$) | F($\downarrow$) |
|---|---|---|---|---|---|---|---|---|---|---|---|---|---|---|---|---|---|
| *Regression* | | | | | | | | | | | | | | | | | |
| QHNet (w/o init) | 20M | 77.7 | 963.5 | 94.8 | 1546.3 | 18257.3 | 17822.6 | 434.63 | 2868.3 | 69.7 | 885.0 | 93.0 | 1046.0 | 25848.8 | 25370.1 | 427.22 | 66.1 |
| QHNet | 20M | 46.0 | 459.6 | 98.3 | 470.6 | 2781.2 | 2967.9 | 36.92 | 16.3 | 38.1 | 170.5 | 97.8 | 255.6 | 1326.8 | 1359.7 | 13.18 | 6.3 |
| WANet | 29M | 80.0 | 833.6 | 96.9 | – | – | – | – | – | – | – | – | – | – | – | – | – |
| SPHNet | 30M | 45.5 | 334.3 | 97.8 | – | – | – | – | – | 43.3 | 186.4 | 98.2 | – | – | – | – | – |
| QHNetV2 (w/o init) | - | 31.5 | 417.9 | 98.6 | – | – | – | – | – | 23.0 | 165.9 | 97.7 | – | – | – | – | – |
| *Flow-based* | | | | | | | | | | | | | | | | | |
| QHFlow | 28M | 23.0 | 119.7 | 99.5 | 179.5 | 438.0 | 553.9 | 2.37 | 4.7 | 20.0 | 84.5 | 99.0 | 130.7 | 321.2 | 395.8 | 8.78 | 3.3 |
| **QHFlow2-s** | 12M | 16.3 | 94.9 | 99.6 | 148.4 | 304.7 | 411.4 | 1.42 | 3.1 | 12.5 | 54.6 | 99.4 | 72.8 | 182.9 | 200.2 | 0.97 | 1.6 |
| **QHFlow2-m** | 43M | 9.2 | 59.2 | 99.7 | 77.7 | 175.7 | 216.0 | 0.69 | 2.1 | 7.8 | 42.1 | 99.5 | 49.5 | 134.8 | 147.3 | 0.51 | 1.7 |
| **QHFlow2-l** | 183M | 5.9 | 45.7 | 99.8 | 50.5 | 103.3 | 119.6 | 0.32 | 1.4 | 5.2 | 31.9 | 99.7 | 33.7 | 99.0 | 105.9 | 0.30 | 0.8 |
| **QHFlow2-xl** | 990M | **4.6** | **39.4** | **99.8** | **40.3** | **73.2** | **87.7** | **0.21** | **1.2** | **4.1** | **26.8** | **99.7** | **28.8** | **97.8** | **105.7** | **0.20** | **0.7** |
| | | \multicolumn{8}{c}{QH9-dynamic-geo} | \multicolumn{8}{c}{QH9-dynamic-mol} |
| *Regression* | | | | | | | | | | | | | | | | | |
| QHNet (w/o init) | 20M | 88.4 | 1170.5 | 93.6 | 2040.1 | 23269.4 | 22408.0 | 448.63 | 119.3 | 121.4 | 5554.4 | 86.0 | 4352.8 | 53505.1 | 50424.9 | 841.74 | 206.5 |
| WANet | 29M | 74.7 | 416.6 | 99.7 | – | – | – | – | – | – | – | – | – | – | – | – | – |
| SPHNet | 30M | 52.2 | 100.9 | 99.1 | – | – | – | – | – | 108.2 | 1724.1 | 91.5 | – | – | – | – | – |
| QHNetV2 (w/o init) | - | 35.6 | 270.0 | 98.8 | – | – | – | – | – | 49.0 | 629.6 | 97.4 | – | – | – | – | – |
| *Flow-based* | | | | | | | | | | | | | | | | | |
| QHFlow | 28M | 25.9 | 103.1 | 99.6 | 175.2 | 425.2 | 547.3 | 4.28 | 5.2 | 45.9 | 442.6 | 98.7 | 479.7 | 1344.7 | 1605.0 | 8.93 | 11.4 |
| **QHFlow2-s** | 12M | 15.4 | 53.8 | 99.7 | 67.3 | 430.1 | 447.5 | 1.34 | 2.9 | 29.7 | 261.3 | 99.2 | 263.2 | 644.0 | 772.9 | 4.29 | 9.0 |
| **QHFlow2-m** | 43M | 8.7 | 42.5 | 99.9 | 51.8 | 113.5 | 144.0 | 0.51 | 1.7 | 22.0 | 240.1 | 99.4 | 207.2 | 378.8 | 468.6 | 2.51 | 8.1 |
| **QHFlow2-l** | 183M | 5.2 | 28.2 | 99.9 | 30.6 | 88.8 | 101.7 | 0.22 | 1.1 | 17.5 | 214.5 | 99.5 | 179.2 | 261.7 | 325.0 | 1.85 | 7.0 |
| **QHFlow2-xl** | 990M | **3.7** | **22.8** | **100.0** | **21.0** | **44.4** | **50.6** | **0.15** | **0.9** | **15.9** | **202.3** | **99.6** | **167.4** | **219.5** | **278.1** | **1.66** | **6.5** |

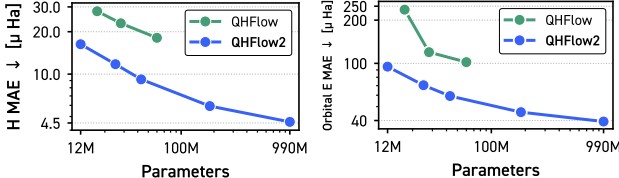

*Figure 6.* **Scaling behavior with model size.** (left) Hamiltonian MAE and (right) Occupied orbital MAE as a function of the number of parameters (log scale).

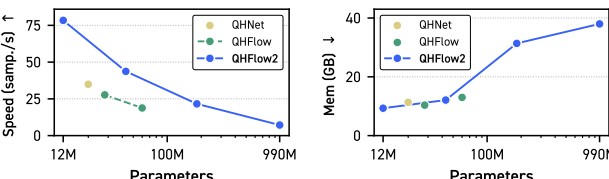

*Figure 7.* **Inference speed and memory scaling on QH9.** (left) inference speed (samples/s) versus parameter count. (right) peak inference GPU memory (GB) versus parameter count.

ergy and force accuracy under direct evaluation. Compared to representative MLIP baselines, QHFlow2 improves energy accuracy rapidly with additional data while maintaining competitive force accuracy at larger training sizes. These trends support the view that increased data can be effectively translated into improved electronic-structure fidelity and downstream physical accuracy on MD trajectories.

**Model parameter scaling on the QH9 benchmark.** We next vary the parameter count of QHFlow2 using size variants (-s/-m/-l/-xl) while keeping the training and evaluation pipeline fixed. Figure 6 summarizes how Hamiltonian MAE and occupied orbital energy MAE change with model capacity on QH9 and includes a direct comparison to QHFlow under the same protocol. Increasing capacity consistently improves both metrics, with QHFlow2 achieving lower errors than QHFlow at comparable parameter budgets. We could not extend the comparison to QHNet, as its train-

ing diverged with NaN losses beyond approximately 40M parameters even with learning-rate warmup and gradient clipping. These results show that increased model capacity yields consistent gains in both Hamiltonian and orbital accuracy, while QHFlow2 remains more parameter-efficient than QHFlow across the explored regime.

### 4.5. Runtime Analysis

We measure inference speed and peak GPU memory on QH9 under a shared evaluation setup. Inference times refer to the forward pass that outputs the predicted Hamiltonian $\hat{\mathbf{H}}$ on an NVIDIA RTX A6000 (48GB) with a batch size of 32, excluding data loading and I/O. Figure 7 summarizes how throughput and memory scale with the parameter count. For flow-based models, we report one-step inference. Across model sizes, QHFlow2 achieves a consistently better speed-memory trade-off than prior Hamiltonian models, delivering higher throughput at comparable memory.

## 5. Conclusion and Limitations

We evaluated machine-learning Hamiltonians (MLHs) via direct downstream energy and force calculations from predicted Hamiltonians, using a unified pipeline with recomputed rMD17 Hamiltonians for controlled comparisons to MLIP baselines. Within this framework, we proposed QH-Flow2, a scalable MLH model that combines an SO(2) backbone with a two-stage pair update to improve Hamiltonian accuracy and robustness. Across datasets, improved Hamiltonian accuracy consistently translated into lower downstream energy and force errors, with favorable scaling trends in model capacity and training data.

Our study has several limitations. First, QHFlow2 is trained under a fixed functional and basis set; transfer across DFT settings could be addressed by explicit conditioning, multimodal training, or lightweight adaptation. We also do not yet evaluate on OMol25-scale molecular datasets, where constructing training-ready Hamiltonian labels from raw quantum-chemistry outputs remains computationally and infrastructurally expensive. Second, we focus on restricted closed-shell systems, leaving spin-resolved Hamiltonian prediction for open-shell systems as future work. Third, one-shot analytic forces are evaluated from non-self-consistent densities induced by predicted Hamiltonians and are therefore not guaranteed to be strictly conservative; conservative forces can be obtained by using the prediction as an SCF initialization. Finally, end-to-end cost can still be dominated by diagonalization and DFT evaluation for large basis sizes, motivating faster Hamiltonian-to-energy/force pipelines.

## Acknowledgements

This work was supported by Institute for Information & communications Technology Planning & Evaluation (IITP) grant funded by the Korean government (MSIT) (RS-2019-II190075, Artificial Intelligence Graduate School Program (KAIST), and RS-2026-25507543), National Supercomputing Center with supercomputing resources, including technical support (KSC-2025-CRE-0602), the AI Computing Infrastructure Enhancement (GPU Rental Support) User Support Program funded by the Ministry of Science and ICT (MSIT), Republic of Korea, the National Research Foundation of Korea (NRF) grant funded by the Korea government (MSIT) (RS-2026-25473475), and a Korea University Grant (K2612031). This work was supported by the GRDC(Global Research Development Center) Cooperative Hub Program through the National Research Foundation of Korea(NRF) grant funded by the Ministry of Science and ICT(MSIT) (No. RS-2024-00436165). This work was supported by the National Research Foundation of Korea(NRF) grant funded by the Ministry of Science and ICT(MSIT) (No. RS-2022-NR072184). This work was supported by the National Research Foundation of Korea(NRF) grant funded by the Korea government(MSIT) (RS-2025-02216257). The authors thank Hyunjin Seo for help with the figures.

## Impact Statement

This work advances machine learning methods for quantum chemistry and molecular modeling. It does not involve human participants, personal data, or other sensitive information, and we therefore do not anticipate direct ethical risks arising from the methodology or experiments. Nevertheless, improved computational tools for molecular simulation may enable downstream applications in areas such as drug discovery and materials design, which can carry broader societal implications. We encourage responsible use of our methods in line with applicable laws, safety considerations, and established scientific and professional guidelines.

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

# A. Density Functional Theory

## A.1. Kohn–Sham density functional theory

Density functional theory (DFT) is a central tool in quantum chemistry and materials science for approximating the electronic structure of many-body systems. Since solving the many-electron Schrödinger equation directly is computationally prohibitive for large numbers of particles (Schrödinger, 1926), DFT reformulates the ground-state problem in terms of the electron density $\rho(\mathbf{r})$. Because $\rho(\mathbf{r})$ depends only on three spatial coordinates, this representation greatly reduces the effective complexity of the problem compared to working with an $N$-electron wavefunction. Here, we generally follow the contents of the DFT overview books.

In practice, DFT is most commonly used through the Kohn–Sham (KS) formulation (Hohenberg & Kohn, 1964; Kohn & Sham, 1965), which replaces the interacting electron system with an auxiliary system of non-interacting electrons constructed to reproduce the same ground-state density as the true system. The wavefunction of this reference system, the Kohn–Sham wavefunction $\Psi_{\mathrm{KS}}$, must satisfy fermionic antisymmetry (Pauli exclusion), and is therefore written as a Slater determinant of $N$ single-particle KS orbitals $\{\psi_i\}_{i=1}^N$:

$$\Psi_{\mathrm{KS}}(\mathbf{r}_1, \ldots, \mathbf{r}_N) = \frac{1}{\sqrt{N!}} \det[\psi_i(\mathbf{r}_j)]_{i,j=1}^N \tag{16}$$

In general, the electron density associated with an $N$-electron wavefunction $\Psi$ is defined by marginalizing out $N-1$ coordinates:

$$\rho(\mathbf{r}) = N \int \cdots \int |\Psi(\mathbf{r}, \mathbf{r}_2, \ldots, \mathbf{r}_N)|^2 \, d\mathbf{r}_2 \cdots d\mathbf{r}_N. \tag{17}$$

For a Slater determinant constructed from orthonormal orbitals, this reduces to a simple sum over the occupied KS orbitals:

$$\rho(\mathbf{r}) = \sum_{i=1}^N |\psi_i(\mathbf{r})|^2. \tag{18}$$

Rather than explicitly manipulating the full KS wavefunction, one can compute $\rho(\mathbf{r})$ directly from the single-particle orbitals. The orbitals $\{\psi_i(\mathbf{r})\}$ and their orbital energies $\{\epsilon_i\}$ are obtained by solving the Kohn–Sham equations,

$$\mathcal{H}[\rho]\psi_i(\mathbf{r}) = \left[ -\frac{1}{2}\nabla^2 + V_{\mathrm{ext}}(\mathbf{r}) + V_{\mathrm{H}}[\rho](\mathbf{r}) + V_{\mathrm{xc}}[\rho](\mathbf{r}) \right] \psi_i(\mathbf{r}) = \epsilon_i \psi_i(\mathbf{r}), \tag{19}$$

where the bracketed operator defines the effective KS Hamiltonian $\mathcal{H}[\rho]$. It consists of the kinetic-energy operator, the external potential $V_{\mathrm{ext}}$ from the nuclei, the Hartree potential $V_{\mathrm{H}}[\rho]$ describing classical electrostatic repulsion due to the electron density, and the exchange–correlation potential $V_{\mathrm{xc}}[\rho]$, which captures the remaining many-body quantum effects.

The ground-state density is then assembled from the $N$ occupied KS orbitals as

$$\rho(\mathbf{r}) = \sum_{i=1}^N |\psi_i(\mathbf{r})|^2. \tag{20}$$

It is worth emphasizing that the KS orbitals $\psi_i$ are auxiliary quantities: they are introduced to reproduce the correct ground-state density, rather than to represent the physical many-electron wavefunction. Moreover, because $\mathcal{H}[\rho]$ depends on $\rho$ and $\rho$ is constructed from the solutions $\{\psi_i\}$, the KS equations define a self-consistent problem.

Density functional theory (DFT) thus provides a tractable route to electronic structure by expressing ground-state properties in terms of $\rho(\mathbf{r})$ (Hohenberg & Kohn, 1964; Kohn & Sham, 1965). Within KS-DFT, the interacting system is mapped to non-interacting orbitals $\{\psi_i\}_{i=1}^N$ with $\psi_i : \mathbb{R}^3 \to \mathbb{C}$, which reproduce the ground-state density $\rho(\mathbf{r}) : \mathbb{R}^3 \to \mathbb{R}_{>0}$. Since the ground-state energy can be expressed as a functional of $\rho(\mathbf{r})$, one can evaluate it without explicitly handling the full many-electron interaction in wavefunction form.

The KS orbitals and density are obtained by solving

$$\hat{H}_{\mathrm{KS}}[\rho]\psi_i(\mathbf{r}) = \epsilon_i \psi_i(\mathbf{r}), \qquad \rho(\mathbf{r}) = \sum_{i=1}^N |\psi_i(\mathbf{r})|^2, \tag{21}$$

where $\epsilon_i$ denotes the orbital energies. The KS Hamiltonian operator $\hat{H}_{\mathrm{KS}}[\rho]$ depends explicitly on the electron density $\rho$, which is in turn constructed from the orbitals.

**Energy and force evaluation in KS-DFT.** Following standard DFT texts (Argaman & Makov, 2000; Neugebauer & Hickel, 2013; Jones, 2015), we summarize how total energies and atomic forces are evaluated in KS-DFT. Given Kohn–Sham (KS) orbitals $\{\psi_i\}$ with occupations $\{f_i\}$, the electron density is constructed as

$$\rho(\mathbf{r}) = \sum_i f_i \, |\psi_i(\mathbf{r})|^2, \tag{22}$$

where typically $f_i \in [0, 2]$ in spin-restricted settings (e.g., $f_i \in \{0, 2\}$ for closed-shell systems).

The KS total energy is written as the sum of kinetic, external, Hartree, exchange-correlation (XC), and nuclear-repulsion contributions:

$$E_{\mathrm{KS}}[\rho] = T_s[\rho] + \int v_{\mathrm{ext}}(\mathbf{r}) \, \rho(\mathbf{r}) \, d\mathbf{r} + E_{\mathrm{H}}[\rho] + E_{\mathrm{xc}}[\rho] + E_{\mathrm{nn}}, \tag{23}$$

where $T_s[\rho]$ is the non-interacting kinetic energy of the KS reference system, $v_{\mathrm{ext}}$ is the electron–nuclear potential, $E_{\mathrm{H}}[\rho]$ is the classical Coulomb (Hartree) energy, $E_{\mathrm{xc}}[\rho]$ accounts for exchange and correlation effects beyond $T_s$ and $E_{\mathrm{H}}$, and $E_{\mathrm{nn}}$ is the nuclear repulsion energy. The Hartree term is

$$E_{\mathrm{H}}[\rho] = \frac{1}{2} \iint \frac{\rho(\mathbf{r})\rho(\mathbf{r}')}{|\mathbf{r} - \mathbf{r}'|} \, d\mathbf{r} \, d\mathbf{r}', \qquad v_{\mathrm{H}}[\rho](\mathbf{r}) = \int \frac{\rho(\mathbf{r}')}{|\mathbf{r} - \mathbf{r}'|} \, d\mathbf{r}', \tag{24}$$

and the XC potential entering the KS equations is defined by the functional derivative

$$v_{\mathrm{xc}}[\rho](\mathbf{r}) = \frac{\delta E_{\mathrm{xc}}[\rho]}{\delta \rho(\mathbf{r})}. \tag{25}$$

In practice, $E_{\mathrm{xc}}[\rho]$ is approximated by a chosen XC functional (e.g., LDA, GGA, meta-GGA, or hybrid).

Atomic forces are obtained as derivatives of the total energy with respect to nuclear coordinates $\{\mathbf{R}_A\}$:

$$\mathbf{F}_A = -\nabla_{\mathbf{R}_A} E_{\mathrm{KS}}[\rho]. \tag{26}$$

When atom-centered basis functions are used, analytic forces typically include the Hellmann–Feynman contribution as well as Pulay terms arising from the nuclear dependence of the basis. Additional contributions may appear when parts of the energy (notably XC terms in GGA/meta-GGA) are evaluated by numerical quadrature on atom-centered grids, since quadrature weights and partition functions can depend on $\{\mathbf{R}_A\}$.

**Energy evaluation from a predicted Hamiltonian.** We now describe how energies are evaluated from a predicted Kohn–Sham Hamiltonian $\hat{\mathbf{H}}$ in QHFlow2. Given $\hat{\mathbf{H}}$ and the overlap matrix $\mathbf{S}$, we solve the generalized eigenproblem $\hat{\mathbf{H}}\,\mathbf{C} = \mathbf{S}\,\mathbf{C}\,\boldsymbol{\epsilon}$ to obtain the orbital coefficients $\mathbf{C}$, and assemble the density matrix $\mathbf{D} = \mathbf{C}\,\mathrm{diag}(\mathbf{o})\,\mathbf{C}^{\top}$ from the RKS occupation vector $\mathbf{o}$. We emphasize that $\hat{\mathbf{H}}$ enters the evaluation pipeline only through this eigenproblem; the total energy itself is not read off from $\hat{\mathbf{H}}$. Instead, we evaluate the KS energy functional (Equation (23)) in matrix form as

$$E_{\mathrm{KS}} = \mathrm{Tr}[\mathbf{D}\,\mathbf{H}_{\mathrm{core}}] + \tfrac{1}{2}\mathrm{Tr}\Big[\mathbf{D}\,\big(\mathbf{J}[\mathbf{D}] + \mathbf{V}_{\mathrm{xc}}[\mathbf{D}]\big)\Big] + E_{\mathrm{nn}}, \tag{27}$$

where the one-electron core Hamiltonian $\mathbf{H}_{\mathrm{core}}$ collects the kinetic and electron–nuclear contributions ($\mathbf{T}_s$ and $\mathbf{V}_{\mathrm{ext}}$ in Equation (23)); the overlap matrix $\mathbf{S}$ and the nuclear–nuclear repulsion $E_{\mathrm{nn}}$ are computed analytically from the molecular geometry. The Coulomb (Hartree) matrix $\mathbf{J}[\mathbf{D}]$ is the matrix-form counterpart of $v_{\mathrm{H}}[\rho]$ in Equation (24), and $\mathbf{V}_{\mathrm{xc}}[\mathbf{D}]$ is the matrix-form counterpart of $v_{\mathrm{xc}}[\rho]$ in Equation (25), obtained by numerical integration of the chosen XC functional on an atom-centered grid. We use PySCF (Sun et al., 2018) (with GPU acceleration via GPU4PySCF (Wu et al., 2024)) for this evaluation. Forces follow as analytic gradients $\mathbf{F}_i = -\nabla_{\mathbf{r}_i} E_{\mathrm{KS}}[\rho]$ (Equation (26)).

**Cost of energy evaluation.** The dominant cost of evaluating $E_{\mathrm{KS}}$ from a predicted $\hat{\mathbf{H}}$ lies in the reconstruction of $\mathbf{J}[\mathbf{D}]$, which involves two-electron Coulomb integrals scaling as $\mathcal{O}(N^4)$ in the number of basis functions $N$ under direct four-center evaluation. In our implementation, we reduce this cost by using density fitting (also known as resolution-of-identity) via GPU4PySCF (Wu et al., 2024), which approximates the four-center integrals with three-center quantities and an auxiliary basis, reducing the asymptotic scaling of the Coulomb build from $\mathcal{O}(N^4)$ to $\mathcal{O}(N^3)$. The exchange–correlation term $\mathbf{V}_{\mathrm{xc}}[\mathbf{D}]$ is comparatively cheaper but requires grid-based numerical integration. The one-electron quantities $\mathbf{H}_{\mathrm{core}}$, $\mathbf{S}$, and $E_{\mathrm{nn}}$ are evaluated analytically and contribute negligibly. Importantly, this corresponds to a *single Fock build* per geometry rather than a full SCF iteration: the predicted $\hat{\mathbf{H}}$ replaces the SCF convergence loop, so the cost of one QHFlow2 energy evaluation corresponds to one iteration of the underlying DFT solver rather than the full converged calculation. This is the main computational bottleneck of QHFlow2's evaluation pipeline relative to MLIPs that predict $E$ and $F$ as direct scalar/vector outputs.

### A.2. Variational energy minimization and self-consistency

Here, we describe KS self-consistency as a variational minimization problem. The KS energy functional $E_{\mathrm{KS}}[\rho]$ and its decomposition are given in Equation (23) (with the associated XC definitions in Equation (25)). The electron density is constructed from KS orbitals as in Equation (22), and the KS equations and Hamiltonian decomposition are summarized in Equation (23).

**Stationarity and the nonlinear eigenproblem.** At the ground state, $E_{\mathrm{KS}}[\rho]$ is stationary with respect to variations of the orbitals under the orthonormality and particle-number constraints. This stationarity yields the KS equations (Equation (19)), where the Hamiltonian depends on the density through the Hartree and XC potentials (Equations (24) and (25)). Consequently, solving KS-DFT amounts to a *nonlinear* eigenvalue problem: the density determines the Hamiltonian, and the Hamiltonian eigenvectors determine the density.

**Self-consistency as a fixed-point iteration.** Self-consistent field (SCF) methods solve the nonlinear KS problem by iterating a density-to-density (or density-matrix-to-density-matrix) map until consistency is reached. Let $\rho^{(k)}$ denote the density at iteration $k$, and define a map $\mathcal{F}$ that returns the output density obtained by (i) building $\hat{H}_{\mathrm{KS}}[\rho^{(k)}]$ (Equation (23)), (ii) solving the KS eigenproblem (Equation (19)), and (iii) forming the density from the resulting orbitals (Equation (22)). This yields the fixed-point form

$$\rho^{(k+1)} = \mathcal{F}(\rho^{(k)}). \tag{28}$$

In practice, direct iteration can be unstable, so one applies *mixing* to damp oscillations and accelerate convergence. A common choice is linear mixing,

$$\rho^{(k+1)} = (1-\alpha)\rho^{(k)} + \alpha\,\tilde{\rho}^{(k+1)}, \qquad \alpha \in (0,1], \tag{29}$$

where $\tilde{\rho}^{(k+1)} = \mathcal{F}(\rho^{(k)})$ is the "output" density computed from the KS solve. More advanced schemes (e.g., Pulay/DIIS) construct $\rho^{(k+1)}$ from a linear combination of several past iterates/residuals to improve robustness.

A typical stopping criterion checks either the density change $\|\rho^{(k+1)} - \rho^{(k)}\|$ or the energy change $|E_{\mathrm{KS}}[\rho^{(k+1)}] - E_{\mathrm{KS}}[\rho^{(k)}]|$ against a tolerance. At convergence, $\rho^\star$ satisfies the self-consistency condition $\rho^\star = \mathcal{F}(\rho^\star)$ and corresponds to a stationary point of the KS functional under the KS constraints.

**Matrix form with finite-basis.** Most implementations expand orbitals in a finite atom-centered basis $\{\phi_\mu\}_{\mu=1}^B$:

$$\psi_i(\mathbf{r}) = \sum_{\mu=1}^B C_{\mu i}\,\phi_\mu(\mathbf{r}), \tag{30}$$

which turns the KS equations (Equation (19)) into a generalized eigenvalue problem,

$$\mathbf{H}[\rho]\,\mathbf{C} = \mathbf{S}\,\mathbf{C}\,\boldsymbol{\epsilon}, \tag{31}$$

where $\mathbf{S}_{\mu\nu} = \langle \phi_\mu, \phi_\nu \rangle$ is the overlap matrix, $\mathbf{H}[\rho]$ is the KS (Fock) matrix assembled from the current density (via the Hartree and XC contributions; cf. Equations (24) and (25)), $\mathbf{C}$ is the coefficient matrix, and $\boldsymbol{\epsilon}$ is the diagonal matrix of orbital energies. The density matrix is formed as

$$\mathbf{P} = \mathbf{C}\,\mathrm{diag}\,\mathbf{o}\,\mathbf{C}^\top, \tag{32}$$

with an occupation vector **o**, it induces the real-space density (equivalent to Equation (22)). In this representation, SCF can be viewed as iterating $\mathbf{P}^{(k)} \mapsto \mathbf{H}[\mathbf{P}^{(k)}] \mapsto \mathbf{P}^{(k+1)}$ (with mixing) until convergence.

**Orbital occupations.** Let $\varepsilon_1 \leq \cdots \leq \varepsilon_B$ be the generalized eigenvalues. For restricted (spin-unpolarized) calculations, the number of occupied spatial orbitals is

$$n_{\mathrm{occ}} = \left\lceil \frac{N_e}{2} \right\rceil, \tag{33}$$

and the occupied subspace corresponds to the $n_{\mathrm{occ}}$ lowest eigenpairs. The electron density (or density matrix) is then constructed from the occupied orbitals, while higher-energy orbitals form the virtual subspace.

## B. Group theory and equivariance

### B.1. Group theory

We review basic group-theoretic notions needed to motivate equivariant architectures for Hamiltonian prediction, following standard references (Kosmann-Schwarzbach et al., 2010; Cornwell, 1997; Wigner, 2012).

**Groups.** A *group* is a pair $(G, \circ)$ where $G$ is a non-empty set and $\circ : G \times G \to G$ is a binary operation satisfying: (i) associativity $(a \circ b) \circ c = a \circ (b \circ c)$, (ii) identity $\exists e \in G$ s.t. $e \circ a = a \circ e = a$, (iii) inverse $\forall a \in G$, $\exists a^{-1} \in G$ s.t. $a \circ a^{-1} = a^{-1} \circ a = e$, and (iv) closure $a \circ b \in G$. We write $ab$ for $a \circ b$ when unambiguous. Many symmetry groups in physics are *Lie groups* (continuous groups) such as $\mathrm{SO}(3)$ and $\mathrm{SE}(3)$ (Cornwell, 1997).

**Group actions.** To formalize symmetries of objects, we specify how a group acts on a set. A (left) *group action* of $G$ on a set $\mathcal{X}$ is a map

$$\cdot_{\mathcal{X}} : G \times \mathcal{X} \to \mathcal{X}, \qquad (g, x) \mapsto g \cdot_{\mathcal{X}} x, \tag{34}$$

satisfying $e \cdot_{\mathcal{X}} x = x$ and $(gh) \cdot_{\mathcal{X}} x = g \cdot_{\mathcal{X}} (h \cdot_{\mathcal{X}} x)$. When clear, we omit the subscript and write $g \cdot x$. For example, $\mathrm{SO}(3)$ acts on $\mathbb{R}^3$ by rotations: $R \cdot \mathbf{r} = R\mathbf{r}$.

**Equivariance and invariance.** Let $G$ act on $\mathcal{X}$ and $\mathcal{Y}$ via $\cdot_{\mathcal{X}}$ and $\cdot_{\mathcal{Y}}$. A map $f : \mathcal{X} \to \mathcal{Y}$ is *G-equivariant* if

$$f(g \cdot_{\mathcal{X}} x) = g \cdot_{\mathcal{Y}} f(x), \qquad \forall g \in G, \ x \in \mathcal{X}. \tag{35}$$

If the output action is trivial ($g \cdot_{\mathcal{Y}} y = y$), this reduces to *invariance*: $f(g \cdot_{\mathcal{X}} x) = f(x)$. Equivariance is the appropriate symmetry notion for tensorial quantities (e.g., vectors, matrices) whose outputs transform under the group.

**Representations.** When $\mathcal{X}$ is a vector space, group actions are often linear. A *(linear) representation* of $G$ on a vector space $V$ is a homomorphism

$$\rho : G \to \mathrm{GL}(V), \qquad \rho(gh) = \rho(g)\rho(h), \ \rho(e) = I. \tag{36}$$

In finite dimensions, choosing a basis identifies $\rho(g)$ with an invertible matrix $D(g) \in \mathbb{C}^{d \times d}$. Two representations $D$ and $D'$ are *equivalent* if there exists an invertible $Q$ such that $D'(g) = Q^{-1}D(g)Q$ for all $g$.

Equivariance Equation (35) is most commonly expressed using representations: if $x \in V_{\mathrm{in}}$ and $f(x) \in V_{\mathrm{out}}$ with actions $D_{\mathrm{in}}(g)$ and $D_{\mathrm{out}}(g)$, then

$$f\big(D_{\mathrm{in}}(g)x\big) = D_{\mathrm{out}}(g)f(x). \tag{37}$$

### B.2. Irreducible representations and $\mathrm{SO}(3)$

**Irreducible representations.** A subspace $W \subseteq V$ is *invariant* under $\rho$ if $\rho(g)W \subseteq W$ for all $g \in G$. A representation $\rho$ is *irreducible* if its only invariant subspaces are $\{0\}$ and $V$. For compact groups (including $\mathrm{SO}(3)$), any finite-dimensional unitary representation decomposes into a direct sum of irreducible representations (irreps) (Wigner, 2012).

**Irreps of** $SO(3)$**.**   The irreps of $SO(3)$ are indexed by $\ell \in \mathbb{N}_0$ and have dimension $2\ell + 1$. They can be realized via Wigner $D$-matrices $D^{(\ell)} : SO(3) \to \mathbb{C}^{(2\ell+1)\times(2\ell+1)}$. A convenient basis is provided by spherical harmonics $Y_m^\ell$ on $\mathbb{S}^2$ (with $m = -\ell, \dots, \ell$), which transform under rotations according to $D^{(\ell)}$ (Wigner, 2012; Cornwell, 1997).

Any square-integrable function on the sphere can be expanded as

$$f(\theta, \phi) = \sum_{\ell=0}^{\infty} \sum_{m=-\ell}^{\ell} f_{\ell m} \, Y_m^\ell(\theta, \phi), \tag{38}$$

and under a rotation $R \in SO(3)$, the coefficient vector $\mathbf{f}^{(\ell)} = (f_{\ell,-\ell}, \dots, f_{\ell,\ell})^\top \in \mathbb{C}^{2\ell+1}$ transforms as

$$\mathbf{f}^{(\ell)} \xmapsto{R} D^{(\ell)}(R)\, \mathbf{f}^{(\ell)}. \tag{39}$$

This viewpoint motivates representing geometric features as collections of $\ell$-typed components (scalars $\ell = 0$, vectors $\ell = 1$, etc.), each of which transforms by the corresponding irrep matrix.

**Clebsch–Gordan (CG) tensor product.**   Given irreps $D^{(\ell_1)}$ and $D^{(\ell_2)}$, their tensor product $D^{(\ell_1)} \otimes D^{(\ell_2)}$ is generally reducible and decomposes as

$$D^{(\ell_1)} \otimes D^{(\ell_2)} \cong \bigoplus_{\ell=|\ell_1-\ell_2|}^{\ell_1+\ell_2} D^{(\ell)}. \tag{40}$$

The CG coefficients implement the change of basis from the product space to the direct-sum irrep basis. Concretely, for $\mathbf{u}^{(\ell_1)} \in \mathbb{C}^{2\ell_1+1}$ and $\mathbf{v}^{(\ell_2)} \in \mathbb{C}^{2\ell_2+1}$, the $\ell$-component $\mathbf{w}^{(\ell)} \in \mathbb{C}^{2\ell+1}$ is

$$w_m^{(\ell)} = \sum_{m_1=-\ell_1}^{\ell_1} \sum_{m_2=-\ell_2}^{\ell_2} C_{(\ell_1,m_1),(\ell_2,m_2)}^{(\ell,m)} u_{m_1}^{(\ell_1)} v_{m_2}^{(\ell_2)}. \tag{41}$$

This construction is equivariant:

$$\left( D^{(\ell_1)}(R)\mathbf{u}^{(\ell_1)} \right) \otimes_{\mathrm{CG}} \left( D^{(\ell_2)}(R)\mathbf{v}^{(\ell_2)} \right) = D^{(\ell)}(R)\mathbf{w}^{(\ell)}, \qquad \forall R \in SO(3). \tag{42}$$

**Equivariant neural layers (TFN-style).**   Many $SE(3)$-equivariant GNNs represent node/edge features as a direct sum of $SO(3)$ irreps and update them using CG tensor products with spherical-harmonic filters (Thomas et al., 2018; Geiger & Smidt, 2022). Let $\mathbf{V}_i^{(\ell)} \in \mathbb{C}^{(2\ell+1)\times C_\ell}$ denote the $\ell$-typed feature at node $i$ with $C_\ell$ channels. Given relative direction $\hat{\mathbf{r}}_{ij} = \mathbf{r}_{ij}/\|\mathbf{r}_{ij}\|$ and distance $r_{ij} = \|\mathbf{r}_{ij}\|$, a typical TFN filter uses a radial network $R(r_{ij})$ and spherical harmonics $Y^{(\ell_f)}(\hat{\mathbf{r}}_{ij})$:

$$\mathbf{F}^{(\ell_f)}(\mathbf{r}_{ij}) = R^{(\ell_f)}(r_{ij})\, Y^{(\ell_f)}(\hat{\mathbf{r}}_{ij}), \tag{43}$$

and messages are formed via CG products so that the output transforms as a valid irrep (cf. Equations (40) and (42)).

### B.3. Symmetry and equivariance of RH-DFT Hamiltonians

**Atomic orbital basis and spherical harmonics.**   To solve the KS equations in practice, molecular orbitals are expanded in an atom-centered basis. A prototypical atomic orbital can be written as (Koch & Holthausen, 2015)

$$\phi_{n\ell m}(\mathbf{r}) = R_{n\ell}(\|\mathbf{r}\|)\, Y_m^\ell(\theta, \phi), \tag{44}$$

where $R_{n\ell}$ is a radial function (e.g., Gaussian or Slater-type) (Gill, 1994; Slater, 1930; Hehre et al., 1969), and $Y_m^\ell$ is a spherical harmonic. A molecular orbital is then expressed as a linear combination of shifted atomic orbitals,

$$\psi_i(\mathbf{r}) = \sum_{\alpha=1}^{B} C_{\alpha i}\, \phi_\alpha(\mathbf{r} - \mathbf{R}_\alpha), \tag{45}$$

with coefficient matrix $\mathbf{C} \in \mathbb{R}^{B \times O}$.

**Hamiltonian blocks as intertwiners.** In a spherical (angular-momentum) basis, basis functions are grouped by $\ell$-type. Let $\mathbf{H}^{(\ell,\ell')}$ denote the submatrix coupling the $\ell$-block on one center to the $\ell'$-block on another center, with $\mathbf{H}^{(\ell,\ell')} \in \mathbb{C}^{(2\ell+1)\times(2\ell'+1)}$. Under a global rotation $R \in \mathrm{SO}(3)$, the spherical basis transforms by block-diagonal Wigner matrices $\mathcal{D}(R) = \mathrm{blkdiag}\big(D^{(\ell)}(R)\big)_{\ell}$. Equivariance of the KS operator in this basis implies that each block obeys the conjugation rule

$$\mathbf{H}^{(\ell,\ell')} \overset{R}{\mapsto} D^{(\ell)}(R)\,\mathbf{H}^{(\ell,\ell')}\,D^{(\ell')}(R)^{-1}. \tag{46}$$

Equivalently, $\mathbf{H}^{(\ell,\ell')}$ is an *intertwiner* between the $\ell'$ and $\ell$ representation spaces. This blockwise transformation law motivates parameterizing Hamiltonian predictors so that their outputs are collections of irrep-typed blocks that satisfy Equation (46) by construction.

# C. Training, Priors, and Architecture

This section summarizes the learning objectives, rotationally invariant priors, and the network architecture used in QHFlow2 to predict Kohn–Sham Hamiltonians in an atom-centered orbital basis. Throughout, $\mathcal{M} = \{(Z_i, \mathbf{r}_i)\}_{i=1}^{N}$ denotes a molecular configuration, $\mathbf{H} \in \mathbb{R}^{B\times B}$ the (real-valued) Hamiltonian matrix in the chosen basis, and $\mathbf{S} \in \mathbb{R}^{B\times B}$ the overlap matrix.

## C.1. Traininig objective as Residual parameterization

Rather than regressing the converged Hamiltonian directly, QHFlow2 learns the *residual* between a cheap initialization and the SCF solution, following residual-learning strategies used in prior Hamiltonian predictors (e.g., WANet and SHNet). For each molecule $\mathcal{M}$, let $\mathbf{H}_{\mathcal{M}}^{(0)}$ be an initial Hamiltonian guess and $\mathbf{H}_{\mathcal{M}}^{\star}$ be the converged SCF solution. We define the residual target

$$\mathbf{H}_{1,\mathcal{M}} := \mathbf{H}_{\mathcal{M}}^{\star} - \mathbf{H}_{\mathcal{M}}^{(0)}. \tag{47}$$

In our implementation, $\mathbf{H}_{\mathcal{M}}^{(0)}$ is computed using the `minao` initialization in PySCF (Sun et al., 2018), which does not involve any SCF iterations. At inference time, we reconstruct the Hamiltonian as

$$\widehat{\mathbf{H}}_{\mathcal{M}}^{\star} = \mathbf{H}_{\mathcal{M}}^{(0)} + \widehat{\mathbf{H}}_{1,\mathcal{M}}. \tag{48}$$

For completeness, we also evaluate a direct parameterization that does not rely on an explicit initialization (i.e., treating $\mathbf{H}_{\mathcal{M}}^{(0)} = \mathbf{0}$ and predicting $\mathbf{H}_{\mathcal{M}}^{\star}$ directly), and report the corresponding results in the Table 12.

## C.2. Rotationally invariant priors for $\mathbf{H}_0$

We introduce two priors $p_0$ for sampling the initial residual $\mathbf{H}_0$: a Gaussian orthogonal ensemble (GOE) prior and a tensor expansion-based (TE) prior (Kim et al., 2025). In all flow-based Hamiltonian prediction experiments in this work, we use the TE prior as the default choice for $p_0$.

**GOE prior.** For the GOE prior, we sample a symmetric matrix by drawing independent entries $M_{ij} \sim \mathcal{N}(0, \sigma^2)$ for $i \leq j$ and setting $M_{ji} = M_{ij}$. We use $\sigma^2 = 1.0$ for MD17 and $\sigma^2 = 0.1$ for QH9.

**Tensor expansion (TE) prior.** The TE prior constructs an $\mathrm{SO}(3)$-invariant distribution by sampling irrep-valued latent vectors and mapping them to matrix blocks via tensor expansion. Let $\mathbf{w}^{(\ell)} \in \mathbb{R}^{2\ell+1}$ be a type-$\ell$ irrep vector. We sample

$$\mathbf{w}^{(\ell)} = r\,D^{(\ell)}(\mathbf{R})\,\mathbf{w}_0^{(\ell)}, \tag{49}$$

where $\mathbf{w}_0^{(\ell)}$ is a fixed unit-norm reference vector, $\mathbf{R} \sim \mathrm{Uniform}(\mathrm{SO}(3))$, and $r$ is sampled independently from a radial distribution. This factorization makes $p(\mathbf{w}^{(\ell)})$ $\mathrm{SO}(3)$-invariant. In practice, we use $r \sim \mathrm{LogNormal}(1, 0.1)$.

Given $\mathbf{w}^{(\ell)}$, tensor expansion maps it into a matrix block of type $(\ell_1, \ell_2)$ via Clebsch–Gordan coefficients:

$$\left(\bar{\otimes}\,\mathbf{w}^{(\ell)}\right)_{m_1 m_2}^{(\ell_1,\ell_2)} = \sum_{m=-\ell}^{\ell} C_{(\ell_1,m_1),(\ell_2,m_2)}^{(\ell,m)}\,w_m^{(\ell)}, \tag{50}$$

where $m_1 \in \{-\ell_1, \ldots, \ell_1\}$ and $m_2 \in \{-\ell_2, \ldots, \ell_2\}$. The resulting block transforms equivariantly as

$$\left(\bar{\otimes}\, \mathbf{w}^{(\ell)}\right)^{(\ell_1, \ell_2)} \xrightarrow{\mathbf{R}} D^{(\ell_1)}(\mathbf{R}) \left(\bar{\otimes}\, \mathbf{w}^{(\ell)}\right)^{(\ell_1, \ell_2)} D^{(\ell_2)}(\mathbf{R})^{-1}. \tag{51}$$

We use this TE prior as a rotationally invariant noise distribution for sampling $\mathbf{H}_0$ in the flow matching path Equation (4).

### C.3. Training and sampling algorithms

For completeness, we summarize the training and sampling procedures in Algorithms 1 and 2. **Unless stated otherwise, we set $p_0$ to the TE prior throughout this work.** The only difference between the GOE and TE settings is the choice of $p_0$ for sampling $\mathbf{H}_0$.

### C.4. Training and Sampling Algorithms of QHFlow2

For completeness, we summarize the training and sampling procedures in Algorithms 1 and 2. The only difference between the GOE and TE settings is the choice of $p_0$ for sampling $\mathbf{H}_0$.

---

**Algorithm 1** QHFlow2 training procedure (residual CFM)

---

**Require:** Dataset $\{(\mathcal{M}_i, \mathbf{S}_{\mathcal{M}_i}, \mathbf{H}^\star_{\mathcal{M}_i}, \mathbf{H}^{(0)}_{\mathcal{M}_i})\}$, model $f_\theta$, prior $p_0$
1: **for** each training step **do**
2:     Sample minibatch $\mathcal{B} = \{\mathcal{M}_i\}_{i=1}^{|\mathcal{B}|}$
3:     **for** each $\mathcal{M}$ in $\mathcal{B}$ **do**
4:         Compute residual target $\mathbf{H}_{1,\mathcal{M}} \leftarrow \mathbf{H}^\star_{\mathcal{M}} - \mathbf{H}^{(0)}_{\mathcal{M}}$
5:         Sample $\mathbf{H}_0 \sim p_0$ and $t \sim \mathcal{U}(0, 1)$
6:         Interpolate $\mathbf{H}_{t,\mathcal{M}} \leftarrow (1 - t)\mathbf{H}_0 + t\,\mathbf{H}_{1,\mathcal{M}}$
7:         Predict $\mathbf{H}^{(\theta)}_{1,\mathcal{M}} \leftarrow f_\theta(\mathbf{H}_{t,\mathcal{M}}, \mathbf{S}_{\mathcal{M}}, \mathcal{M}, t)$
8:         Accumulate loss $\mathcal{L}_{\mathcal{M}} \leftarrow \|\mathbf{H}^{(\theta)}_{1,\mathcal{M}} - \mathbf{H}_{1,\mathcal{M}}\|_F^2$
9:     **end for**
10:   Update $\theta$ by gradient descent on $\mathcal{L} = \frac{1}{|\mathcal{B}|} \sum_{\mathcal{M} \in \mathcal{B}} \mathcal{L}_{\mathcal{M}}$
11: **end for**

---

**Algorithm 2** QHFlow2 sampling procedure (ODE discretization)

---

**Require:** Molecular configuration $\mathcal{M}$, overlap matrix $\mathbf{S}_{\mathcal{M}}$, initialization $\mathbf{H}^{(0)}_{\mathcal{M}}$, model $f_\theta$, prior $p_0$, time grid $\{t_k\}_{k=0}^K$ with $t_0 = 0, t_K = 1$
1: Sample $\mathbf{H}_0 \sim p_0$ and set $\mathbf{H}_{t_0} \leftarrow \mathbf{H}_0$
2: **for** $k = 0$ to $K - 1$ **do**
3:     $\mathbf{H}^{(\theta)}_{1,\mathcal{M}} \leftarrow f_\theta(\mathbf{H}_{t_k}, \mathbf{S}_{\mathcal{M}}, \mathcal{M}, t_k)$
4:     $v_{t_k,\theta} \leftarrow (\mathbf{H}^{(\theta)}_{1,\mathcal{M}} - \mathbf{H}_{t_k})/(1 - t_k)$
5:     $\mathbf{H}_{t_{k+1}} \leftarrow \mathbf{H}_{t_k} + (t_{k+1} - t_k)\, v_{t_k,\theta}$
6: **end for**
7: Output residual $\widehat{\mathbf{H}}_{1,\mathcal{M}} \leftarrow \mathbf{H}_{t_K}$
8: Reconstruct Hamiltonian $\widehat{\mathbf{H}}^\star_{\mathcal{M}} \leftarrow \mathbf{H}^{(0)}_{\mathcal{M}} + \widehat{\mathbf{H}}_{1,\mathcal{M}}$

---

## D. Model architecture

### D.1. Two-stage pair refinement

QHFlow2 maintains explicit pairwise representations $\mathbf{z}_{ij}$ to improve off-diagonal Hamiltonian blocks. The pair module follows the design of the *non-diagonal pair update* in QHNet (adapted to our setting), but is implemented with SO(2)-equivariant operations in edge-aligned local frames. Concretely, we perform a two-stage update: (i) *initialization* from geometric/chemical edge cues, and (ii) *refinement* using the final node context. We summarize the abstraction as

$$\mathbf{z}_{ij} \leftarrow \text{InitPair}(\mathbf{e}_{ij}), \qquad \mathbf{z}_{ij} \leftarrow \text{UpdPair}(\mathbf{z}_{ij}, \mathbf{z}_i, \mathbf{z}_j, \mathbf{e}_{ij}), \tag{52}$$

where both stages are type-preserving (irrep-wise) and equivariant by construction.

**Notation and local frames.** Let $\mathbf{z}_i = \bigoplus_{\ell=0}^{\ell_{\max}} \mathbf{z}_i^{(\ell)}$ be the node irreps and $\mathbf{z}_{ij} = \bigoplus_{\ell=0}^{\ell_{\max}} \mathbf{z}_{ij}^{(\ell)}$ the pair irreps. Edge attributes $\mathbf{e}_{ij}$ include distance encodings and chemical types (and optionally a reference-frame cue). For each edge $(i,j)$, let $\mathbf{R}_{ij} \in \text{SO}(3)$ align $\hat{\mathbf{r}}_{ij}$ with the $z$-axis. We evaluate pair operations in the local frame $\tilde{\mathbf{z}}_i^{(\ell)} = D^{(\ell)}(\mathbf{R}_{ij}) \mathbf{z}_i^{(\ell)}$ (and similarly for $j$), where the residual symmetry reduces to an SO(2) action around the axis.

**Stage I: pair initialization.** The initialization stage injects purely edge-level information into $\mathbf{z}_{ij}$ (before any node-context refinement). We parameterize

$$\mathbf{z}_{ij}^{(\ell)} = \Phi_{\text{init}}^{(\ell)}(\mathbf{e}_{ij}), \qquad \ell = 0, \ldots, \ell_{\max}, \tag{53}$$

where $\Phi_{\text{init}}^{(\ell)}$ outputs an $\ell$-type feature (with channels) using SO(2)-equivariant edgewise operations in the local frame (followed by rotation back to the global frame). Intuitively, $\Phi_{\text{init}}$ provides a geometry-aware seed representation for each pair, which is later refined by node features.

**Stage II: non-diagonal pair refinement via tensor-product filtering.** The refinement stage updates $\mathbf{z}_{ij}$ using the final node features, mirroring the non-diagonal pair update of QHNet. The core idea is to form an equivariant *pair message* by (a) gating and self-interacting the node irreps, (b) computing an attention-like scalar weight per channel path, and (c) applying a Clebsch–Gordan tensor product to couple the two node features into a pair irrep.

*(a) Norm-gate and self-interaction.* For each $\ell_{\text{in}}$, we first apply a norm gate to stabilize magnitudes and then a type-preserving self-interaction (channel mixing):

$$\bar{\mathbf{z}}_i^{(\ell_{\text{in}})} = \text{GateNorm}^{(\ell_{\text{in}})}(\tilde{\mathbf{z}}_i^{(\ell_{\text{in}})}), \qquad \hat{\mathbf{z}}_i^{(\ell_{\text{in}})} = \mathbf{W}_{\text{SI}}^{(\ell_{\text{in}})} \bar{\mathbf{z}}_i^{(\ell_{\text{in}})}, \tag{54}$$

and similarly for $j$. Here $\mathbf{W}_{\text{SI}}^{(\ell_{\text{in}})}$ acts only on channels, hence preserves equivariance.

*(b) Edge-conditioned filter and attentive weight.* For each path $(\ell_{\text{in}}, \ell_{\text{in}} \to \ell_{\text{out}})$, we produce a scalar filter from the edge attributes:

$$R^{(\ell_{\text{in}}, \ell_{\text{out}})}(\mathbf{e}_{ij}) = \text{MLP}_R^{(\ell_{\text{in}}, \ell_{\text{out}})}(\mathbf{e}_{ij}), \qquad a_{ij}^{(\ell_{\text{in}}, \ell_{\text{out}})} = \text{MLP}_a^{(\ell_{\text{in}}, \ell_{\text{out}})}\left(\mathbf{z}_i^{(0)}, \mathbf{z}_j^{(0)}, \mathbf{e}_{ij}\right), \tag{55}$$

and combine them as in QHNet (Yu et al., 2023b):

$$F_{ij}^{(\ell_{\text{in}}, \ell_{\text{out}})} = a_{ij}^{(\ell_{\text{in}}, \ell_{\text{out}})} R^{(\ell_{\text{in}}, \ell_{\text{out}})}(\mathbf{e}_{ij}). \tag{56}$$

The weight $F_{ij}$ is a scalar over irrep indices, so it does not affect equivariance.

*(c) Tensor-product pair message.* We then form an equivariant pair message by coupling the two node irreps through a Clebsch–Gordan tensor product (computed in the local frame):

$$\tilde{\mathbf{f}}_{ij}^{(\ell_{\text{out}})} = \sum_{\ell_{\text{in}}} \left( F_{ij}^{(\ell_{\text{in}}, \ell_{\text{out}})} \hat{\mathbf{z}}_i^{(\ell_{\text{in}})} \otimes_{\text{CG}} \hat{\mathbf{z}}_j^{(\ell_{\text{in}})} \right)^{(\ell_{\text{out}})}. \tag{57}$$

Finally, we rotate the message back to the global frame and update the stored pair feature by combining the previous pair state $\mathbf{z}_{ij}$ and the new message $\mathbf{f}_{ij}$ with an e3nn linear tensor product:

$$\mathbf{f}_{ij}^{(\ell)} = D^{(\ell)}(\mathbf{R}_{ij})^{-1} \tilde{\mathbf{f}}_{ij}^{(\ell)}, \qquad \Delta \mathbf{z}_{ij} = \text{LTP}\left(\mathbf{z}_{ij}, \mathbf{f}_{ij}, \mathbf{e}_{ij}\right), \qquad \mathbf{z}_{ij} \leftarrow \mathbf{z}_{ij} + \Delta \mathbf{z}_{ij}, \tag{58}$$

where $\mathrm{LTP}(\cdot)$ denotes the e3nn *linear tensor product* (LinearTensorProduct), which maps two irrep-valued inputs to an irrep-valued output via Clebsch–Gordan coupling with learnable channel weights. The edge attributes $\mathbf{e}_{ij}$ are used to produce the tensor-product weights (e.g., through an MLP), while equivariance is preserved by construction.

## D.2. Tensor expansion for Hamiltonian construction.

To represent on-site and inter-site interactions in a common equivariant form, we define a unified feature for each ordered pair $(i, j)$:

$$\mathbf{w}_{ij} = \begin{cases} \mathbf{z}_i, & i = j, \\ \mathbf{z}_{ij}, & i \neq j, \end{cases} \qquad \mathbf{w}_{ij} = \bigoplus_{\ell=0}^{\ell_{\max}} \mathbf{w}_{ij}^{(\ell)}, \tag{59}$$

where $\mathbf{z}_i$ denotes the final node feature and $\mathbf{z}_{ij}$ the final pair feature. Each component $\mathbf{w}_{ij}^{(\ell)}$ is a type-$\ell$ irrep feature (channels suppressed) and transforms under a global rotation $\mathbf{R} \in \mathrm{SO}(3)$ as $\mathbf{w}_{ij}^{(\ell)} \xrightarrow{\mathbf{R}} D^{(\ell)}(\mathbf{R})\mathbf{w}_{ij}^{(\ell)}$. To map these irrep features into Hamiltonian blocks consistent with orbital angular-momentum coupling, we construct for each $(\ell_1, \ell_2)$ an equivariant matrix block $\mathbf{H}_{ij}^{(\ell_1,\ell_2)} \in \mathbb{R}^{(2\ell_1+1)\times(2\ell_2+1)}$ by combining tensor expansion with a learnable channel projection:

$$\mathbf{H}_{ij}^{(\ell_1,\ell_2)} = \sum_{\ell=|\ell_1-\ell_2|}^{\ell_1+\ell_2} \mathbf{W}_{ij}^{(\ell \to \ell_1,\ell_2)} \left(\bar{\otimes}\, \mathbf{w}_{ij}^{(\ell)}\right)^{(\ell_1,\ell_2)}, \tag{60}$$

where $\left(\bar{\otimes}\, \mathbf{w}_{ij}^{(\ell)}\right)^{(\ell_1,\ell_2)}$ is the tensor expansion defined in [Equation (50)](#) and $\mathbf{W}_{ij}^{(\ell \to \ell_1,\ell_2)}$ acts only on channel dimensions (and may depend on edge attributes), thereby preserving equivariance. By construction, each block satisfies

$$\mathbf{H}_{ij}^{(\ell_1,\ell_2)} \xrightarrow{\mathbf{R}} D^{(\ell_1)}(\mathbf{R})\, \mathbf{H}_{ij}^{(\ell_1,\ell_2)}\, D^{(\ell_2)}(\mathbf{R})^{-1}, \tag{61}$$

and assembling all blocks across $(i, j)$ yields a Hamiltonian that obeys the global similarity transform, guaranteeing exact $\mathrm{SO}(3)$ equivariance in the atom-centered orbital basis.

## D.3. Comparison with related MLHs

We expand on the relationship between QHFlow2 and two recent MLHs that share elements of our backbone or pair-update design: HELM ([Kaniselvan et al., 2025](#)) and QHNetV2 ([Yu et al., 2025](#)). Where fair quantitative comparison is feasible, we include the results in the main tables; where it is not, we report the available comparisons in this appendix. We note that, to our knowledge, the codebases for both HELM and QHNetV2 are not publicly available at the time of writing, which limits the scope of direct comparison.

**HELM.** HELM and QHFlow2 share an eSEN-style SO(2)-equivariant backbone for atom-wise message passing, but differ in two main respects.

*(i) Hamiltonian construction.* HELM maps the backbone's node and edge embeddings to Hamiltonian blocks via gated nonlinearities and a node-parity expansion applied on the irrep features, without an additional SO(3) tensor-product refinement of pair features. QHFlow2 instead introduces a two-stage pair update that first constructs SO(2)-equivariant pair features and then refines them with SO(3) tensor-product coupling ([Section D.1](#)), followed by SO(3) tensor expansion to orbital-pair matrix blocks. Our ablation ([Table 10](#)) shows that the SO(3) refinement is important for off-diagonal accuracy.

*(ii) Energy evaluation.* The two methods take different design choices for downstream energy evaluation. HELM trains a dedicated energy head via fine-tuning on top of the Hamiltonian-pretrained backbone, whereas QHFlow2 derives energies and analytic forces directly from the predicted $\mathbf{H}$ via the standard DFT energy functional. QHFlow2's approach requires one Fock build per evaluation but produces energies and forces directly consistent with the predicted electronic structure. We note that HELM also reports energies (but not forces) evaluated directly from the predicted Hamiltonian on MD17, which allows a like-for-like comparison with QHFlow2.

The two approaches can be compared on MD17 ([Table 12](#)), where HELM achieves competitive Hamiltonian MAE but yields substantially higher energy errors than QHFlow2.

**QHNetV2.** QHNetV2 and QHFlow2 share two ingredients: the SO(2) Linear operation from eSCN (Passaro & Zitnick, 2023) used in the backbone, and the tensor-expansion readout from QHNet for converting irrep features into orbital-pair blocks. The architectures differ in both the backbone and the off-diagonal refinement.

*(i) Backbone.* QHNetV2's backbone consists of three modules per layer (Yu et al., 2025, Figure 2): (a) a node-wise interaction module that rotates neighbor features into edge-local SO(2) frames, applies SO(2) Linear and gating with radial-basis-conditioned weights, and aggregates them back to the global frame; (b) a node feature update that applies a continuous SO(2) tensor product in a node-local SO(2) frame, inspired by MACE's symmetric contraction (Batatia et al., 2022); and (c) an off-diagonal SO(2) FFN that keeps pair features inside the edge-local SO(2) frame throughout the layers. QHFlow2 instead uses the eSEN backbone of Fu et al. (2025), which includes a smoothed cutoff envelope and edgewise interaction blocks that are not part of the QHNetV2 backbone.

*(ii) Off-diagonal pair update.* QHNetV2 refines off-diagonal blocks with an SO(2) feed-forward network in local frames, applied as a single SO(2) stage (Yu et al., 2025, Eq. 17). This is conceptually similar to the first stage of our pair update (Equation (53)), in which pair features are constructed in the local frame by concatenating node features and applying SO(2) operations. QHFlow2 then applies an additional SO(3) tensor-product refinement (Equation (57)), which we find further improves off-diagonal accuracy (Table 10).

QHNetV2 reports Hamiltonian and orbital metrics under the without-init setting, which we include in the main text for direct comparison (Table 2); QHFlow2-m achieves lower Hamiltonian MAE and lower occupied orbital MAE on QH9-stable-id. QHNetV2 does not report downstream energy or force, so further comparison on those quantities is not feasible.

# E. Experimental study settings

## E.1. Dataset preparation

To demonstrate the effectiveness of flow-matching-based training, we conduct experiments on two molecular datasets: MD17 and QH9. The MD17 represents a relatively simple task compared to the QH9, focusing solely on small systems and their conformational space. The PubChemQH9 is not considered since their dataset and codebase are not publicly released.

**MD17.** The MD17 (Chmiela et al., 2017; Schütt et al., 2019) dataset consists of quantum chemical simulations for four small organic molecules: ethanol ($C_2H_5OH$), malondialdehyde ($CH_2(CHO)_2$), and uracil ($C_4H_4N_2O_2$). It provides a comprehensive set of molecular properties, including geometries, total energies, forces, Kohn–Sham Hamiltonian matrices, and overlap matrices. All reference computations were implemented via the ORCA electronic structure package (Neese et al., 2020) using the PBE exchange–correlation functional (Perdew et al., 1996; Weigend & Ahlrichs, 2005) and the def2-SVP Gaussian-type orbital (GTO) basis set. We follow the standard data split protocol used in prior work (Schütt et al., 2019; Unke et al., 2021a; Yu et al., 2023b) to divide each molecule's conformational data into training, validation, and test sets. The detailed dataset statistics are summarized in Table 3 and MOs in the table imply molecular orbitals (*i.e.*, s, p, d, f)

*Table 3.* The statistics of MD17 dataset (Schütt et al., 2019).

| Dataset | # of structures | Train | Val | Test | # of atoms | # of orbitals | # of occupied MOs |
|---|---|---|---|---|---|---|---|
| Ethanol | 30,000 | 25,000 | 500 | 4,500 | 9 | 72 | 10 |
| Malondialdehyde | 26,978 | 25,000 | 500 | 1,478 | 9 | 90 | 19 |
| Uracil | 30,000 | 25,000 | 500 | 4,500 | 12 | 132 | 26 |

**rMD17.** The revised MD17 (rMD17) dataset (Christensen & Von Lilienfeld, 2020) was introduced to mitigate inconsistencies in the DFT settings of the original MD17 trajectories by recomputing energies and forces at the PBE/def2-SVP level of theory with very tight SCF convergence criteria and a dense numerical integration grid (using ORCA (Neese et al., 2020)). We build on rMD17 because its reference energies and forces are reproducible across standard DFT packages, which enables controlled comparisons between Hamiltonian-based models and machine-learning interatomic potentials.

In this work, we further consider a larger molecular system than in our MD17 Hamiltonian benchmark to study a more challenging regime. We recompute reference quantities with PySCF at the PBE/def2-SVP level, using density fitting (via GPU4PySCF (Wu et al., 2024)) and an SCF convergence tolerance of $10^{-7}$. Unless otherwise noted, we use a numerical integration grid of level 3.

**QH9.** QH9 (Yu et al., 2023a) is a large-scale quantum chemistry benchmark for training and evaluating models that predict Hamiltonian matrices across diverse chemical structures. Built on QM9 (Ruddigkeit et al., 2012; Ramakrishnan et al.,

*Table 4.* The statistics of rMD17 dataset (Schütt et al., 2019).

| Dataset | # of structures | Train | Val | Test | # of atoms | # of orbitals | # of occupied MOs |
|---|---|---|---|---|---|---|---|
| Salicylic acid | 30,000 | 25,000 | 500 | 4,500 | 16 | 170 | 36 |
| Naphthalene | 30,000 | 25,000 | 500 | 4,500 | 18 | 180 | 34 |
| Aspirin | 30,000 | 25,000 | 500 | 4,500 | 21 | 222 | 47 |

2015), it contains 130,831 Hamiltonians from equilibrium geometries and 2,698 molecular dynamics (MD) trajectories. The dataset covers small organic molecules with up to nine heavy atoms (C, N, O, and F). All Hamiltonians are computed with PySCF (Sun et al., 2018) using the B3LYP (Stephens et al., 1994) exchange–correlation functional and the def2-SVP Gaussian-type orbital basis. We report detailed dataset statistics in Table 5.

QH9 consists of two subsets, `QH9-stable` and `QH9-dynamic-300k`, and defines four standard evaluation splits: stable-id, stable-ood, dynamic-300k-geo, and dynamic-300k-mol. For stable-id, `QH9-stable` is randomly split into training, validation, and test sets. For stable-ood, the split is defined by molecular size: molecules with 3–20 atoms are used for training, 21–22 atoms for validation, and 23–29 atoms for testing. This setup evaluates out-of-distribution generalization to larger and more complex molecules.

The dynamic-300k splits are constructed from MD trajectories, where each of the 2,698 molecules is associated with 100 geometry snapshots. In dynamic-300k-`geo`, snapshots are split within each molecule, with an 80/10/10 train/validation/test ratio. As a result, all molecular identities appear in every split, while the conformations are disjoint, isolating geometric generalization. In contrast, dynamic-300k-`mol` splits molecular identities into disjoint train/validation/test sets with the same 80/10/10 ratio, and assigns all 100 snapshots of a molecule to the same split. This setting is more challenging than `geo`, as it requires generalization to unseen molecules rather than unseen conformations.

*Table 5.* The statistics of QH9 dataset (Yu et al., 2023a).

| Dataset | # of structures | # of Molecules | Train | Val | Test |
|---|---|---|---|---|---|
| Stable-id | 130,831 | 130,831 | 104,664 | 13,083 | 13,084 |
| Stable-ood | 130,831 | 130,831 | 104,001 | 17,495 | 9,335 |
| Dynamic-300k-geo | 269,800 | 2,698 | 215,840 | 26,980 | 26,980 |
| Dynamic-300k-mol | 269,800 | 2,698 | 215,800 | 26,900 | 27,100 |

### E.2. Evaluation Metrics

We evaluate Hamiltonian prediction models at three levels: (i) element-wise reconstruction of the Hamiltonian matrix, (ii) spectral fidelity of the generalized eigenvalue problem, and (iii) accuracy of frontier-orbital quantities. Throughout this section, we use the following notation.

**Notation.** For a molecular configuration $\mathcal{M} = \{(Z_i, \mathbf{r}_i)\}_{i=1}^{N}$, let $\mathbf{H}^\star \in \mathbb{R}^{B \times B}$ denote the reference (SCF-converged) Kohn–Sham Hamiltonian matrix in an atom-centered orbital basis with $B$ basis functions. The model outputs a prediction $\hat{\mathbf{H}} \in \mathbb{R}^{B \times B}$. We denote by $\mathbf{S} \in \mathbb{R}^{B \times B}$ the overlap matrix of the same basis, which is fixed given $\mathcal{M}$ and the basis choice.

Given $(\mathbf{H}, \mathbf{S})$, orbital energies and coefficients are obtained by solving the generalized eigenvalue problem

$$\mathbf{H}\,\mathbf{C} = \mathbf{S}\,\mathbf{C}\,\boldsymbol{\epsilon}, \tag{62}$$

where $\mathbf{C} \in \mathbb{R}^{B \times B}$ collects generalized eigenvectors (orbital coefficients) and $\boldsymbol{\epsilon} = \mathrm{diag}(\epsilon_1, \ldots, \epsilon_B)$ contains the corresponding eigenvalues (orbital energies), ordered such that $\epsilon_1 \leq \cdots \leq \epsilon_B$. We write $(\hat{\mathbf{C}}, \hat{\boldsymbol{\epsilon}})$ for the solution obtained from $(\hat{\mathbf{H}}, \mathbf{S})$ and $(\mathbf{C}^\star, \boldsymbol{\epsilon}^\star)$ for that obtained from $(\mathbf{H}^\star, \mathbf{S})$ under the same numerical setup.

**Hamiltonian MAE.** This metric measures element-wise reconstruction accuracy of the Hamiltonian matrix:

$$\mathrm{MAE}(\mathbf{H}) = \frac{1}{B^2} \sum_{p=1}^{B} \sum_{q=1}^{B} \left| \hat{H}_{pq} - H_{pq}^\star \right|. \tag{63}$$

Here, $H_{pq}^\star$ and $\hat{H}_{pq}$ denote entries of the reference and predicted Hamiltonians in the same basis ordering. We report this quantity in $\mu$Ha in the main tables, following standard Hamiltonian benchmarks.

**Occupied orbital energy MAE ($\epsilon_{\text{occ}}$).**    To assess spectral fidelity in the energy range relevant to the ground state, we compare orbital energies for occupied orbitals only:

$$\text{MAE}(\epsilon_{\text{occ}}) = \frac{1}{n_{\text{occ}}} \sum_{p \in \mathcal{I}_{\text{occ}}} \left| \hat{\epsilon}_p - \epsilon_p^\star \right|. \tag{64}$$

This metric focuses on the part of the spectrum that directly determines the ground-state density in RKS. Importantly, the energies $\hat{\epsilon}_p, \epsilon_p^\star$ are obtained by solving the generalized eigenproblem Equation (62) with the same overlap matrix $\mathbf{S}$.

**Orbital coefficient similarity ($\mathcal{S}_c$).**    We measure alignment between predicted and reference orbital coefficient vectors. Let $\hat{\mathbf{c}}_p$ and $\mathbf{c}_p^\star$ denote the $p$-th columns of $\hat{\mathbf{C}}$ and $\mathbf{C}^\star$, respectively. Because each eigenvector is defined up to a sign (and may be unstable within degenerate or near-degenerate subspaces), we use an absolute cosine similarity and average over occupied orbitals:

$$\mathcal{S}_C(\hat{\mathbf{C}}, \mathbf{C}^\star) = \frac{1}{n_{\text{occ}}} \sum_{p=1}^{n_{\text{occ}}} \frac{\left| \left\langle \hat{\mathbf{c}}_p, \mathbf{c}_p^\star \right\rangle \right|}{\|\hat{\mathbf{c}}_p\|_2 \|\mathbf{c}_p^\star\|_2}. \tag{65}$$

Here $\langle \cdot, \cdot \rangle$ is the standard Euclidean inner product and $\| \cdot \|_2$ is the $\ell_2$ norm. The absolute value makes the score invariant to sign flips ($\mathbf{c}_p \mapsto -\mathbf{c}_p$), which do not change the electron density.

**HOMO, LUMO, and gap ($\epsilon_{\text{HOMO}}, \epsilon_{\text{LUMO}}, \epsilon_\Delta$).**    Frontier orbitals are key for chemical reactivity and are often sensitive to generalization. Under the RKS occupation rule, the HOMO index is $p_{\text{H}} = n_{\text{occ}}$ and the LUMO index is $p_{\text{L}} = n_{\text{occ}} + 1$. We report absolute errors for HOMO energy, LUMO energy, and the HOMO–LUMO gap:

$$\text{MAE}(\epsilon_{\text{HOMO}}) = \left| \hat{\epsilon}_{p_{\text{H}}} - \epsilon_{p_{\text{H}}}^\star \right|, \tag{66}$$

$$\text{MAE}(\epsilon_{\text{LUMO}}) = \left| \hat{\epsilon}_{p_{\text{L}}} - \epsilon_{p_{\text{L}}}^\star \right|, \tag{67}$$

$$\text{MAE}(\Delta\epsilon) = \left| (\hat{\epsilon}_{p_{\text{L}}} - \hat{\epsilon}_{p_{\text{H}}}) - (\epsilon_{p_{\text{L}}}^\star - \epsilon_{p_{\text{H}}}^\star) \right|. \tag{68}$$

These quantities are computed from the eigenvalues of Equation (62) and therefore probe whether the predicted Hamiltonian reproduces the correct frontier spectrum relative to the reference.

### E.3. Experimental setup

**Environment.** All experiments were run using a single GPU per model. Most runs were conducted on NVIDIA RTX 3090 and NVIDIA RTX A6000 GPUs. For larger model variants that required higher memory and throughput, we used NVIDIA H100 and NVIDIA B200 GPUs. For the extra-large model variant, we trained on an NVIDIA H100 with gradient accumulation over 4 steps to fit within memory constraints.

Our implementation is based on PyTorch 2.1.2 and PyG 2.3.0, compiled with CUDA 12.1. We additionally rely on standard scientific and atomistic ML libraries, including PySCF (Sun et al., 2018), e3nn (Geiger & Smidt, 2022), and ASE (Larsen et al., 2017). Full environment specifications (package versions and hardware details) will be released upon publication to support reproducibility.

To improve reproducibility, we fix random seeds wherever possible. Minor non-determinism may still remain due to GPU kernels and low-level library implementations.

**Shared hyperparameters.** For fair comparison, we follow the hyperparameter choices of the baseline QHNet (Yu et al., 2023b) whenever possible. QHFlow2 shares most architectural and training settings with QHNet so that performance differences primarily reflect our flow-matching formulation rather than extensive hyperparameter tuning. Key training and inference hyperparameters for each dataset are summarized in Table 9.

*Table 6.* Training and inference hyperparameters of QHFlow2 used across datasets.

| Hyperparameter | Description | QH9 | MD17 |
|---|---|---|---|
| Learning Rate | Initial learning rate | 1e-3 | 1e-3 |
| Minimum Learning Rate | Minimum learning rate | 1e-7 | 1e-9 |
| Batch Size | Number of molecules per batch | 32 | 10 |
| Scheduler | Learning rate scheduler | Polynomial | Polynomial |
| LR Warmup Steps | Warmup steps for linear warmup | 1,000 | 1,000 |
| Max Steps | Maximum number of training steps | 260,000 | 200,000 |
| Fine-tuning LR | Initial learning rate for fine-tuning | 1e-5 | – |
| Fine-tuning Minimum LR | Minimum learning rate for fine-tuning | 1e-7 | – |
| Fine-tuning Steps | Maximum number of fine-tuning steps | 60,000 | – |
| Prior Distribution | Prior for flow matching | TE | TE |
| Sampling Steps | Number of ODE steps at inference | 3 | 3 |

**Architecture hyperparameters.** Table 7 reports the architectural and input-feature settings for QHFlow2. Across all experiments, the model is conditioned on the time-dependent Hamiltonian state $H_t$. We additionally provide the overlap matrix $S$ for QH9, while omitting it for MD17/rMD17. The remaining hyperparameters specify the equivariant backbone: the maximum irrep degree $\ell_{\max}$, embedding and bottleneck widths, the numbers of SO(2) backbone layers and two-stage update layers, the SO(2) neighbor cutoff radius, and the channel sizes used for spherical and edge features.

*Table 7.* QHFlow2 architecture and input settings by model size.

| Setting | QHFlow2-S | QHFlow2-S (md) | QHFlow2-M | QHFlow2-M (md) | QHFlow2-L | QHFlow2-XL |
|---|---|---|---|---|---|---|
| Parameters | 14M | 12M | 43.3M | 42.8M | 183M | 990M |
| Using $H_t$ as embedding | True | True | True | True | True | True |
| Using $H_{\text{init}}$ as embedding | False | False | False | False | True | True |
| Using $S$ as embeding | True | False | True | False | True | True |
| Model order ($\ell_{\max}$) | 4 | 4 | 4 | 4 | 4 | 4 |
| Embedding dimension | 64 | 64 | 128 | 128 | 256 | 512 |
| Bottleneck hidden size | 32 | 32 | 64 | 32 | 64 | 128 |
| Number of SO(2) backbone layers | 3 | 2 | 3 | 3 | 4 | 5 |
| Number of two-stage update layers | 2 | 2 | 2 | 2 | 2 | 3 |
| SO(2) max radius (Å) | 5.0 | 5.0 | 5.0 | 5.0 | 5.0 | 5.0 |
| Sphere channels | 64 | 64 | 128 | 128 | 256 | 512 |
| Edge channels | 64 | 64 | 128 | 128 | 256 | 512 |

**Hamiltonian model hyperparameters.** In Table 8, we report the key architectural choices of baseline Hamiltonian predictors. Training strategies follow the original papers or are kept consistent with our main training setup; For model-scale comparisons, we vary only the capacity-related parameters while keeping the remaining architectural choices fixed.

*Table 8.* Architecture and input hyperparameters of Hamiltonian predictors.

| Hyperparameter | SPHNet(18M/36M) | QHNet | QHFlow |
|---|---|---|---|
| Using $S$ as embedding | False / False | False | True |
| Model order ($\ell_{\max}$) | 4 / 4 | 4 | 4 |
| Embedding dimension | 128 / 184 | 128 | 128 |
| Bottleneck hidden size | 32 / 32 | 32 | 32 |
| Number of backbone GNN layers | 4 / 4 | 5 | 5 |
| Cutoff radius (Å) | 15 / 15 | 15 | 15 |
| Sphere channels | 128 / 184 | 128 | 128 |
| Edge channels | 32 / 32 | 32 | 32 |

# F. MLFF Experimental Details

## F.1. Dataset and Preprocessing

**Dataset.** We employed high-fidelity recalculated trajectories based on the MD17 benchmark. Specifically, we utilized the Revised MD17 (rMD17) dataset for four molecules: aspirin, naphthalene, salicylic acid, and ethanol. Additionally,

we included uracil, malondialdehyde, and ethanol obtained from recalculated MD17 trajectories. For consistency across experiments, we employed a subset of 30,000 configurations for each molecule.

**Data Splits and Representation.** All molecular configurations are represented as atomistic graphs where nodes correspond to atoms with atomic number $Z$ and 3D coordinates, and edges connect atom pairs within a model-specific cutoff radius ($r_c$). The graphs are treated as undirected, implemented via bidirectional directed edges (i.e., both $i \to j$ and $j \to i$ are included) without periodic boundary conditions.

For each molecule, we employed fixed train/validation/test splits of 25,000/500/4,500 configurations, totaling 30,000 samples. These splits were pre-generated and consistently applied across all models to ensure fair comparison. For low-data experiments, we keep the validation and test sets unchanged and subsample the training set to N $\in \{10,000, 5,000, 1,000\}$ by taking the first N configurations from the pre-shuffled 25,000-sample training pool. Since the training pool is already randomly ordered by construction, this procedure yields nested random subsets while remaining fully deterministic. Energies are reported in eV, and forces in eV/Å.

**Target Normalization and Preprocessing.** We adopted the standard hyperparameters and configurations provided by the MDsim framework for all reference models to ensure reproducibility.

- **GemNet-T & DimeNet:** We did not apply explicit label normalization (`normalize_labels: False`), consistent with their standard implementations.

- **NequIP:** We used the default configuration which automatically applies per-species rescaling based on dataset statistics (shifting by mean per-atom energy and scaling by force RMS).

### F.2. Model Architectures and Hyperparameters

We evaluate three reference models: NequIP, GemNet-T, and DimeNet. Table 9 summarizes the key hyperparameters and architectural details adopted in our experiments. To ensure fair comparison, consistent loss weights were applied across models where applicable.

*Table 9.* Hyperparameters and architectural details for NequIP, GemNet-T, and DimeNet. Note that the loss weights ($\lambda$) denote the ratio between energy and force terms in the objective function.

| Hyperparameter | NequIP | GemNet-T | DimeNet |
|---|---|---|---|
| **Architecture** | | | |
| Cutoff Radius ($r_c$) | 4.0 Å | 5.0 Å | 5.0 Å |
| Interaction Blocks / Layers | 5 layers | 4 blocks | 6 blocks |
| RBF Basis Size | 8 | 6 (radial), 7 (spherical) | 6 (radial), 7 (spherical) |
| Hidden Dimension | 32 | 128 (atom), 64 (triplet) | 128 |
| **Training Objective** | | | |
| Loss Weights ($\lambda_E : \lambda_F$) | 1 : 1000 | 1 : 1000 | 1 : 1000 |
| **Optimization** | | | |
| Optimizer | Adam | AdamW | AdamW |
| Learning Rate | $5 \times 10^{-3}$ | $1 \times 10^{-3}$ | $1 \times 10^{-3}$ |
| Batch Size | 5 | 1 | 32 |
| Max Epochs | 2000 | 10000 | 10000 |
| **Complexity** | | | |
| # of Parameters | 1,053,816 | 1,890,125 | 2,100,070 |

### F.3. Training Infrastructure

**Hardware and Software.** All models were trained on a high-performance computing cluster utilizing NVIDIA Ampere and Hopper architecture GPUs (including A100, H100, and H200). Each training session was executed on a single GPU node. The software environment was configured using the MDsim framework[0]. Key dependencies include PyTorch 2.0 and CUDA 11.8, chosen to ensure compatibility across different GPU architectures.

---

[0]The source code and environment details are available at: https://github.com/kyonofx/MDsim

**Computational Cost.** The training duration varied significantly depending on the model architecture and convergence rate. On average, a full training run required approximately 24 to 48 hours of wall-clock time on a single A100 GPU.

### F.4. Evaluation metric and protocol

We assessed the model performance on a held-out test set comprising 4,500 configurations. We report the Mean Absolute Error (MAE) and Root Mean Squared Error (RMSE) for both potential energy ($E$) and atomic forces ($F$).

- **Energy Metrics:** MAE and RMSE are computed on the predicted total potential energy per conformation (Unit: eV).

- **Force Metrics:** MAE and RMSE are calculated component-wise for all atomic forces (Unit: eV/Å), averaged over all atoms and spatial directions ($x, y, z$).

## G. Additional results

### G.1. Ablation on QHFlow2 design

**Ablation of two-stage edge update.** Table 10 ablates the two-stage update by toggling the reference-frame cue $g_{\mathrm{ref}}$ and the SO(2) edge cutoff, while fixing the SO(2) backbone and parameter budget (43M) and training on QH9Stable-iid. Without $g_{\mathrm{ref}}$, a larger cutoff is needed to reduce off-diagonal error. Enabling $g_{\mathrm{ref}}$ improves all metrics across cutoffs and narrows the gap between small and large cutoffs, giving strong performance already at the default 5 Å. The best result is obtained with $g_{\mathrm{ref}}$ and a 15 Å cutoff.

*Table 10.* **SO(2) ablations over two-stage update.** All variants use the same SO(2) backbone and are trained on QH9Stable-iid with the parameter budget fixed to 43M. We vary whether the reference-frame cue $g_{\mathrm{ref}}$ is enabled and the SO(2) edge cutoff (default: 5 Å). We report Hamiltonian MAE and its diagonal/off-diagonal components (lower is better).

| $g_{\mathrm{ref}}$ | cutoff | $H$ MAE ↓ | Diag.↓ | Off-diag. ↓ |
|---|---|---|---|---|
| ✓ | – | 27.91 | 31.37 | 14.35 |
| ✗ | 5 | 14.91 | 22.89 | 27.64 |
| ✗ | 8 | 9.58 | 22.06 | 8.63 |
| ✗ | 12 | 9.36 | 21.79 | 8.41 |
| ✗ | 15 | 9.86 | 21.11 | 9.00 |
| ✓ | 5 | 9.18 | 18.45 | 8.47 |
| ✓ | 12 | 8.55 | 17.77 | 7.84 |
| ✓ | 15 | **8.43** | **17.30** | **7.75** |

**Ablation of flow matching.** Table 11 compares flow matching with a regression-style variant of QHFlow2 that fixes $t = 0$ and $H_t = 0$ during training and evaluation, using the same architecture and trained on QH9Stable-iid. Flow matching improves Hamiltonian MAE on both diagonal and off-diagonal blocks and reduces orbital energy error. The gap between 1-step and 3-step is small, and we find that 1–3 steps are generally sufficient in our setting.

*Table 11.* **Regression vs. Flow-based Hamiltonian prediction.** Comparison between pointwise regression and flow-based modeling with different numbers of flow steps. Errors are reported separately for diagonal and off-diagonal Hamiltonian blocks. Lower is better.

| Metric | Regression | Flow-based | |
|---|---|---|---|
| | | 1-step | 3-step |
| All H MAE ↓ | 10.77 | 9.19 | **9.18** |
| Diagonal H MAE ↓ | 21.01 | 18.49 | **18.46** |
| Off-diagonal H MAE ↓ | 10.00 | 8.55 | **8.47** |

**Ablation of initial Hamiltonian.** Table 12 ablates the use of an initial Hamiltonian by evaluating models without providing an initial guess at inference. QHFlow2 retains strong accuracy under this setting and yields energy and force errors in the MLIP range on MD17. In contrast, prior Hamiltonian predictors show reasonable matrix-level scores but fail to produce accurate energies when the initial Hamiltonian is removed, highlighting the robustness of QHFlow2 for direct downstream evaluation.

*Table 12.* **MD17 benchmark w/o initial Hamiltonian.** Best results are shown in **bold**.

| Model | Param. | Ethanol (9 atoms) | | | | | malondialdehyde (9 atoms) | | | | | Uracil (12 atoms) | | | | |
|---|---|---|---|---|---|---|---|---|---|---|---|---|---|---|---|---|
| | | $H \downarrow$ | $\epsilon_{\text{occ}} \downarrow$ | $S_c \uparrow$ | Energy$\downarrow$ | Force$\downarrow$ | $H \downarrow$ | $\epsilon_{\text{occ}} \downarrow$ | $S_c \uparrow$ | Energy$\downarrow$ | Force$\downarrow$ | $H \downarrow$ | $\epsilon_{\text{occ}} \downarrow$ | $S_c \uparrow$ | Energy$\downarrow$ | Force$\downarrow$ |
| *w/o initial Hamiltonian* | | | | | | | | | | | | | | | | |
| SchNOrb | – | 187.4 | 334.4 | **100.00** | – | – | 191.1 | 400.6 | 99.00 | – | – | 227.8 | 1760. | 90.00 | – | – |
| PhiSNet | – | 20.09 | 102.04 | 99.81 | – | – | 21.31 | 100.6 | 99.89 | – | – | 18.65 | 143.36 | 99.86 | – | – |
| QHNet | 20M | 27.99 | 99.33 | 99.99 | – | – | 29.60 | 100.16 | 99.92 | – | – | 26.80 | 127.93 | 99.99 | – | – |
| HELM | – | **5.79** | – | – | 28.93 | – | **4.86** | – | – | 20.15 | – | **3.61** | – | – | 22.69 | – |
| **QHFlow2-m** | 43M | 12.49 | 21.32 | **100.00** | **0.136** | 1.320 | 13.14 | 26.64 | 99.97 | **1.145** | 2.296 | 10.47 | 29.13 | 99.89 | **0.638** | 2.414 |

*Table 13.* **QH9 benchmark init.** Results shown in **bold** denote the best result in each column, whereas those that are underlined indicate the second best.

| Dataset | Model | Param. | $H \downarrow [\mu E_h]$ | $\epsilon_{\text{occ}} \downarrow [\mu E_h]$ | $S_c \uparrow [\%]$ | $\epsilon_{\text{LUMO}} \downarrow [\mu E_h]$ | $\epsilon_{\text{HOMO}} \downarrow [\mu E_h]$ | $\epsilon_\Delta \downarrow [\mu E_h]$ |
|---|---|---|---|---|---|---|---|---|
| | QHFlow | 28M | 22.95 | 119.67 | 99.51 | 437.96 | 179.48 | 553.87 |
| | **QHFlow2**-m | 43M | 9.18 | 59.23 | 99.73 | 175.68 | 77.67 | 215.99 |
| Stable-iid | **QHFlow2**-l | 183M | **5.93** | **45.73** | **99.81** | **103.32** | **50.50** | **119.62** |
| | **QHFlow2**-m (w/o init) | 43M | 25.50 | 279.84 | 98.31 | 834.31 | 255.53 | 1025.42 |
| | **QHFlow2**-m (+ diagonal init.) | 43M | 22.81 | 130.88 | 99.39 | 965.63 | 218.24 | 1057.18 |
| | **QHFlow2**-l (+ diagonal init.) | 183M | 16.22 | 150.04 | 99.52 | 794.71 | 109.16 | 825.32 |

We additionally explore a simple *diagonal initialization* scheme. Since the on-atom (diagonal) Hamiltonian blocks are rotationally invariant for isolated atoms, we initialize each diagonal block using the corresponding atom's reference Hamiltonian computed from a one-time atomic DFT calculation. This precomputation is cheap and amortized across molecules. Table 13 summarizes the results. Diagonal initialization improves over using no initialization, but still lags behind the full initialization setting, especially for orbital energies and the HOMO–LUMO gap. This suggests that accurate initialization must also capture inter-atomic couplings and environment-dependent on-site effects, which are not provided by isolated-atom diagonal blocks.

**Effect of Hamiltonian readout degree.** We study the effect of the maximum angular degree used in the final Hamiltonian readout, denoted by $L_{\text{readout}}$. Here, the readout is the tensor-expansion module that maps the equivariant feature $q_{ij}$, produced after the SO(2) backbone and two-stage pair update, into an atom-pair Hamiltonian block:

$$\tilde{\mathbf{H}}_{ij} = E_{Z_i, Z_j}(q_{ij}), \qquad E_{Z_i, Z_j} : \{q[\ell]\}_{\ell=0}^{L_{\text{readout}}} \to \mathbb{R}^{B_i \times B_j}. \tag{69}$$

This ablation fixes the message-passing degree to $L_{\text{MP}} = 4$ and varies only the maximum degree retained in the tensor-expansion readout input.

For an orbital block with angular momenta $(\ell_1, \ell_2)$, a complete tensor expansion requires Clebsch–Gordan paths with

$$\ell_{\text{in}} = |\ell_1 - \ell_2|, \ldots, \ell_1 + \ell_2. \tag{70}$$

Thus, reducing $L_{\text{readout}}$ keeps the output matrix shape unchanged but removes some angular coupling paths from the final expansion. Since def2-SVP includes $s$, $p$, and $d$ orbitals for heavy atoms, the $d$–$d$ block requires paths up to $\ell_{\text{in}} = 4$. Consequently, $L_{\text{readout}} = 2$ leaves the $p$–$d$ and $d$–$d$ blocks incomplete, while $L_{\text{readout}} = 3$ still misses the highest $d$–$d$ coupling. As shown in Table 14, the truncated readout severely degrades Hamiltonian prediction, and $L_{\text{readout}} = 4$ gives the best overall accuracy.

*Table 14.* Effect of the Hamiltonian readout degree $L_{\text{readout}}$ on QH9-stable-id. $L_{\text{readout}}$ denotes the maximum angular degree retained in the tensor-expansion readout input. We fix $L_{\text{MP}} = 4$ and vary only $L_{\text{readout}}$ at matched parameter count. CG paths denotes the number of Clebsch–Gordan coupling paths used in the final Hamiltonian readout.

| $L_{\text{readout}}$ | Params (M) | CG paths | H MAE ($\mu$Ha) | $\epsilon_{\text{occ}}$ ($\mu$Ha) | $S_c$ (%) |
|---|---|---|---|---|---|
| 1 | 42.94 | 9 | 1111.1 | 424122.8 | 24.84 |
| 2 | 43.21 | 15 | 322.1 | 285.1 | 99.20 |
| 3 | 43.31 | 18 | 72.5 | 62.8 | 99.73 |
| 4 | **43.34** | **19** | **9.4** | **61.1** | **99.74** |

## G.2. Exact values for plotted results

This section reports the exact numerical values used to generate the figures in the main text. Table 15 lists the values used in Figure 2, and Table 16 lists the values used in Figure 3.

*Table 15.* **MLIP vs. QHFlow2 comparison on MD17.** Energy and force errors are reported in meV and meV/Å. Molecules Eta., Malon., and Ura. belong to MD17, while SA, Naph., and Asp. belong to rMD17. Best results are shown in **bold**, and second best are underlined.

| | **MD17** | | | | | | **rMD17** | | | | | |
| | Eta. | | Malon. | | Ura. | | SA | | Naph. | | Asp. | |
| **Model** | E↓ | F↓ | E↓ | F↓ | E↓ | F↓ | E↓ | F↓ | E↓ | F↓ | E↓ | F↓ |
| *MLIP* | | | | | | | | | | | | |
| DimeNet | 10.32 | 1.84 | 12.98 | 1.16 | 10.68 | 2.13 | 12.21 | 4.06 | 11.92 | 2.83 | 26.04 | 5.45 |
| Gemnet-T | 7.44 | 3.77 | 8.12 | 5.73 | 9.37 | 5.51 | 15.64 | 4.85 | 19.62 | 2.87 | 10.27 | 5.28 |
| Nequip | 0.33 | 0.61 | 0.73 | 0.88 | 0.45 | 0.60 | 1.71 | 1.43 | 2.01 | 0.76 | 1.87 | 2.08 |
| *QHFlow2* | | | | | | | | | | | | |
| **s-14M** | 0.005 | 0.36 | 0.058 | 0.57 | 0.234 | 1.01 | 0.110 | 1.48 | 0.063 | 1.17 | 0.124 | 1.42 |
| **m-43M** | **0.003** | **0.34** | **0.010** | **0.51** | **0.062** | **0.90** | **0.070** | **1.38** | **0.040** | 1.10 | **0.084** | **1.35** |

*Table 16.* **MLIP vs. QHFlow2 comparion on QH9.** We report a subset of orbital- and energy-related metrics for representative MLFF baselines on QM9 and QHFlow-v2 variants on QH9-stable (iid).

| Dataset | Model | $H(\downarrow)$ | LUMO(↓) | HOMO(↓) | GAP(↓) | Energy(↓) |
|---|---|---|---|---|---|---|
| | *MLFF baseline* | | | | | |
| QM9[†] | Equiformer | – | 551.24 | 514.49 | 1102.48 | 6.59 |
| | MACE | – | 808.48 | 698.24 | 1543.47 | 4.10 |
| | EquiformerV2 | – | 514.49 | 477.74 | 1065.73 | 6.17 |
| | *QHFlow2* | | | | | |
| QH9-stable | **QHFlow2-s** | 16.29 | 304.69 | 148.42 | 411.41 | 1.42 |
| (iid) | **QHFlow2-m** | 9.18 | 175.68 | 77.67 | 215.99 | 0.69 |
| | **QHFlow2-l** | 5.93 | 103.32 | 50.50 | 119.62 | 0.32 |
| | **QHFlow2-xl** | **4.58** | **73.19** | **40.30** | **87.74** | **0.21** |

## G.3. SCF Acceleration

We evaluate whether predicted Hamiltonians provide useful initial guesses for iterative KS-DFT. All models are trained on QH9-stable-iid, and we randomly sample 300 molecules from the corresponding test split. For each molecule, we use the predicted Hamiltonian $\hat{\mathbf{H}}$ to solve the generalized eigenvalue problem with the overlap matrix $\mathbf{S}$, assign occupations by eigenvalue ordering under the restricted closed-shell setting, and construct the initial density matrix. This density matrix is then used as the initial guess for SCF.

**SCF setup.**    To ensure a controlled comparison, all runs use the same DFT and solver settings; only the initialization differs. We use the B3LYP(VWN5)/def2-SVP setting to match the QH9 reference configuration. All calculations are performed with GPU4PySCF on a single NVIDIA H200 GPU, using density fitting and grid level 3. We set the SCF convergence tolerance to `conv_tol` $= 10^{-9}$ and keep the DIIS procedure at the PySCF default configuration. The MinAO baseline uses the default PySCF initial guess, while each Hamiltonian predictor provides an initial density constructed from its predicted Hamiltonian. Post-processing for Hamiltonian predictors includes Hamiltonian matrix reconstruction, orbital reordering when needed, and density-matrix construction before entering SCF.

SCF fixed points can be sensitive to small numerical differences, including grid settings, integral screening, hardware backend, and the precise B3LYP parametrization. In particular, B3LYP is implemented with either VWN5 or VWN3 correlation across quantum chemistry packages. We therefore match the VWN5 convention used in QH9; otherwise, a functional mismatch can lead to a different self-consistent solution and inflate the apparent number of remaining SCF cycles. We include two references: *Ref*, which initializes SCF with the dataset Hamiltonian $\mathbf{H}$ and represents the data-side limit, and *RC*, which initializes with a Hamiltonian recomputed under our evaluation settings and represents the solver-side limit.

**Metrics.** We report both SCF cycle reduction and wall-clock speedup. The cycle ratio is normalized by the MinAO baseline. The end-to-end speedup is computed as

$$\text{Speedup} = \frac{\text{SCF time from scratch}}{\text{Inference} + \text{Post-processing} + \text{SCF with predicted } \hat{\mathbf{H}}}, \tag{71}$$

where the denominator includes all model-side overheads before and during SCF. As shown in Figure 4 and Table 17, stronger Hamiltonian predictors consistently reduce SCF cycles and wall-clock time. QHFlow2-l achieves the best end-to-end speedup of $2.18\times$, while QHFlow2-xl slightly saturates at $2.16\times$ because its larger inference cost offsets the additional cycle reduction.

*Table 17.* SCF acceleration on QH9-stable under the B3LYP(VWN5)/def2-SVP setting. All runs use GPU4PySCF with density fitting, grid level 3, PySCF-default DIIS, and `conv_tol` $= 10^{-9}$ on a single NVIDIA H200 GPU. Only the initialization differs. Cycle ratio is normalized by the MinAO baseline. Post-processing includes Hamiltonian matrix reconstruction, orbital reordering, and density-matrix construction.

| Model | Params | H MAE ($\mu$Ha) | Cycles | Ratio | Infer. (ms) | Post-Proc. (ms) | SCF (ms) | Speedup |
|---|---|---|---|---|---|---|---|---|
| MinAO (scratch) | — | — | 11.03 | 100% | — | — | 9,733 | 1.00$\times$ |
| QHNet | 27M | 44.66 | 7.18 | 65.1% | 9.4 | 24.8 | 5,610 | 1.73$\times$ |
| QHFlow | 30M | 22.01 | 6.47 | 58.7% | 10.7 | 25.2 | 5,332 | 1.83$\times$ |
| QHFlow2-s | 12M | 15.75 | 6.15 | 55.8% | 4.3 | 23.6 | 5,246 | **1.82$\times$** |
| QHFlow2-m | 43M | 8.92 | 5.73 | 51.9% | 7.3 | 26.1 | 4,830 | **1.98$\times$** |
| QHFlow2-l | 183M | 5.66 | 5.23 | 47.4% | 14.8 | 25.6 | 4,369 | **2.18$\times$** |
| QHFlow2-xl | 990M | 4.34 | 5.08 | 46.1% | 67.0 | 26.5 | 4,331 | **2.16$\times$** |
| Ref (data) | — | — | 3.10 | 28.1% | — | — | — | — |
| RC (solver) | — | — | 3.10 | 28.1% | — | — | — | — |

### G.4. Energy and force calculation runtime on MD17

Table 18 reports a runtime breakdown for MLIPs and Hamiltonian predictors on MD17, averaged over 100 structures each from salicylic acid, naphthalene, and aspirin. For MLIPs, the reported time is end-to-end energy and force evaluation (network energy prediction plus autodiff forces). For Hamiltonian predictors, we separate (i) Hamiltonian inference and (ii) a downstream PySCF step that maps $\hat{\mathbf{H}}$ to energies and analytic forces. We use the same PySCF pipeline for all Hamiltonian models, so the downstream cost is comparable across predictors and provides a controlled runtime baseline. Under this decomposition, differences between Hamiltonian predictors are reflected in the Hamiltonian inference time, while the overall cost is currently dominated by the shared PySCF post-processing. A remaining limitation is that this evaluation depends on a DFT solver stack; while it ensures physically grounded energies and forces, the total runtime inherits solver-side overhead that is not optimized by the predictor itself.

### G.5. Comparison with eSEN on rMD17

We additionally compare against eSEN (Fu et al., 2025), a recent SO(2)-equivariant MLIP whose backbone we adopt in QHFlow2. Since eSEN does not have reported results on MD17 or rMD17, we train it ourselves under the same evaluation protocol as our other MLIP baselines. This comparison situates QHFlow2 against a recent MLIP at matched force accuracy, while emphasizing that QHFlow2 derives energies and forces from the predicted Hamiltonian, which additionally provides access to orbitals, electron density, and other electronic-structure quantities.

eSEN reaches force accuracy in the same range as QHFlow2 on both molecules, consistent with its strong performance on atomic-level prediction tasks. We emphasize that the comparison is not on the same footing: eSEN predicts energies and forces directly as scalar and vector outputs, whereas QHFlow2 predicts the full Hamiltonian and derives energies and forces from it, which additionally provides access to orbitals, electron density, and other electronic-structure quantities that MLIPs do not produce.

*Table 18.* **Speed and runtime breakdown on the MD benchmark.** Runtime is reported in seconds (s), and speed is reported in samples/s, averaged over 100 structures each from salicylic acid, naphthalene, and aspirin. Inf. is the model forward-pass time (end-to-end energy/force evaluation for MLIPs; Hamiltonian prediction for Hamiltonian models). For Hamiltonian models, E and F are the downstream PySCF times to compute energy and forces from the predicted Hamiltonian $\hat{\mathbf{H}}$, so Total = Inf. + E + F, while Total = Inf. + F for MLIPs. Best results are in **bold** within each model family.

| Model | Inf.$\downarrow$ | E | F | Total $\downarrow$ | Speed $\uparrow$ |
|---|---|---|---|---|---|
| *MLIPs* | | | | Inf./E: direct model output; F: autograd | |
| Dimenet | 0.121 | – | 0.003 | 0.124 | 8.03 |
| Gemnet-T | **0.086** | – | 0.001 | **0.087** | **11.44** |
| Nequip | 0.164 | – | 0.002 | 0.167 | 5.97 |
| *Hamiltonian Models* | | | | Inf.: predict $\hat{\mathbf{H}}$; E/F: PySCF from $\hat{\mathbf{H}}$ | |
| SPHNet-18M | 0.089 | 0.504 | 0.883 | 1.483 | 0.694 |
| SPHNet-36M | 0.091 | 0.507 | 0.881 | 1.477 | 0.693 |
| **QHFlow2-s** | **0.086** | 0.506 | 0.882 | **1.474** | **0.699** |
| **QHFlow2-m** | 0.159 | 0.507 | 0.890 | 1.556 | 0.663 |

*Table 19.* **eSEN energy and force accuracy on rMD17.** Energy and force errors are reported in meV and meV/Å. Two eSEN model sizes (S, M) are evaluated across five training set sizes on Aspirin (Asp.) and Salicylic acid (SA).

| | Aspirin | | | | Salicylic acid | | | |
|---|---|---|---|---|---|---|---|---|
| | eSEN-S (449K) | | eSEN-M (2.5M) | | eSEN-S (449K) | | eSEN-M (2.5M) | |
| Train size | E$\downarrow$ | F$\downarrow$ | E$\downarrow$ | F$\downarrow$ | E$\downarrow$ | F$\downarrow$ | E$\downarrow$ | F$\downarrow$ |
| 1k | 4.1 | 10.2 | 2.8 | 8.8 | 1.4 | 7.0 | 1.2 | 6.1 |
| 2.5k | 1.8 | 5.5 | 2.0 | 6.1 | 0.8 | 3.6 | 1.0 | 4.9 |
| 5k | 1.3 | 3.7 | 1.3 | 3.7 | 0.5 | 2.5 | 0.4 | 2.3 |
| 15k | 2.6 | 2.9 | 1.0 | 2.2 | 2.5 | 2.1 | 1.4 | 1.1 |
| 25k | 1.0 | 2.5 | 0.7 | 1.6 | 0.3 | 1.7 | 0.2 | 1.1 |

## G.6. Hamiltonian prediction on $\nabla^2$DFT

To assess QHFlow2 on a larger and more chemically diverse benchmark, we evaluate on $\nabla^2$DFT, a dataset of drug-like organic molecules computed at the $\omega$B97X-D/def2-SVP level. All models are trained on the $\mathcal{D}^{\text{medium}}$ train-10k split (9,689 molecules, 49,725 conformations) and evaluated on test-2k under the without-init setting (no initial Hamiltonian is provided). For a controlled comparison, we cap QHFlow2 training at a budget of 1,920 GPU hours, matching the cost of the baselines. Table 20 reports Hamiltonian MAE alongside parameter count and training cost.

QHFlow2 attains the lowest Hamiltonian error across all model sizes under this matched training budget. Even the smallest variant, QHFlow2-s ($46.12\,\mu E_h$), outperforms all baselines including HELM ($59\,\mu E_h$), and QHFlow2-l further reduces the error to $17.92\,\mu E_h$. These gains confirm that the accuracy advantages observed on QH9 and the MD benchmark transfer to a different functional and a more diverse chemical space.

*Table 20.* **Hamiltonian prediction on $\nabla^2$DFT.** $\nabla^2$DFT contains organic molecules computed at the $\omega$B97X-D/def2-SVP level. All models are trained on the $\mathcal{D}^{\text{medium}}$ train-10k split (9,689 molecules, 49,725 conformations) and evaluated on test-2k under the without-init setting (no initial Hamiltonian is provided). Baselines are evaluated from pre-trained checkpoints on the official GitHub and tested on the same splits. QHFlow2 training is stopped at a budget of 1,920 GPU hours to match the baselines, on 2× NVIDIA H200 (144 GB) with DDP. [†]SchNOrb result is taken from the reference paper (full test set).

| Model | Params | GPU hours | H MAE ($\mu E_h$) $\downarrow$ |
|---|---|---|---|
| SchNOrb[†] | — | 1,920 | 19600 |
| PhiSNet | 21M | 1,920 | 340 |
| QHNet | 22M | 1,920 | 530 |
| HELM | — | 1,560 | 59 |
| QHFlow2-s | 12M | 1,920 | 46.12 |
| QHFlow2-m | 43M | 1,920 | 27.62 |
| QHFlow2-l | 183M | 1,920 | **17.92** |

*Table 21.* **Zero-shot generalization to molecules larger than the training distribution (up to 98 atoms).** QHFlow2 models trained on QH9-stable-id are evaluated on PubChemQH test molecules (CHNOF, 45–98 atoms) without retraining. $N$ denotes the number of test molecules per bin. QHFlow2-xl (990M) is excluded due to OOM for molecules with $\geq 60$ atoms. All inference is run on a single NVIDIA H200 GPU (144 GB).

| Model | Atoms | $N$ | H MAE ($\mu$Ha) $\downarrow$ | E MAE (meV/at.) $\downarrow$ | SCF cycle ratio (%) $\downarrow$ |
|---|---|---|---|---|---|
| QHFlow2-s | 45–59 | 8 | 82.7 | 2.0 | 67.3 |
| | 60–79 | 9 | 58.5 | 444.5 | 75.0 |
| | 80–98 | 9 | 51.6 | 2036.9 | 213.7 |
| QHFlow2-m | 45–59 | 8 | 81.1 | 1.9 | 67.3 |
| | 60–79 | 9 | 53.3 | 7.5 | 60.2 |
| | 80–98 | 9 | 47.8 | 1875.1 | 261.6 |
| QHFlow2-l | 45–59 | 8 | 77.3 | 1.3 | 65.4 |
| | 60–79 | 9 | 49.3 | 3.8 | 58.9 |
| | 80–98 | 9 | 41.2 | 1344.2 | 261.9 |

### G.7. Zero-shot prediction on PubChem

To probe generalization beyond the training distribution, we evaluate QHFlow2 models trained on QH9-stable-id (up to 9 heavy atoms) directly on PubChemQH test molecules (CHNOF, 45–98 atoms) without retraining. Table 21 reports Hamiltonian MAE, per-atom energy MAE, and the SCF cycle ratio across three size bins.

Hamiltonian MAE stays consistent across all bins, but downstream quantities degrade beyond $\sim$80 atoms: in the 80–98 bin, energy MAE exceeds 1 eV/atom and the SCF cycle ratio rises above 100%, meaning the predicted Hamiltonian slows rather than accelerates convergence. A similar degradation reported in HELM is attributed to orbital ill-conditioning, where small Hamiltonian errors are amplified near a clustered occupied/virtual boundary. Overall, QHFlow2 transfers effectively at roughly $3\times$ the training-distribution size up to $\sim$80 atoms, with orbital ill-conditioning as the main obstacle to scaling further.

### G.8. Additional Analysis of Energy and Force Errors

We provide additional analyses to explain why energies can be substantially more accurate than forces under direct evaluation from predicted Hamiltonians. We first give a perturbative explanation for the different sensitivity of energies and forces to density errors. We then empirically verify this trend by measuring the scaling between Hamiltonian, density, energy, and force errors. Next, we compare analytic and finite-difference forces to assess the practical effect of non-self-consistency in one-shot force evaluation. Finally, we decompose Hamiltonian errors in the molecular-orbital basis to show why global elementwise Hamiltonian MAE does not fully determine downstream energy accuracy.

**Perturbative explanation of the energy–force gap.** Let $H_\star$ and $\rho_\star$ denote the reference self-consistent Kohn–Sham Hamiltonian and density, and suppose that the model predicts

$$\hat{H} = H_\star + \delta H. \tag{72}$$

Solving the generalized eigenvalue problem with $\hat{H}$ induces a predicted density

$$\hat{\rho} = \rho_\star + \delta\rho, \tag{73}$$

where first-order perturbation theory gives $\delta\rho = \mathcal{O}(\delta H)$ away from exact degeneracies or near-zero orbital gaps. Expanding the Kohn–Sham energy functional around the self-consistent density gives

$$E[\hat{\rho}] = E[\rho_\star] + \int \left.\frac{\delta E}{\delta\rho(\mathbf{r})}\right|_{\rho_\star} \delta\rho(\mathbf{r})\,d\mathbf{r} + \mathcal{O}(\|\delta\rho\|^2). \tag{74}$$

By the Kohn–Sham stationarity condition under the fixed-electron-number constraint,

$$\left.\frac{\delta E}{\delta\rho(\mathbf{r})}\right|_{\rho_\star} = \mu, \tag{75}$$

*Table 22.* Empirical error scaling between Hamiltonian, density, energy, and force errors.

| Scaling | Exponent | $R^2$ | Theory |
|---|---|---|---|
| $\delta\rho_{\mathrm{int}} \sim \delta H$ | 0.86 | 0.901 | 1 |
| $\Delta E \sim \delta\rho_{\mathrm{int}}$ | 1.91 | 0.897 | 2 |
| $\Delta F \sim \delta\rho_{\mathrm{int}}$ | 1.14 | 0.790 | 1 |
| $\Delta E \sim \delta H$ (direct) | 1.65 | 0.812 | 2 |
| $\Delta F \sim \delta H$ (direct) | 1.08 | 0.862 | 1 |

where $\mu$ is the chemical potential. Since the predicted density preserves the number of electrons, $\int \delta\rho(\mathbf{r})\, d\mathbf{r} = 0$, the first-order energy term vanishes:

$$\int \left.\frac{\delta E}{\delta\rho(\mathbf{r})}\right|_{\rho_\star} \delta\rho(\mathbf{r})\, d\mathbf{r} = \mu \int \delta\rho(\mathbf{r})\, d\mathbf{r} = 0. \tag{76}$$

Therefore, the leading energy error is second order in the density error:

$$\Delta E = E[\hat{\rho}] - E[\rho_\star] = \mathcal{O}(\|\delta\rho\|^2) = \mathcal{O}(\|\delta H\|^2). \tag{77}$$

In contrast, force errors can retain a first-order dependence on the density error. Schematically, for atom $A$, the force variation contains terms of the form

$$\Delta F_A \approx -\mathrm{Tr}\left[\delta D \frac{\partial H_{\mathrm{KS}}}{\partial \mathbf{R}_A}\right], \tag{78}$$

where $\delta D$ is the density-matrix error induced by the predicted Hamiltonian. Thus,

$$\Delta F = \mathcal{O}(\|\delta D\|) = \mathcal{O}(\|\delta\rho\|) = \mathcal{O}(\|\delta H\|). \tag{79}$$

This perturbative argument explains why energy errors can decrease faster than force errors as Hamiltonian prediction improves.

**Empirical error scaling.** We empirically test this perturbative picture by measuring how density, energy, and force errors scale with Hamiltonian error. Let $\delta H$ denote the Hamiltonian prediction error and let

$$\delta\rho_{\mathrm{int}} = \int \left|\rho_{\mathrm{pred}}(\mathbf{r}) - \rho_{\mathrm{GT}}(\mathbf{r})\right| d\mathbf{r} \tag{80}$$

denote the integrated density error. We compute these quantities across 200 QH9-stable molecules using four QHFlow2 model sizes and fit the exponent by log–log linear regression. As shown in Table 22, the density error scales approximately linearly with Hamiltonian error. More importantly, the energy error scales close to quadratically with the density error, while the force error scales close to linearly, supporting the perturbative explanation above.

**Analytic versus finite-difference forces.** The forces reported in our MD benchmark are analytic gradients evaluated from the density induced by the predicted Hamiltonian under the same PBE/def2-SVP setting used to generate the MD17/rMD17 reference labels. Since this density is not guaranteed to be fully self-consistent, these analytic forces may differ from the finite-difference derivative of the one-shot energy. To estimate the practical scale of this effect, we compare analytic forces with finite-difference forces obtained by re-predicting the Hamiltonian at displaced geometries.

Specifically, $F_{\mathrm{anal}}^{\mathrm{pred}}$ denotes the analytic force evaluated from the predicted density, while $F_{\mathrm{FD}}^{\mathrm{pred}}$ denotes the finite-difference force obtained by re-predicting $H_\theta(\mathbf{R} \pm \delta)$ at displaced geometries. We also compute the corresponding converged-DFT quantities, $F_{\mathrm{anal}}^{\mathrm{DFT}}$ and $F_{\mathrm{FD}}^{\mathrm{DFT}}$, using the same PBE/def2-SVP setting to estimate the numerical discrepancy between analytic and finite-difference forces. In Table 23, column (2) corresponds to the force metric reported in the main benchmark, column (3) measures the discrepancy between analytic and finite-difference predicted forces, column (4) evaluates the finite-difference force induced by the predicted Hamiltonian against the converged DFT reference, and column (5) measures the intrinsic finite-difference discrepancy of converged DFT under the same numerical settings. The discrepancy between analytic and finite-difference predicted forces is comparable in scale to the finite-difference discrepancy observed for converged DFT, suggesting that the non-self-consistency contribution is not easily separable from finite-difference and DFT numerical artifacts in this setting.

*Table 23.* Analytic and finite-difference force comparison on MD17 under the PBE/def2-SVP setting. QHFlow2-m is evaluated on 10 test geometries per molecule with finite-difference step $\delta = 10^{-3}$ Å. All force values are reported as MAE in meV/Å.

| Molecule | (1) $\Delta E$ (meV) | (2) $F_{\text{anal}}^{\text{pred}}$ vs. $F_{\text{anal}}^{\text{DFT}}$ (meV/Å) | (3) $F_{\text{anal}}^{\text{pred}}$ vs. $F_{\text{FD}}^{\text{pred}}$ (meV/Å) | (4) $F_{\text{FD}}^{\text{pred}}$ vs. $F_{\text{anal}}^{\text{DFT}}$ (meV/Å) | (5) $F_{\text{FD}}^{\text{DFT}}$ vs. $F_{\text{anal}}^{\text{DFT}}$ (meV/Å) |
|---|---|---|---|---|---|
| Ethanol | $0.003 \pm 0.001$ | $0.24 \pm 0.08$ | $0.52 \pm 0.09$ | $0.42 \pm 0.15$ | $0.40 \pm 0.13$ |
| malondialdehyde | $0.006 \pm 0.003$ | $0.51 \pm 0.10$ | $0.94 \pm 0.25$ | $0.70 \pm 0.27$ | $0.74 \pm 0.27$ |
| Uracil | $0.018 \pm 0.009$ | $0.69 \pm 0.36$ | $0.86 \pm 0.34$ | $0.43 \pm 0.07$ | $0.41 \pm 0.02$ |

*Table 24.* MO-basis block-wise Hamiltonian error decomposition of QHFlow2-m. For each block, we report the elementwise MAE of $\Delta H^{\text{MO}}$ in the reference molecular-orbital basis. The AO overall error denotes the elementwise MAE in the atomic-orbital basis before rotation. All errors are reported in meV, except for vv/oo ratios.

| Setting | Block | Ethanol | malondialdehyde | Uracil | QH9-stable |
|---|---|---|---|---|---|
| w/o MinAO | AO (overall) | 0.340 | 0.402 | 0.372 | – |
| | oo | 0.402 | 0.397 | 0.384 | – |
| | ov | 1.415 | 2.157 | 2.611 | – |
| | vv | 5.345 | 10.864 | 18.050 | – |
| | vv / oo | 13.3× | 27.3× | 47.0× | – |
| w/ MinAO | AO (overall) | 0.072 | 0.055 | 0.050 | 0.255 |
| | oo | 0.106 | 0.068 | 0.074 | 0.270 |
| | ov | 0.192 | 0.208 | 0.259 | 1.024 |
| | vv | 0.899 | 1.353 | 2.091 | 9.778 |
| | vv / oo | 8.5× | 19.9× | 28.1× | 36.2× |

**MO-basis block-wise error decomposition.** Finally, we analyze how Hamiltonian errors project onto occupied and virtual molecular-orbital subspaces. We rotate the Hamiltonian error into the reference molecular-orbital basis as

$$\Delta H^{\text{MO}} = \mathbf{C}_{\text{ref}}^{\top} \Delta H \mathbf{C}_{\text{ref}}, \tag{81}$$

and decompose it into occupied–occupied (oo), occupied–virtual (ov), and virtual–virtual (vv) blocks. For each block, we report the elementwise mean absolute error,

$$\text{MAE}_{ab} = \frac{1}{|\Omega_{ab}|} \sum_{(p,q) \in \Omega_{ab}} \left| \left( \Delta H^{\text{MO}} \right)_{pq} \right|, \qquad a, b \in \{o, v\}, \tag{82}$$

where $\Omega_{ab}$ denotes the set of matrix entries belonging to block $ab$. The AO overall error is the elementwise MAE computed in the atomic-orbital basis before the MO-basis rotation.

To first order, the energy error is governed primarily by the occupied subspace. In contrast, the virtual–virtual block does not directly contribute to the first-order energy correction, although it contains a large fraction of matrix elements and can dominate global elementwise Hamiltonian MAE. Table 24 shows that the vv block is substantially larger than the oo block across both initialization settings. For example, in the without-MinAO setting, the vv error is up to $47.0\times$ larger than the oo error. This explains why global Hamiltonian MAE can be dominated by virtual-space errors and therefore does not always predict downstream energy accuracy. In particular, a model with larger global Hamiltonian MAE can still achieve lower energy error if its occupied-space error is smaller.

### G.9. Geometry Optimization and Local PES Analysis

We further evaluate whether forces obtained from predicted Hamiltonians reproduce the local DFT potential energy surface (PES), beyond force MAE on MD snapshots. We test this through geometry optimization, local curvature analysis using finite-difference Hessians, and a diagnostic comparison between model-reported and DFT-recomputed residual forces.

**Geometry optimization.** We follow a nablaDFT-style protocol by optimizing 20 MD17 test structures per molecule with BFGS until $f_{\text{max}} < 50$ meV/Å. QHFlow2-m and NequIP are trained on the MD17 25k split, and the optimized structures

*Table 25.* Geometry optimization on MD17 under a nablaDFT-style protocol. We optimize 20 test structures per molecule with BFGS until $f_{\max} < 50$ meV/Å. All metrics are evaluated by PBE/def2-SVP DFT single-point calculations. res MAE is the residual energy relative to the DFT-optimized structure; $\text{pct}_T$ is the fraction of DFT energy decrease recovered; succ. denotes the fraction within 1 kcal/mol; $|\Delta f|$ is the difference between model and DFT $f_{\max}$ at the optimized geometry.

| Mol. | Model | res MAE (meV) ($\text{pct}_T$) | succ. | RMSD (Å) | $|\Delta f|$ (meV/Å) |
|---|---|---|---|---|---|
| Ethanol | QHFlow2-m | **0.013** (100.00%) | **100%** | **0.0002** | **0.2** |
| | NequIP | 0.748 (100.13%) | 100% | 0.0064 | 18.2 |
| | xTB | 51.478 (90.59%) | 0% | 0.0277 | 804.9 |
| | MMFF | 57.032 (89.52%) | 0% | 0.0360 | 1136.8 |
| Malon. | QHFlow2-m | **0.001** (100.00%) | **100%** | **0.0000** | **0.4** |
| | NequIP | 0.416 (100.08%) | 100% | 0.0026 | 17.0 |
| | xTB | 109.583 (79.70%) | 0% | 0.0422 | 1702.6 |
| | MMFF | 216.606 (59.81%) | 0% | 0.0683 | 2972.3 |
| Uracil | QHFlow2-m | **0.006** (100.00%) | **100%** | **0.0002** | **0.5** |
| | NequIP | 0.829 (100.10%) | 100% | 0.0044 | 19.1 |
| | xTB | 103.078 (86.62%) | 0% | 0.0169 | 1611.2 |
| | MMFF | 181.204 (76.46%) | 0% | 0.0232 | 2501.6 |

*Table 26.* Local PES analysis on MD17. Both QHFlow2-m and NequIP are trained on the MD17 25k split. Each model is used to relax geometries to $f_{\max} < 0.001$ eV/Å. Hessians are computed by finite differences with displacement step $\delta = 10^{-2}$ Å. The relative Hessian error is $\|\mathbf{H}_{\text{model}} - \mathbf{H}_{\text{DFT}}\|_F / \|\mathbf{H}_{\text{DFT}}\|_F \times 100$.

| Metric | Model | Ethanol | malondialdehyde | Uracil |
|---|---|---|---|---|
| Freq. MAE (cm$^{-1}$) | QHFlow2-m | **0.12** ± 0.01 | **0.08** ± 0.01 | **0.09** ± 0.01 |
| | NequIP | 608.1 ± 0.3 | 508.5 ± 3.0 | 466.0 ± 0.1 |
| Hessian rel. err. (%) | QHFlow2-m | **0.020** ± 0.002 | **0.040** ± 0.002 | **0.040** ± 0.004 |
| | NequIP | 89.03 ± 0.01 | 88.92 ± 0.01 | 89.03 ± 0.00 |
| Hessian MAE (eV/Å$^2$) | QHFlow2-m | **0.0009** ± 0.0001 | **0.0021** ± 0.0002 | **0.0020** ± 0.0003 |
| | NequIP | 2.811 ± 0.224 | 3.612 ± 0.300 | 2.732 ± 0.090 |
| Hessian RMSE (eV/Å$^2$) | QHFlow2-m | **0.0015** ± 0.0001 | **0.0038** ± 0.0002 | **0.0040** ± 0.0004 |
| | NequIP | 6.799 ± 0.013 | 8.331 ± 0.016 | 8.824 ± 0.002 |
| Hessian Max (eV/Å$^2$) | QHFlow2-m | **0.010** ± 0.001 | **0.031** ± 0.007 | **0.040** ± 0.007 |
| | NequIP | 48.81 ± 1.41 | 62.90 ± 4.25 | 82.68 ± 0.35 |
| NequIP / QHFlow2 ratio | Freq. MAE | 5,060× | 6,192× | 4,984× |
| | Hessian MAE | 3,123× | 1,720× | 1,366× |

are evaluated by PBE/def2-SVP DFT single-point calculations. As shown in Table 25, QHFlow2-m achieves near-zero residual energies and much smaller force discrepancy $|\Delta f|$ than NequIP, indicating that its optimized geometries lie closer to the DFT PES.

**Local PES curvature.** We next compare local curvature near optimized geometries. Each model relaxes structures to $f_{\max} < 0.001$ eV/Å, and Hessians are computed by finite differences of model forces with displacement step $\delta = 10^{-2}$ Å. We compare the resulting Hessians and vibrational frequencies against the DFT reference. As shown in Table 26, QHFlow2-m yields sub-cm$^{-1}$ frequency errors and below $0.05\%$ relative Hessian error, whereas NequIP shows substantially larger curvature errors despite competitive force MAE on MD trajectories. DimeNet produced substantially larger errors and is omitted for clarity.

**Loose-threshold optimization diagnostic.** Finally, we compare the model-reported residual force with the DFT-recomputed residual force at geometries optimized with the same loose threshold used above, $f_{\max} < 50$ meV/Å. This diagnostic separates optimization success from PES fidelity: two models may both satisfy their own stopping criteria, while only one agrees with DFT at the resulting geometry. As shown in Table 27, QHFlow2-m's reported $f_{\max}$ closely matches the DFT-recomputed $f_{\max}$, whereas NequIP exhibits a gap of about 16–19 meV/Å.

*Table 27.* Geometry optimization diagnostic on MD17. Both QHFlow2-m and NequIP are trained on the MD17 25k split and optimized from MD17 snapshots using BFGS with threshold $f_{\max} < 50$ meV/Å. Model $f_{\max}$ denotes the residual force reported by the model at convergence, while DFT $f_{\max}$ denotes the residual force recomputed by DFT at the same geometry.

| Metric | Model | Ethanol | Malon. | Uracil |
|---|---|---|---|---|
| Success rate | QHFlow2-m | 100% | 100% | 100% |
| | NequIP | 100% | 100% | 100% |
| Model $f_{\max}$ (meV/Å) | QHFlow2-m | $41.0 \pm 6.3$ | $41.5 \pm 6.3$ | $41.3 \pm 6.0$ |
| | NequIP | $40.4 \pm 8.8$ | $41.6 \pm 7.0$ | $37.6 \pm 12.5$ |
| DFT $f_{\max}$ (meV/Å) | QHFlow2-m | $40.9 \pm 6.3$ | $41.5 \pm 6.2$ | $41.3 \pm 6.0$ |
| | NequIP | $22.1 \pm 3.8$ | $22.4 \pm 3.6$ | $21.2 \pm 4.7$ |
| \|Model $-$ DFT\| (meV/Å) | QHFlow2-m | 0.1 | 0.0 | 0.0 |
| | NequIP | 18.3 | 19.2 | 16.4 |

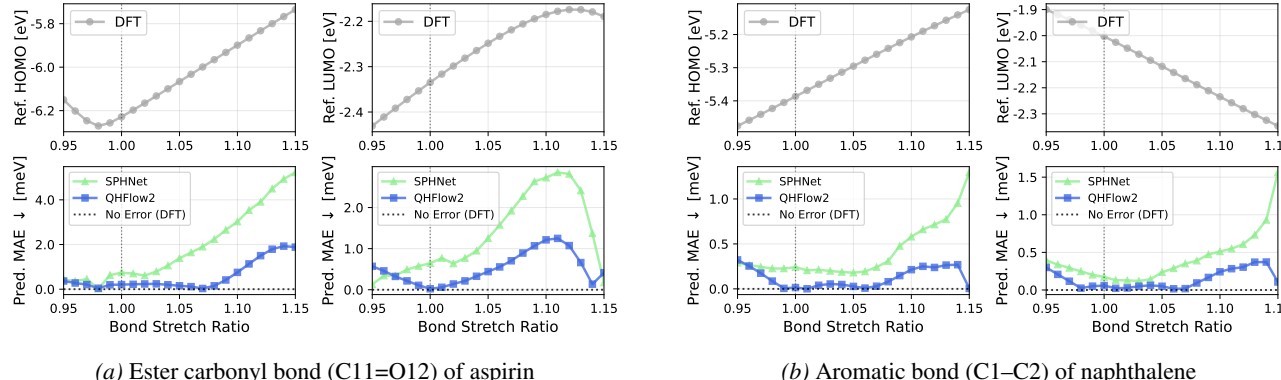

*(a) Ester carbonyl bond (C11=O12) of aspirin*  *(b) Aromatic bond (C1–C2) of naphthalene*

*Figure 8.* Robustness of frontier molecular orbital (FMO) predictions under reactive bond stretching. The plots show the MAE of HOMO and LUMO energies computed from predicted Hamiltonians, relative to DFT values, as the target bond length is scaled from $0.95\times$ to $1.15\times$ the equilibrium length. The two cases correspond to the ester carbonyl bond (C11=O12) of aspirin and the aromatic C1–C2 bond of naphthalene.

### G.10. Extrapolation on Molecular Reactive Sites

We further examine whether Hamiltonian predictors can support electronic-structure-based reactivity analysis, which is not directly accessible to standard MLIPs. We consider frontier molecular orbital (FMO) energies, *i.e.*, HOMO and LUMO energies, which are widely used as global reactivity descriptors in conceptual DFT (Fukui, 1982; Geerlings et al., 2003). Since MLIPs predict only energies and forces, they do not directly provide orbital- or density-based descriptors, whereas Hamiltonian predictors can recover these quantities from the predicted electronic structure.

We design a bond-stretching experiment on two molecules from the MD benchmark: the ester carbonyl bond (C11=O12) in aspirin, corresponding to an electrophilic site, and the aromatic C1–C2 bond in naphthalene, corresponding to a nucleophilic site. For each molecule, we scale the target bond length from $0.95$ to $1.15$ times its reference length and evaluate the HOMO and LUMO energies obtained from the predicted Hamiltonian. We compare the predicted FMO energies against DFT values and report the MAE along the stretching coordinate.

As shown in Figure 8, QHFlow2 more robustly reproduces DFT-level FMO trends than SPHNet under bond stretching. Both models achieve low errors near the equilibrium geometry, but SPHNet deteriorates rapidly in the extrapolation regime, especially for stretch ratios larger than $1.05$. In contrast, QHFlow2 maintains consistently low errors across the full range. For HOMO energies, QHFlow2 keeps the MAE below $2.0$ meV even at the largest distortion, while SPHNet exceeds $5.0$ meV. Similar behavior is observed for LUMO energies. These results suggest that QHFlow2 captures electronic variations more robustly beyond equilibrium geometries, supporting its use in orbital-based reactivity analysis.

### G.11. Direct density matrix prediction vs. Hamiltonian prediction

A natural alternative to predicting the Hamiltonian $\mathbf{H}$ is to predict the density matrix $\mathbf{D}$ directly, since both share the same SO(3) symmetry. We compare three strategies under the same QHFlow2 architecture and matched parameter counts: $\mathbf{H}$ only (default), $\mathbf{D}$ only, and joint $\mathbf{H} + \mathbf{D}$ with separate heads. For $\mathbf{D}$-only and $\mathbf{H} + \mathbf{D}$, downstream quantities are evaluated from the predicted $\mathbf{D}$ with the SCF initialized from the superposition of atomic densities (SAD).

As shown in Table 28, Hamiltonian prediction is substantially more effective. At 44M parameters, $\mathbf{H}$-only achieves $0.7\,\mathrm{meV}$ energy MAE and a $51.8\%$ SCF cycle ratio, while $\mathbf{D}$-only yields $494.3\,\mathrm{meV}$ and $69.4\%$. Joint prediction does not help. This supports our choice of $\mathbf{H}$ as the prediction target.

*Table 28.* **Direct density matrix prediction vs. Hamiltonian prediction on QH9-stable-id.** Comparison of three training strategies under the same QHFlow2 architecture at matched parameter counts. For $\mathbf{D}$-only and $\mathbf{H} + \mathbf{D}$, the SCF is initialized from the superposition of atomic densities (SAD).

| Setting | Params (M) | H MAE ($\mu$Ha) | D MAE ($\mu$) | E MAE (meV) | $\epsilon_{\mathrm{occ}}$ ($\mu$Ha) | SCF cycle ratio $\downarrow$ |
|---|---|---|---|---|---|---|
| H only (default) | 12.3 | 16.3 | – | 1.4 | 94 | 55.6% |
| H only (default) | 44.1 | 9.2 | – | 0.7 | 49 | 51.8% |
| D only | 12.3 | – | 403 | 716.6 | 5055 | 72.7% |
| D only | 44.1 | – | 224 | 494.3 | 2855 | 69.4% |
| H + D (sep. heads) | 12.5 | 44.0 | 466 | 1211.6 | 6427 | 74.5% |
| H + D (sep. heads) | 44.5 | 24.0 | 253 | 490.2 | 3019 | 70.2% |

