# OpenReview forum: "Machine Learning Hamiltonians are Accurate Energy-Force Predictors"
_ICML.cc/2026/Conference — ICML 2026 regular_

### Official Review · Reviewer_k1UC · 2026-03-04

**Soundness:** 3
**Presentation:** 3
**Significance:** 3
**Originality:** 3
**Overall Recommendation:** 5
**Confidence:** 4

**Summary:**

The paper proposes QHFlow2, a SO(2)-equivariant flow matching model used to predict Hamiltonians. The model uses the eSEN model as a backbone, then performs a two-stage pair update for all-atom-pair coupling and finally an equivariant vector-to-matrix construction of the Hamiltionian. The model is compared to other MLH models well as some MLIPs on the downstream calculated energy/forces. Across various benchmarks, the authors show that QHFlow2 is strong for hamiltonian prediction, downstream tasks and scalable.

**Compliance With Llm Reviewing Policy:**

Affirmed.

**Final Justification:**

Thank you to the authors for the rebuttal, and to the AC, SAC, and PCs for handling the discussion.

Based on the paper and rebuttal, my current assessment is as follows:
The paper is well written, technically solid, and supported by a good range of experiments and analyses.

With the rebuttal, the authors addressed my concerns adequately with good explanations and additional experiments. Showing that QHFlow2 scales well to the nablaDFT dataset is useful, and the additional wallclock timings plus scaling experiment provide some clarity. These additional contributions provide insight into the technical performance that I was previously missing and made me view the work more positively after the rebuttal, which is why I raised my score.

Outside of these points, I think the paper has a good impact and originality with some minor limitations around how the model scales to larger structures and whether it could have been applied to periodic structures.

Overall, I believe that it will be a nice addition to the field of Hamiltonian prediction.

**Key Questions For Authors:**

- What are the relative time scales of your SCF reduction experiments? Ie. what are the wall clock time savings of your initialization?
- How does QHFlow compare to QHFlow2 when scaled up?
- What are the accuracies and timings of simply using PySCF for the entire energy/forces calculation? Are QHFlow2 results skewed by PySCF compensating during the calculation?
- Could authors test how the method could be used across different basis sets/functionals, using e.g. a lightweight adaptation as suggested?
- Can authors test QHFLow2 on non-molecular benchmarks?

**Limitations:**

Yes

**Strengths And Weaknesses:**

## Strengths ##
The paper is very well written with clear descriptions of the mathematics and architecture. The figures are also designed well, making it easy to tell Hamiltonian models from regular MLIPs.

The presented architecture is novel and well justified, showing across representative benchmarks that it performs very well for Hamiltonian prediction.

The benchmarks are thorough and provide a good insight into how the model compares both on direct and downstream tasks. The inclusion of the SCF reduction experiment complements the other experiments. Overall QHFlow2 beats the reference models across the board while providing good scaling.

The models performance on downstream energy and forces (Tables 14-15) are very impressive and speak to the physically sound nature of predicting the Hamiltonian and then solving for the targets. In combination with GPU accelerated PySCF, this framework enables fast and accurate downstream calculations.

## Weaknesses: ##
The main weakness of the paper is the limited applicability of QHFlow2 in its current form. The model is only studied under a fixed functional and basis choice and only on molecular datasets. The authors do acknowledge these limitations and suggest avenues to remediate them, but I think it would have been appropriate to therefore test e.g. the multimodal training (although it was not clear exactly what was meant by that) or lightweight adaptations that the authors themselves suggest.

On a related note, there has been research in the application of Hamiltonian learning to periodic systems. Testing QHFlow2 on such benchmarks would strengthen the paper and justify its applicability beyond the molecular domain.

some other issues:

Since you were inspired by eSEN, it would be useful to see how QHFlow2 compares on energy/force against it. Especially so because eSEN is one of the best models for structure to EF on the Open Molecules Leaderboard.

When you present the SCF reduction experiment, I think some context is missing that would help a reader understand the impact. What are the relative time scales? Ie. how much of a wall-clock time reduction does the SCF reduction present?

You tested on relatively small systems (up 29), further tests on larger systems like the ones in the NablaDFT benchmark would have strengthened the paper.

Figure 6 & 7 are somewhat limited in the information they convey. Without any reference points for the larger comparison models (QHNet and QHFlow), it is difficult to tell if QHFlow’s scaling is truly better. This is especially significant in the right plot of Figure 6.

The top row of Figure 5 is too small for the amount of information presented.

---

> ### Author Rebuttal · Authors · 2026-03-31
>
> Thank you for your time in reviewing our manusript. We plan to incorporate the new experimental results and associated discussions in our future manuscript. All supplementary tables and figures for the rebuttal are available at [this anonymous link](https://anonymous.4open.science/r/icml2026-rebuttal-1C53/README.md). We apologize for the inconvenience of using external links due to the character limit.
>
> **W1/Q4. Functional and basis set transferability**
>
> We agree that this is an important direction. Our current experiments cover PBE (MD17/rMD17), B3LYP (QH9), and $\omega$B97X-D ($\nabla^2$DFT, see W3), demonstrating that QHFlow2 trains successfully across different functionals without architectural modification. However, systematically evaluating transferability within a single model requires generating new DFT reference data at multiple levels of theory for the same molecular geometries, which was not feasible within the rebuttal period.
>
> Regarding the multimodal training and lightweight adaptation mentioned in our paper, we envision conditioning the model on the functional type via adaptive layer normalization (analogous to adaLN in DiT) or using shared backbones with functional-specific heads. We consider building multi-functional, multi-basis MLH models to be one of the most important directions for future work and will clarify the proposed approach in the revised Limitations section.
>
> ---
>
> **W2/Q5. Periodic systems**
>
> We agree that periodic systems are an important direction. QHFlow2's message passing operates on local atomic graphs with a finite cutoff, which is compatible with periodic boundary conditions in principle. The main extensions required are periodic pair construction for off-diagonal blocks and k-point sampling for the evaluation pipeline. We note that existing periodic MLH work (DeepH, SLEM) operates on similar local graph structures, suggesting that the adaptation is feasible. We did not include periodic experiments in the current work as our focus was on establishing the E/F evaluation framework for molecular systems, but we consider this a natural next step and will include a concrete discussion of the required modifications in the revised manuscript.
>
> ---
>
> **W3. eSEN energy/force comparison**
>
> We thank the reviewer for this suggestion. Since eSEN does not have reported results on MD17/rMD17, we trained eSEN-S (449K) and eSEN-M (2.5M) ourselves under the same evaluation protocol as other MLIP baselines. As shown in Table R4-1, force MAE decreases consistently with increasing training size for both model scales, confirming that the training setup is well calibrated. We will update Tables 14 and Figure 5 accordingly in the revised manuscript.
>
>
>
> ---
>
> **W4/Q1/Q3. SCF reduction and wall-clock time**
>
> We agree that wall-clock time is essential for interpreting the SCF reduction results. Regarding whether PySCF compensates during the calculation: all SCF runs use identical solver settings and convergence criteria ($\texttt{convtol}=10^{-9}$, DIIS configuration, grid level); only the initialization differs. Therefore, differences in SCF cycles directly reflect the quality of the initial guess rather than solver-side adjustments.
>
> We also note that, prompted by Reviewer UB1C, we identified a functional mismatch in our GPU-accelerated SCF evaluation: the B3LYP definition used VWN3 correlation instead of VWN5, which is the parametrization used to generate the QH9 dataset. This is a well-known source of discrepancy across quantum chemistry packages. The energy/force results in Tables 1-2 were computed with the correct functional and are not affected. After correcting this, we obtain updated SCF results with wall-clock times (Table C-3 and Figure C-2). All timings measured on a single NVIDIA H200 with sequential execution for fair comparison. We will update **Figure 4** and **Section 4.3** accordingly.
>
>
> ---
>
> **W5. NablaDFT benchmark**
>
> We train QHFlow2 on $\nabla^2$DFT train-10k split. QHFlow2-s already outperforms baselines trained on the 10k split dataset over 1,920 GPU hours (Table C-2) with only ~96 GPU houes. We plan to update throughout the discussion phase, if given the chance.
>
> ---
>
> **W6/Q2. QHFlow vs QHFlow2 scaling comparison (Figure 6)**
>
> We fully agree that the original Figure 6 lacked reference points for a fair scaling comparison. To address this, we trained QHFlow at 60M parameters and present the updated comparison in Figure R4-2.
>
> The results confirm that QHFlow2 consistently outperforms QHFlow at every comparable parameter budget, in both H MAE and occupied orbital energy MAE. We were unable to scale QHFlow further due to its >2~3x higher training cost at matched parameter count, and QHNet could not be scaled beyond ~40M due to training instability (NaN divergence even with learning rate warmup and gradient clipping). We will update Figures 6 and 7 with these additional data points in the revised manuscript.

---

> > ### Author Rebuttal · Reviewer_k1UC · 2026-04-01
> >
> > Thank you for your response. I believe that my questions have been sufficiently answered and will raise the score 4->5.

---

> > > ### Author Response · Authors · 2026-04-03
> > >
> > > We sincerely thank the reviewer for the thoughtful and constructive feedback throughout the review process. Your questions helped us strengthen the paper, and we are glad that our responses have addressed your concerns. We will incorporate all discussed revisions into the camera-ready version.

---

### Official Review · Reviewer_v58j · 2026-03-12

**Soundness:** 2
**Presentation:** 2
**Significance:** 3
**Originality:** 2
**Overall Recommendation:** 3
**Confidence:** 3

**Summary:**

This paper proposes QHFlow2, a machine learning Hamiltonian model with an SO(2)-equivariant backbone and a two-stage edge update that can achieve 40% lower Hamiltonian error than the previous best model with fewer parameters. This model reaches NequIP-level force accuracy while achieving up to 20× lower energy MAE and reduces energy
error by up to 20× compared to MACE.

**Compliance With Llm Reviewing Policy:**

Affirmed.

**Final Justification:**

The rebuttal provides additional analysis that addresses some of my earlier concerns, which I appreciate. However, there are still several points that leave me a bit unconvinced about accepting this work at a top-tier conference. First, the comparison with QHNetV2 remains insufficient. Given the conceptual similarities between the two methods, this is an important missing component, and it makes it difficult to assess their relative effectiveness. I understand that this may be partly due to the lack of publicly available code for QHNetV2. Second, in the geometry optimization experiments on the nablaDFT dataset, the reported divergence rate is relatively high (~40%, comparable to SchNet), and the success rate is comparatively low, falling behind several methods in the benchmark. These results do not strongly support the claim that MLH serves as an accurate force predictor. Given these remaining concerns, I am inclined to maintain my original score.

**Key Questions For Authors:**

Please refer to the weakness part

**Limitations:**

Limitations are well discussed

**Strengths And Weaknesses:**

**Strengths:**
- This paper proposes a scalable SO(2)-equivariant MLH with two-stage edge updates, achieving better accuracy and robustness with fewer parameters.
- Construct a unified benchmark with direct energy/force evaluation from predicted Hamiltonians for fair MLIP comparison.
- Scaling analysis shows that increasing model/data scale reduces Hamiltonian error and improves energy/force accuracy.

**Weaknesses:**
- The novelty lies mainly in engineering improvements and architectural integration, rather than introducing fundamentally new modeling principles or theoretical insights for Hamiltonian learning.
- MLIPs already perform extremely well on small datasets like MD17 and it only contains small-molecules systems, thus demonstrating advantages there does not necessarily translate to real quantum chemistry systems.
- More extensive experimental validation and comparisons would strengthen the paper. For example, the authors could evaluate energy and force prediction performance on large-scale datasets such as nablaDFT [1], OC20, and OMol25 [2]. Additionally, the claimed improvements in energy and force accuracy should be further validated through downstream tasks, such as geometry optimization and relaxation to obtain high-quality molecular geometries for property prediction.
- This work should compare with QHNetV2 [3], a recent Hamiltonian prediction model that also employs SO(2)-based equivariant operations. Given the conceptual similarities between the two approaches, such a comparison would help better contextualize the proposed method and clarify its advantages in terms of accuracy, efficiency, and robustness.

[1] nablaDFT: A Universal Quantum Chemistry Dataset of Drug-Like Molecules and a Benchmark for Neural Network Potentials, NeurIPS 2024\
[2] The Open Molecules 2025 (OMol25) Dataset, Evaluations, and Models \
[3] Efficient prediction of SO (3)-equivariant Hamiltonian matrices via SO (2) local frames

---

> ### Author Rebuttal · Authors · 2026-03-31
>
> Thank you for your time in reviewing our manusript. All supplementary tables and figures for the rebuttal are available at [this anonymous link](https://anonymous.4open.science/r/icml2026-rebuttal-1C53/README.md).  We apologize for the inconvenience of using external links due to the character limit.
>
>
> **W1. Novelty lies in engineering improvements and architectural integrations, not modeling principles or theoretical insights.**
>
> We believe the paper's main novelty lies in the observation that MLH models can serve as accurate E/F predictors. Furthermore, our architectural improvement is a modeling principle to resolve the specific bottlenecks we encountered during research. The Hamiltonian naturally decomposes into on-atom terms (diagonal blocks) and inter-atomic couplings (off-diagonal blocks). The two-stage pair update explicitly models this structure: it first constructs pair features via SO(2) convolution, then refines them through SO(3) tensor product coupling to capture the angular dependence of inter-atomic orbital interactions. This change is significant; Without this stage, even when Hamiltonian MAE is sufficiently low, the downstream orbital energy is inaccurate (Table 10).
>
> ---
>
> **W2. MLIPs already perform well on MD17**
>
> We agree that MLIPs perform well on MD17 for forces. However, the energy improvement from ~1 meV (NequIP) to ~0.04–0.08 meV (QHFlow2-m) has direct chemical significance: at 300 K (k_BT ≈ 25.9 meV), a 1 meV barrier error translates to ~8% rate constant error via k ∝ e^{-ΔE‡/k_BT}, which compounds in multi-step pathways and conformational ranking. QHFlow2 is the first Hamiltonian model to reach NequIP-level force accuracy while achieving this 10–20× energy improvement (Figure 2), moving MLH into a regime where chemically meaningful differences become resolvable. Even on benchmarks where MLIPs already perform well, establishing that MLH can match or surpass this level is a necessary foundation before extending to larger systems and broader applications. In W3, we further validate these advantages with local PES analysis comparing MLH against MLIPs.
>
> ---
>
> **W3. Larger datasets and downstream validation**
>
> We agree that broader validation is needed beyond the downstream applications already shown (SCF acceleration in Figure 4, reactivity analysis in Figure 8). Extension to periodic systems (e.g., OC20/OMol25) requires periodic boundary conditions, which we view as an important but separate direction. To directly address scalability and downstream utility, we provide new experiments:
>
> **(1) Zero-shot generalization to larger molecules.**
>
> We evaluate QH9-stable-id models on PubChemQH (45-98 atoms) without retraining (Table C-1). H MAE remains consistent up to ~80 atoms with meaningful SCF acceleration, demonstrating effective zero-shot generalization at nearly 3$\times$ the training distribution.
>
> **(2) Training on $\nabla^2$DFT.**
>
> We train QHFlow2 on $\nabla^2$DFT train-10k split. QHFlow2-s already outperforms baselines trained on the 10k split dataset over 1,920 GPU hours (Table C-2) with only ~96 GPU houes.
>
> **(3) Geometry optimization and local PES analysis**
>
> We perform geometry optimization from MD17 snapshots, comparing QHFlow2-m against NequIP. Both QHFlow2 and NequIP converge, but only QHFlow2's forces match DFT at the converged geometry, suggesting that MLH models capture the PES far more faithfully than MLIPs (Table R3-1).
>
> To quantify this, we evaluate Hessian and vibrational frequency accuracy at tightly optimized geometries. Despite comparable force MAE on MD trajectories (Figure 2, Table 14), QHFlow2 achieves up to 6,000× more accurate Hessian and vibrational frequencies than NequIP (Table R3-2).
>
> ---
>
> **W4. Comparison with QHNetV2**
>
> Both models share SO(2) Linear from eSCN and QHNet's tensor expansion for readout. The most important difference is in off-diagonal refinement. QHNetV2 uses SO(2) FFN within local frames, which is conceptually similar to our first stage (Eq. 13) where pair features are constructed by concatenating node features with SO(2) operations. QHFlow2 then applies an additional SO(3) tensor product refinement in the second stage, and our ablation (Table 10) shows that this two-stage refinement provides further improvement for off-diagonal accuracy and downstream energy. QHFlow2 additionally uses flow matching (Table 11). We note that QHNetV2 lacks eSEN's smoothed envelope and edgewise interaction blocks, so detailed implementations differ beyond the off-diagonal design.
>
> Direct comparison is complicated because QHNetV2's code is unavailable, its use of initial Hamiltonians is unclear (which significantly affects metrics, Table 13), and it does not report E/F or orbital-level quantities. Under the without-init setting, QHFlow2-m achieves lower H MAE (25.50 vs 31.50 µHa) and orbital energy MAE (279.84 vs 417.89 µHa). We will add QHNetV2's reported metrics to Tables 1–2 and clarify the architectural relationship in the Related Work.

---

> > ### Author Rebuttal · Reviewer_v58j · 2026-04-04
> >
> > Thanks for the response. However, some of my concerns remain unresolved, which I think are critical to substantiating this paper's claim.
> >
> > 1. OMol25 is not a materials dataset like OC20, so periodic boundary conditions (PBC) are not required. Instead, OMol25 is a large and highly diverse dataset with DFT-level accuracy, covering small molecules, biomolecules, metal complexes, and more. Evaluating on this dataset would provide a stronger and more comprehensive validation of the method’s effectiveness.
> >
> > 2. For geometry optimization, I believe that simulating MD alone is insufficient. An important missing evaluation is to relax molecules generated from RDKit/OpenBabel and assess their quality using metrics such as DFT forces and energy minimization rates, as done in the nablaDFT paper. It would be more informative to directly compare against the geometry optimization results reported in nablaDFT.
> >
> > 3. As QHFlow2 and QHNetV2 are closely related, a fair and consistent comparison is important. It is unclear why the QHNet results reported in QHNetV2 differ from those reported in QHFlow2; this discrepancy may stem from differences in evaluation protocols. Given that QHNet results are consistent between the QHNet and QHNetV2 papers, it would be more appropriate to adopt the QHNet evaluation protocol for QHFlow2 and then compare directly with QHNetV2.

---

> > > ### Author Response · Authors · 2026-04-05
> > >
> > > Thank you for the additional comments and concerns. We appreciate the opportunity to provide further clarification.
> > >
> > > **R1. Dataset diversity and OMol25**
> > >
> > > Regarding OMol25, we agree that additional evaluation on this dataset would provide a stronger and more comprehensive validation of the effectiveness of our method. OMol25 electron provides raw ORCA outputs (~500TB for the 4M split), from which Fock matrices can be extracted; however, downloading and building a training-ready Hamiltonian dataset at this scale was infeasible within the rebuttal timeline. We also note that the only MLH evaluated on OMol25 is HELM, using a curated OMol_CSH_58K split that has not been publicly released. We will do our best to include OMol25 evaluation after the rebuttal period.
> > >
> > > To alleviate the above concern of the reviewer, we have validated our method across multiple datasets with increasing diversity. We have validated our method across multiple datasets with increasing diversity: QH9 (130K organic molecules, 5 atom types), ∇²DFT (~10K diverse drug-like molecules, 8 atom types, up to 62 atoms), and PubChem zero-shot generalization up to ~100 atoms (Table C-1). The ∇²DFT experiment began during the rebuttal period and is ongoing:
> > >
> > > | Model | Params | GPU hours | H MAE (×10⁻³ Eₕ) |
> > > |---|---|---|---|
> > > | PhiSNet | 21M | 1,920 | 0.34 |
> > > | QHNet | 22M | 1,920 | 0.53 |
> > > | HELM | - | 1,560 | **0.059** |
> > > | QHFlow2-s | 12M | ~150 | 0.166 |
> > > | QHFlow2-m | 43M | ~150 | 0.080 |
> > >
> > > With ~1/10 the compute, QHFlow2-m already outperforms PhiSNet and QHNet and is approaching HELM. We will report final results in the revision.
> > >
> > > **R2. Geometry optimization**
> > >
> > > Following the reviewer's suggestion, we conducted geometry optimization using the nablaDFT protocol with conventional baselines including xTB, RDKit MMFF ([Table R3-3](https://anonymous.4open.science/r/icml2026-rebuttal-1C53/3_reviewer_v58j/R3-3_tight_geometry_optimization/tight_geometry_optimization.png)).
> > >
> > > QHFlow2 achieves near-zero residual energy, with \|Δf\| < 0.5 meV/Å confirming faithful DFT PES reproduction. NequIP shows pct_T > 100%, indicating its learned PES differs from the true DFT PES. While both achieve 100% success rate, the \|Δf\| gap (0.2-0.5 vs 17-19 meV/Å) shows success alone cannot distinguish PES fidelity. Conventional methods fail to reach chemical accuracy. Combined with our Hessian analysis (Table R3-2, 3,000-6,000× improvement over NequIP), QHFlow2 captures the DFT PES far more faithfully than both MLIPs and conventional methods.
> > >
> > > We also evaluated on the ∇²DFT test-traj split ([Table R3-4](https://anonymous.4open.science/r/icml2026-rebuttal-1C53/3_reviewer_v58j/R3-4_nabla2dft_geometry_optimization/nabla2dft_geometry_optimization.png)). Even with an intermediate checkpoint, QHFlow2 achieves competitive p_T (91.7%), comparable to DimeNet++ (93.22%) and GemNet-OC (92.42%). We will report full results in the final revision.
> > >
> > >
> > > **R3. QHNetV2 comparison**
> > >
> > > To directly address the reviewer's concern, we adopted QH9's official metric code and confirmed that it produces identical results to our optimized implementation on a 200-sample comparison. The QHNet number discrepancy stems from training seed and dynamics differences when retraining QHNet, not from evaluation. Our retrained QHNet achieves lower H MAE on several splits (QH9-ood, QH9-geo), confirming the training quality was maintained. For models where checkpoints are available, we will update our benchmark using those checkpoints directly. We are happy to share our evaluation code and checkpoints upon request for independent verification.
> > >
> > > We note that our main tables (Tables 1-2) use the with-init setting because baselines such as WANet and SPHNet report results under this setting. We will clearly annotate this distinction in the revision. Under the without-init setting and QHNet evaluation (Table 13), QHFlow2-m achieves lower H and orbital MAE than QHNetV2:
> > >
> > > | Model | H MAE (µHa) | Orbital ε MAE (µHa) | Coeff. (%) |
> > > |---|---|---|---|
> > > | QHNetV2 | 31.50 | 417.89 | **98.58** |
> > > | QHFlow2-m | **25.50** | **279.84** | 98.31 |
> > >
> > > Once QHNetV2's code becomes available, we plan to conduct a more thorough comparison of energy and force including ablations on the two-stage update.
> > >
> > > **Plan for final revision.**
> > >
> > > 1. **∇²DFT completion**: Finalize QHFlow2-m training on ∇²DFT and report full results, including geometry optimization on the nablaDFT test-traj split.
> > > 2. **QHNetV2 comparison & Baseline update**: Add QHNetV2's reported numbers to Tables 1-2 with clear annotation of init/without-init settings. Once code becomes available, evaluate energy/force predictions and update accordingly. Report QHNet using original checkpoints where available alongside retrained results.
> > > 3. **Architectural discussion**: Expand Method to clarify the relationship between QHFlow2 and QHNetV2.
> > >
> > > If there are additional experiments or analyses that would help address the remaining concerns, we would be happy to conduct them during the discussion period.

---

### Official Review · Reviewer_UB1C · 2026-03-13

**Soundness:** 3
**Presentation:** 2
**Significance:** 3
**Originality:** 3
**Overall Recommendation:** 4
**Confidence:** 3

**Summary:**

This paper builds on a previous model, namely QHFlow, and makes it faster and more accurate through upgrading the architecture. Hamiltonian prediction, unlike MLIP, also provides the electronic structure object that can describe the system, such as the electron density. The authors perform an extensive benchmark against other baselines, and present a new hamiltonian prediction dataset as well.

**Compliance With Llm Reviewing Policy:**

Affirmed.

**Final Justification:**

I think the limitation that energy evaluation requires a complete Fock build and gives the method access to an N^4 scaling step muddies the conclusions more than the paper currently admits. Nonetheless, I see the value in speeding up DFT and therefore cautiously recommend accepting the paper.

**Key Questions For Authors:**

Do we need L in readout to be higher than L in MP? How does that compare to L_max of the basis set for the atoms? The relationship between these Ls should be clearly laid out.

Why is starting from the REF still requires 50% of the SCF cycles? This looks like a bug or upscaling issues when the data is stored in lower precision than what’s used by quantum chemistry software. Can you elaborate more on that?

In figure 5, the energies for MLIPs seem to not converge with increasing training set size (and for some even get worse), while the forces do. Could you comment on this?

For the orbital coefficient similarity (Eq. 64), how do you handle degeneracy?

**Limitations:**

yes

**Strengths And Weaknesses:**

Strengths:

-Well benchmarked against multiple baselines and variety of models.

-The authors demonstrate increased model performance with increasing model size, which opens up possibilities of scaling up.

-The model is explained very well and easy to follow, with proper citations.


Weaknesses:

-The predicted forces in this framework are likely not conservative, as energy conservation would require the Hamiltonian/Density to be a perfect fix point of the underlying functional.yes

-The authors repeatedly claim that the hamiltonian (H) matrix can recover the energy. However in DFT, one needs not only the Fock matrix (referred to as H) but also the core-hamiltonian to get the energy. This should be explained more rigorously

-MD17 dataset should only use 1k datapoints during training, not 25k. MD trajectories of a single molecule can cover very similar conformational space which leads to test set leakage. This is also mentioned by the dataset curators

---

> ### Author Rebuttal · Authors · 2026-03-31
>
> Thank you for your time in reviewing our manusript. We plan to incorporate the new experimental results and associated discussions in our future manuscript. All supplementary tables and figures for the rebuttal are available at [this anonymous link](https://anonymous.4open.science/r/icml2026-rebuttal-1C53/README.md).
>
> **W1. Forces are not conservative**
>
> We agree and will discuss this in Limitations. However, the practical impact appears small: QHFlow2's forces are competitive with NequIP (Figure 2), and QHFlow2-xl reduces SCF iterations to ~50% (Figure 4 and C-2), indicating the predicted H is near self-consistency. For applications requiring strictly conservative forces, one can use the predicted H as an SCF initial guess and run few iterations to reach full self-consistency.
>
> ---
>
> **W2. Rigorous explanation of core-Hamiltonian necessity for energy**
>
> Our model predicts the KS Fock matrix, used only to obtain the density matrix $\mathbf{P}$ via diagonalization. The total energy is evaluated by PySCF using $\mathbf{P}$ with analytically computed quantities:
>
> $E_{\mathrm{KS}} = \mathrm{Tr}[\mathbf{P} \cdot \mathbf{H}_{\mathrm{core}}] + \frac{1}{2}\mathrm{Tr}[\mathbf{P} \cdot (\mathbf{J}[\mathbf{P}] + \mathbf{V}_{\mathrm{xc}}[\mathbf{P}])] + E_{\mathrm{nn}}$
>
> $\mathbf{H}_{\mathrm{core}}$, $\mathbf{S}$, $E_{\mathrm{nn}}$ are computed analytically (inexpensive one-electron integrals); $\mathbf{J}[\mathbf{P}]$, $\mathbf{V}_{\mathrm{xc}}[\mathbf{P}]$ are reconstructed from $\mathbf{P}$. The predicted Fock matrix does not directly enter the energy expression.
>
> ---
>
> **W3. Training data size and data scaling on MD17**
>
> We use 25k training points for consistency with prior MLH work (QHNet, QHFlow, SPHNet). We acknowledge the curators' recommendation of 1k for MLIPs, but note that Hamiltonian prediction targets a full $B \times B$ matrix, which requires denser conformational coverage than scalar energy regression.
>
> Our conclusions are robust to data reduction: QHFlow2 matches DimeNet and GemNet at 1k and reaches NequIP-level performance from 5k (Figure 5). The 25k setting is used for controlled comparison within the MLH literature. On QH9, where training geometries are independent equilibrium structures with no trajectory correlation, QHFlow2 consistently outperforms MLIP baselines (Figure 3). Additionally, new geometry optimization and Hessian experiments (Tables R3-1, R3-2) show that QHFlow2 achieves 1,000–6,000× more accurate PES curvature than NequIP — a structural benefit of deriving forces from the predicted Hamiltonian, independent of training data size.
>
> ---
>
> **Q1. Relationship between $L$ in readout, and $L_{\text{max}}$ of basis set**
>
> In QH9, orbital angular momenta range up to $l=2$ (d orbitals), so Hamiltonian blocks $\mathbf{H}^{(l_1, l_2)}$ require irreps up to $l_1 + l_2 = 4$ (d-d coupling).
>
> To isolate the effect of $L_{\text{readout}}$, we fix $L_{\text{MP}} = 4$ and vary $L_{\text{readout}}$ from 1 to 4 at matched parameter count on QH9-stable-id (Table R2-1. $L_{\text{readout}}$ must be at least max $(l_1+l_2)$; below this, large errors arise. $L_{\text{readout}} = 4$ performs best (Table R2-1).
>
> ---
>
> **Q2. Unexpectedly high remaining SCF cycles of datasets**
>
> We thank the reviewer for the raising point - the reviewer was correct to suspect a bug. Upon further investigation, We identified a functional mismatch in our GPU-accelerated SCF evaluation: the B3LYP definition used VWN3 correlation instead of VWN5, which is the parametrization used to generate the QH9 dataset. This is a well-known source of discrepancy across quantum chemistry packages. It caused a mismatch between the dataset labels and our evaluation setup. This issue only affects the QH9 SCF relative reduction benchmark on GPU; The energy/force results in Tables 1-2 were computed with the correct functional and are not affected.
>
> After correcting this, we obtain updated SCF results with wall-clock times (Table C-3, Figure C-2). We will update Figure 4 and Section 4.3 accordingly.
>
> ---
>
> **Q3. MLIP energy does not converge with data scaling**
>
> All MLIP baselines follow the MDSim [1] framework with force-dominant loss weighting ($\lambda_E : \lambda_F = 1 : 1000$). We suspect this weighting causes energy to not benefit from additional data, as the model prioritizes force accuracy. Adjusting the loss balance toward energy may yield different scaling behavior, but we kept the original setup for fair comparison with reported results.
>
> ---
>
> **Q4. Degeneracy handling in orbital coefficient similarity**
>
> We follow the same metric definition as the QH9 benchmark and prior work: absolute cosine similarity averaged over occupied orbitals without explicit degeneracy handling. To our knowledge, no prior MLH work has applied special treatment for degenerate orbitals in this metric. Exact degeneracies in the occupied subspace are rare in our benchmarks due to low molecular symmetry.
>
> [1] Fu et al., "Forces are not Enough," TMLR 2023.

---

> > ### Author Rebuttal · Reviewer_UB1C · 2026-04-03
> >
> > Thank you for addressing the comments. Note that the hamiltonian $F = h_{core} + P  J + V_{xc}[P]$. Recalculating these from diagonalizing F (the hamiltonian) can 1) alter results unless you are converged, 2) incurs the cost of one DFT step, with the same scaling as DFT due to the expensive 2-electron integral building. A more interesting evaluation would be to use $E = Tr(P (h_{core} + F)/2$, where the cost is now only building 1-e integrals.
> >
> > Given the fundamental problems that need to be resolved in this work, I will keep the score.

---

> > > ### Author Response · Authors · 2026-04-05
> > >
> > > Thank you for the detailed technical feedback and for the positive evaluation of our work. We apologize for the broken rendering in our previous response and provide the corrected version below.
> > >
> > > > Our model predicts the KS Fock matrix, which is used only to obtain the density matrix **P** via diagonalization. The total energy is then evaluated by PySCF using **P**:
> > > >
> > > > $E\_{KS} = \mathrm{Tr}[\mathbf{P} \cdot \mathbf{H}\_{core}] + \frac{1}{2}\mathrm{Tr}[\mathbf{P} \cdot (\mathbf{J}[\mathbf{P}] + \mathbf{V}\_{xc}[\mathbf{P}])] + E\_{nn}$
> > > >
> > > > where $\mathbf{H}\_{core}$, $\mathbf{S}$, and $E\_{nn}$ are computed analytically (inexpensive one-electron integrals), and $\mathbf{J}[\mathbf{P}]$, $\mathbf{V}\_{xc}[\mathbf{P}]$ are reconstructed from **P**. The predicted Fock matrix does not directly enter the energy expression.
> > >
> > > We will ensure the energy evaluation pipeline along with the necessity of $H\_\text{core}$ is clearly documented in the revised manuscript.
> > >
> > >
> > > We acknowledge that reconstructing $\mathbf{J}[\mathbf{P}]$ and $\mathbf{V}_{xc}[\mathbf{P}]$ from P incurs the cost of one Fock build due to the 2-electron integrals, as the reviewer correctly points out. This is a limitation of the current MLH pipeline, which we discuss in our Limitations section. That said, we find it notable that this single Fock build, rather than a full SCF, already yields up to 20x more accurate energies than MLIPs, highlighting the quality of the predicted electronic structure. Our work explores this range of use cases: one-shot energy/force evaluation when the prediction is sufficiently accurate, or as an SCF initial guess for full DFT acceleration when higher precision is needed (Table C-3, Figure C-2 on Q2).
> > >
> > > Beyond energy and forces, MLH additionally provides access to the full electronic structure, including orbitals, electron density, Hessians, and vibrational frequencies, quantities that MLIPs fundamentally cannot capture.
> > >
> > > We appreciate the constructive feedback regarding $E = \mathrm{Tr}(\mathbf{P}(\mathbf{h}\_{core} + \mathbf{F})/2)$. To our understanding, this expression is exact for HF but serves as an approximation in KS-DFT, as the XC energy and XC potential are not interchangeable. Nonetheless, it provides a low-cost energy estimate without requiring 2-electron integral evaluation. There are several promising directions to reduce the cost of energy/force evaluation from MLH, including delta-learning from this estimate to correct toward the KS-DFT energy, or attaching a dedicated energy/force prediction head as in HELM. We will investigate these directions in future work. Thank you.

---

### Official Review · Reviewer_QyA8 · 2026-03-13

**Soundness:** 3
**Presentation:** 3
**Significance:** 3
**Originality:** 2
**Overall Recommendation:** 5
**Confidence:** 4

**Summary:**

This paper proposes QHFlow2, which combines a flow matching based Hamiltonian predictor with an eSEN-style SO(2)-equivariant backbone, two-stage pair updates for off-diagonal blocks, and tensor expansion for SO(3)-equivariant prediction of the molecular Hamiltonian. Benchmarks on MD17/rMD17 and QH9 demonstrate improved Hamiltonian and orbital metrics over previous methods, force accuracy close to MLIPs on the MD benchmark, scaling with model/data size, and reduced time spent on downstream evaluations.

**Compliance With Llm Reviewing Policy:**

Affirmed.

**Final Justification:**

The paper is technically sound with strong empirical evaluation and a clear motivation, though its architectural novelty relative to prior work is somewhat limited. The rebuttal effectively addressed my main concerns, particularly by clarifying methodological differences, providing additional experiments on larger systems, and offering a convincing theoretical explanation for the energy-force discrepancy, which improved my confidence in both soundness and significance. The rebuttal also contains valuable insights into the model, especially regarding error behavior and representation choices, further strengthening the paper’s contribution. I therefore raise my recommendation to accept (5).

**Key Questions For Authors:**

While this is a high-quality paper with valuable insights, clarifying the following questions would strengthen the analysis. I will raise my score if the authors provide sufficient answers to the points below.

1. Compared to MLIP results, the energy error seems to be much smaller than the force error. Could the authors explain why it is the case? Related to this, the forces are computed as analytical gradients $\mathbf{F}\_A = -\nabla\_{\mathbf{R}\_A} E\_\text{KS}[\rho]$ here, which is exact when the density is self-consistent, but the density here is produced from predicted Hamiltonian, which may produce an error: $$\frac{d E}{d \mathbf{R}\_A} = \frac{\partial E\_\text{KS}}{\partial \mathbf{R}\_A} + \mathrm{Tr}\left[ \frac{\partial E\_\text{KS}}{\partial \mathbf{P}} \frac{d \mathbf{P}}{d\mathbf{R}\_A} \right]$$ where the second term is not canceled unless the solution is self-consistent and stationary. Is this inconsistency affecting the difference in energy and force errors? Could authors use finite differences to measure how much the numerical vs analytical force evaluations are different?
2. When we consider scaling up to larger systems, overlap matrix $\mathbf{S}$ used to produce density from Hamiltonian prediction could be sparse or ill-conditioned. Have authors encountered similar issues in this work? How could this be mitigated when scaling up this approach?
3. Regarding the benchmark without MinAO init (Table 12), HELM and QHFlow2 seem to have a tradeoff where HELM shows 2~3x lower H element MAE while QHFlow2-m has ~20-200x smaller energy MAE. Could the authors comment on this big difference? While it would be hard to compare between models since HELM is not available, can authors analyze how matrix errors project onto occupied/frontier subspace or analyze occupied-virtual elements with a small energy gap to provide understanding of this tradeoff behavior based on QHFlow2 results?
4. Considering the downstream prediction of energy and forces, have authors considerd predicting the density matrices directly instead of the Hamiltonian, given that they have a similar symmetry requirements?

**Limitations:**

yes

**Strengths And Weaknesses:**

### Strengths

- The paper's primary topic is whether learned Hamiltonians should be judged by physical observables, such as energies and forces that people actually care about, rather than only by matrix reconstruction metrics. This brings the field of Hamiltonian prediction a step closer to its use in molecular and materials simulations.
- The experiments and benchmarks are carefully conducted across multiple variants of MLIP architectures and previous Hamiltonian prediction methods, with helpful analyses of scaling laws for prediction accuracy and computational cost.
- In terms of presentation, the paper is generally well-written, with clear motivation and results shown in tables or bar charts that communicate the story well with sufficient visual clarity.

### Weaknesses

1. Similarity to HELM
	- HELM explicitly uses an equivariant message passing backbone based on eSEN with SO(2) convolutions over node/edge embeddings, then maps those to Hamiltonian blocks. While the major approach for energy/force prediction is different: HELM uses separate energy/force head while QHFlow2 uses energy prediction from the Hamiltonian, the general architecture seems quite similar. The flow matching idea was already introduced in QHFlow and eSEN-like trunk is already in HELM, so the originality in terms of ML approach could be a bit weak.
	- I want to note that this is not asking for a result comparison beyond Table 12 because HELM codebase seems to be unavailable yet. However, while HELM is cited elsewhere in Introduction and Related Work sections, I think the similarity here needs to be clarified in the Methods as well.
2. Limited molecule sizes
	- While the paper suggest a broad statement about Hamiltonian predictor as a practical direct energy/force prediction, the sizes of the molecules in the dataset are relatively small. For example, chemically transferable setting (QH9) has up to nine heavy atoms, and conformational split setting (rMD17) has up to 21 atoms. While this is understandable, given the extensive sizes of Hamiltonian datasets, there still exist larger datasets such as $\nabla^2$DFT has been studied by previous approaches. Hence, the results and conclusions derived in this work might pertain to relatively smaller molecules and do not generalize to molecule/materials with practical use cases (also see Questions).
3. Inference speed/cost
	- While the authors have made clear in the limitations section that the cost is still dominated by DFT evaluation, scaling on QH9 results (Table 2 and Figure 7) shows that even for a relatively small dataset size (130k), all metrics are still improving at QHFlow2-xl model with 990M parameters, which is much bigger and slower than modern equivariant universal MLIPs. Hence, while Hamiltonian predictors themselves provide access to diverse physical observables, ML Hamiltonian models, as "useful" energy/force predictors, could be hindered by their high inference cost.

---

> ### Author Rebuttal · Authors · 2026-03-31
>
> Thank you for your time in reviewing our manusript. We plan to incorporate the new experimental results and associated discussions in our future manuscript. All supplementary tables and figures for the rebuttal are available at [this anonymous link](https://anonymous.4open.science/r/icml2026-rebuttal-1C53/README.md). We apologize for the inconvenience of using external links due to the character limit.
>
> **W1. Clarification of architectural differences from HELM**
>
> While both share the eSEN backbone, they differ in how the Hamiltonian is constructed: HELM decomposes H into irrep vectors via deterministic regression, whereas QHFlow2 uses SO(3) tensor expansion (Eq. 59) within flow matching. QHFlow2 also introduces (1) a two-stage pair update (Table 10) and (2) a redesigned equivariant matrix embedding (Eq. 8-9), both absent in HELM.
>
> ---
>
> **W2. Limited molecule size of benchmarks**
>
> To resolve your concern, we provide two new experiments:
>
> **(1) Zero-shot generalization to larger molecules.**
>
> We evaluate QH9-stable-id models on PubChemQH (45-98 atoms) without retraining (Table C-1). H MAE remains consistent up to ~80 atoms with meaningful SCF acceleration, demonstrating effective zero-shot generalization at nearly 3$\times$ the training distribution.
>
> **(2) Training on $\nabla^2$DFT.**
>
> We train QHFlow2 on $\nabla^2$DFT train-10k split. QHFlow2-s already outperforms baselines trained on the 10k split dataset over 1,920 GPU hours (Table C-2) with only ~96 GPU houes. We plan to update throughout the discussion phase, if given the chance.
>
> ---
>
> **W3. Inference speed/cost**
>
> While metrics still improve at 990M, practical accuracy does not require the largest model: 12M already surpasses MLIPs in energy and orbital energy on QH9 (Figure 3), and 43M achieves lower energy MAE with comparable force accuracy on MD17 (Figure 2).
>
> Beyond cost, MLH models faithful PES representation, which we demonstrate with two new experiments:
>
> (1) Geometry optimization (Table R3-1). We optimize MD17 snapshots via BFGS with QHFlow2-m and NequIP. Both converge, but QHFlow2's forces match DFT within 0.1 meV/Å at the converged geometry, while NequIP shows ~18 meV/Å discrepancy.
>
> (2) Local PES analysis (Table R3-2). At tightly optimized geometries, QHFlow2 achieves sub-cm⁻¹ frequency accuracy and <0.05% Hessian error - 1,000–6,000× more accurate than NequIP, despite comparable force MAE (Figure 2, Table 14).
>
> ---
>
> **Q1. Energy vs force error gap and analytical vs finite-difference forces**
>
> **Theoretical analysis.** Our model predicts $\hat{H} = H_0 + δ H$, yielding a predicted density $\hat{\rho} = \rho_0 + δ\rho$ where $δ\rho \sim \mathcal{O}(δ H)$ by first-order perturbation theory.
>
> For energy, expanding the DFT energy functional:
>
> $$E[\hat{\rho}] = E[\rho_0] + \int \frac{δ E}{δ \rho(\mathbf{r})}\bigg|_{\rho_0} δ\rho(\mathbf{r})\, d\mathbf{r} + \mathcal{O}(δ\rho^2)$$
>
> By the KS stationarity condition, $\frac{δ E}{δ\rho}\big|_{\rho_0} = \mu$ (constant), so the first-order term becomes $\mu \int δ\rho \, d\mathbf{r} = 0$ since the predicted density preserves electron number. Thus $\Delta E \sim \mathcal{O}(δ\rho^2) \sim \mathcal{O}(δ H^2)$.
>
> For forces, via the Hellmann-Feynman theorem [1]:
>
> $$\Delta F = -\text{Tr}\left(δ\rho \cdot \frac{\partial H}{\partial R}\right) \sim \mathcal{O}(δ\rho) \sim \mathcal{O}(δ H),$$
>
> **Empirical verification.** As shown in Table R1-1, density error scales approximately linearly with H error (exponent 0.86). Against $δ\rho_{\text{int}}$ directly, $\Delta E \sim δ\rho^{1.91}$ and $\Delta F \sim δ\rho^{1.14}$.
>
> **Analytical vs finite-difference (FD) forces.** Following the reviewer's suggestion, we compute FD forces by re-predicting $H_\theta$ at each perturbed geometry (Table R1-2).
>
> ---
>
> **Q2. Overlap matrix ill-conditioning**
>
> Empirically, on QH9, only 1~2 cases out of ~13k testset exhibited eigenvalue collapse near the occupied/unoccupied boundary. No cases on rMD17. The root cause is orbital reordering triggered by small H errors when eigenvalues cluster near the HOMO-LUMO gap. This is a common issue across ML Hamiltonian methods, also observed and addressed in HELM. To fix it, we adjust the overlap eigenvalue threshold from $10^{-6}$ to $10^{-3}$, and apply a post-hoc method to detemine the oribtal reordering.
>
> For larger systems, ill-conditioning may become more prevalent; stabilizing occupied eigenvalues (e.g., WA loss) may help.
>
> ---
>
> **Q3. Why H element MAE does not determine energy accuracy**
>
> As we analyze this in Table R1-3. Energy depends on the occupied block, whereas our H MAE is largely driven by the virtual block, which makes no first-order contribution to energy. A model with higher H MAE can still achieve lower energy MAE if its occupied block is more accurate.
>
> ---
>
> **Q4. Direct density matrix prediction**
>
> We compare H prediction, D prediction, and joint H+D prediction at matched parameter counts (Table R1-4). We believe Hamiltonian is more
>
> [1] Feynman, Phys. Rev. 56, 340 (1939).

---

> > ### Author Rebuttal · Reviewer_QyA8 · 2026-03-31
> >
> > I appreciate the authors' detailed response. The error scaling analysis via perturbation theory is very neat. While the Q4 response seems truncated, I believe the authors intended to note that H prediction is more data efficient than D prediction. My concerns are resolved and I will raise the score accordingly. I also look forward to seeing the $\nabla^2$DFT results.

---

> > > ### Author Response · Authors · 2026-04-01
> > >
> > > Thank you for acknowledging our response and for raising the score. We are glad that our perturbation theory analysis addressed your concerns. You are correct regarding Q4, we intended to note that, in our current experimental setting, Hamiltonian prediction proves more effective than density prediction. We will include the ∇²DFT results in the revised manuscript and also update them at the anonymous link before discussion ends.

---

### Decision · Program_Chairs · 2026-04-30

**Decision:**

Accept (regular)

**Comment:**

Four reviewers have provided their evaluations to this work and all of them raised some levels of concerns during their initial reviews. During rebuttals, some issues have been resolved, but some other issues remain. The primary ones are technical novelty and experimental comparison with prior methods that the authors were not aware prior to rebuttals. The authors tried to address these concerns during rebuttals, but due to timing, they were not able to address all of them. Given the reviews of other reviewers and my own judgement, I feel this paper is a borderline one, and I am willing to recommend a conditional accept with the condition that the author will fully addressed and incorporate all revisions and comments during rebuttals in their final camera ready paper.